# Proteomic landscape of Ewing sarcoma primary tumors and metastases

Sagi Gordon[1,2,8], Vishnu Mohan [1,8], Rachel Shukrun[2,3], Ofra Golani[4], Shani Metzger[1], Osnat Sher[5], Michal Manisterski[3], Roni Oren [6], Liat Fellus-Alyagor[6], Lir Beck[2], Yoseph Addadi [4], Benjamin Dekel[2,7], Ronit Elhasid[2,3] & Tamar Geiger [1] ✉

Ewing sarcoma (EWS), a rare pediatric bone tumor, poses unique therapeutic challenges due to its distinct microenvironment and limited molecular understanding. To gain a comprehensive molecular and functional view of the tumors in their microenvironment, we perform a deep mass spectrometry-based proteomic analysis of 170 tumor samples from 74 patients from primary, relapsed, and metastatic tumors. Analysis of more than 10,000 proteins across patients reveals insights into cancer prognosis, chemo-resistance, and progression. Our analyses suggest that ferroptosis pathways may be associated with chemotherapy response in EWS, and we delineate molecular subclasses that correlate the tumor immune landscape with DNA damage repair, ubiquitin-related proteins, and patient outcomes. Multiplexed immuno-fluorescence imaging indicates possible associations between neutrophils and poorer prognosis, and between macrophages/T cells and a more favorable prognosis. Altogether, this investigation provides valuable insights into the intricate biology of EWS, paving the way for developing therapeutic strategies.

Pediatric solid tumors are typically rare tumor types, which present as embryonal tumors below the age of 10 years, and non-embryonal cancer thereafter. Their rarity, their ambiguous cellular origin, and lack of research models translate to limited molecular understanding, hindering the development of targeted therapies and prognostic biomarkers[1–3]. Ewing sarcoma (EWS) is a pediatric malignancy that primarily affects the bones and soft tissues in adolescents and young adults. It is characterized by a single chromosomal translocation event, resulting in the formation of chimeric fusion proteins involving the *EWSR1* gene and *ETS* family transcription factors. The most common fusion protein observed in EWS is EWSR1::FLI1, which plays a crucial role in driving tumorigenesis by dysregulating essential cellular processes such as transcriptional control, DNA repair, and signal transduction pathways[4–7]. Despite being the only genetic driver event, the role of the fusion protein EWSR1::FLI1 and its therapeutic potential remain enigmatic. While its expression fuels cancer cell proliferation, paradoxically, low levels were associated with higher migration and invasion[8]. This complexity suggests that solely targeting the genetic cause may be insufficient for effective treatment[4,5,9,10].

Current treatments for EWS primarily rely on chemotherapy and local treatments, including surgery and radiation. Despite these interventions, patients diagnosed at a pre-metastatic stage exhibit approximately 75% five-year survival, while those patients diagnosed with metastatic disease or relapse have a significantly reduced survival rate of less than 30%[4,5]. These significant differences in outcomes highlight the urgent clinical need to identify specific drug targets that mediate the aggressive tumor phenotypes associated with EWS[5,9].

[1]Department of Molecular Cell Biology, Weizmann Institute of Science, Rehovot, Israel. [2]School of Medicine, Tel Aviv University, Tel Aviv, Israel. [3]Department of Pediatric Hemato-Oncology, Tel Aviv Medical Center, Tel Aviv, Israel. [4]Department of Life Sciences Core Facilities, Weizmann Institute of Science, Rehovot, Israel. [5]Department of Pathology, Tel Aviv Medical Center, Tel Aviv, Israel. [6]Department of Veterinary Resources, Weizmann Institute of Science, Rehovot, Israel. [7]Pediatric Stem Cell Research Institute and Division of Pediatric Nephrology, Edmond and Lily Safra Children's Hospital, Sheba Medical Center, Ramat-Gan, Israel. [8]These authors contributed equally: Sagi Gordon, Vishnu Mohan. ✉e-mail: tami.geiger@weizmann.ac.il

Transcriptomic analyses have been instrumental in unraveling the molecular characteristics of EWS[8,11–14]. Variable mRNA expression of the fusion protein has been observed in EWS cell lines and tumors, revealing the downstream effects of these proteins and their impact on tumorigenesis[15–17]. Deconvolution of RNA expression data has provided some insights regarding the immune landscape of EWS, associating immune profiles with patient survival[18]. In general, the EWS tumor microenvironment (TME) is known to be immunosuppressive. It is characterized by low abundance of tumor-infiltrating lymphocytes (TILs) and a predominant presence of immunosuppressive cells that dampen anti-tumor responses[15,16]. Other studies suggested that poor prognosis is associated with increased neutrophils[19] and M2 macrophages[20], while a higher presence of CD8 T cells correlated with a more favorable prognosis[21,22]. However, current immuno-oncological treatments for EWS have shown limited efficacy, indicating the need for a deeper understanding of cancer-immune interactions and improved patient stratification to unlock the potential of immunotherapy for EWS patients[15–17].

While transcriptomics and genomics have greatly improved our understanding of EWS[23–26], analysis of these layers alone provides a partial snapshot of the molecular landscape, neglecting post-transcriptional regulation, a crucial driver of cellular functions. Direct protein-mRNA comparisons in clinical samples have revealed significant discrepancies in multiple biological processes and signaling pathways, leading to distinct tumor classification and functional annotation[27,28]. Thus, the application of proteomics holds great potential to unravel tumor functionalities and identify actionable drug targets.

In this work, we analyze a large EWS cohort comprising primary, post-neoadjuvant chemotherapy, relapse, and metastatic tumors, and correlate their proteomic landscapes to patient clinical profiles and the tumor immune landscape. We discover proteins associated with EWS progression and drug resistance using mass spectrometry-based proteomics of patient samples. These EWS proteomic profiles highlight potential therapeutic targets that may pave the way for effective therapies in the future.

## Results

### Proteomics of the EWS clinical cohort

Aiming to identify protein regulators of EWS progression and drug resistance, we performed deep MS-based proteomic profiling of a large cohort of EWS patients. Altogether, we analyzed 170 EWS samples, originating from 74 patients. Among those, the cohort included 61 primary surgical biopsy samples used initially for tumor diagnosis, nine post-neoadjuvant chemotherapy (NACT) surgical samples with poor response (necrosis ≤ 90%), six relapsed samples, three progression, 14 metastasis samples, and in some cases, two samples were taken from a single tumor. All tumors were localized to the bones or adjacent soft tissue (limbs, ribs, spine, sacrum, and pelvis) except for lung metastases (Fig. 1a–c). To associate between protein profiles and clinical response, we assembled all available clinical information, including survival time, relapse, treatment regimens, pathological response, and metastatic state at diagnosis (Fig. 1d and Supplementary Data 1–3). The cohort included 17 matched samples taken from the same patients at distinct progression stages of the disease (Supplementary Fig. S1a). Altogether, the size of the cohort and the clinical information set a unique opportunity to discover EWS regulators and targets.

Formalin-fixed paraffin-embedded (FFPE) tumor sections were obtained upon ethical approval from the Sourasky Medical Center, Tel Aviv, Israel. Samples were de-paraffinized and macrodissected after pathological assessment to enrich for cancer cell regions. We ensured relatively homogeneous cancer populations by dissecting small regions ranging between 5 and 12 mm², and in cases of several such regions in single tumors, we collected and analyzed separate samples

from the same tumor. Proteins were trypsin digested and analyzed on the Exploris480 mass spectrometer, using gas phase fractionation and data-independent acquisition (DIA) to increase depth and robustness[29,30]. Altogether, we identified 10,220 proteins, and an average of 7370 proteins per sample, with insignificant differences in proteomic depth among the cancer stages (Supplementary Fig. S1b and Supplementary Data 4). Similarly, insignificant differences were found in the number of identified proteins from decalcified and non-decalcified samples (Supplementary Fig. S1c). Overall sample-wise Pearson correlations ranged from 0.70 to 0.96, and biological replicates from the same tumor sections presented higher correlations of 0.85–0.96 (Supplementary Fig. S1d). Matched samples from the different stages showed a high correlation, except for the correlation between the primary tumor and post-treatment samples, which presented a median correlation of 0.81, attesting to the major impact of chemotherapy (Supplementary Fig. S1e). Consistent with the precise tissue dissection, analysis of cancer tissue purity using the ESTIMATE algorithm[31] provided an estimation of >80% tumor purity in almost all samples, including post-NACT samples, within the proteomic regions of interest (ROIs; Supplementary Fig. S1f).

Initial data evaluation focused on examining the expression levels of known EWS-related proteins. EWSR1 was identified in all samples; the three known EWS proteins NKX2.2[32], LINGO1[33] and CD99[4,34] were identified in 96%, 98%, and 79% of the samples, respectively (Fig. 1e). Among the fusion partners, FLI1, the most frequent fusion partner of EWSR1, was highly expressed in 153 samples, while ERG, the second most common fusion partner, showed high expression in five samples. In seven samples, we did not quantify FLI1 or ERG, and three samples expressed low levels of ERG. Examination of the EWS-related proteins across the different progression states showed largely comparable expression profiles (Supplementary Fig. S1g). Interestingly, the expression levels of these proteins varied considerably across primary tumors, with a range exceeding 30-fold (Fig. 1e). We observed variable and weak correlation between their expression and tumor purity (Supplementary Fig. S1h), indicating that stromal admixture accounts for a minor fraction of the variance and suggesting a high degree of heterogeneity within the cancer cells. Notably, while FLI1, ERG, and EWSR1 are central components of the oncogenic fusions that drive Ewing sarcoma biology, proteomic measurements reflect both wild-type and fusion-derived protein fragments, as the mass spectrometric data could not distinguish between them in the absence of patient-specific fusion-junction peptides.

### Identification of potential regulators of EWS progression and drug resistance

Comparing the proteomes of EWS cancer stages can unravel protein networks associated with disease progression and the aggressive cancer phenotype. Specifically, investigation of post-treatment samples can propose mechanisms of chemotherapy resistance, as the residual disease after treatment reflects resistant cells that may induce tumor relapse and metastasis, reducing survival rates to only 30%. Thus, identification of the changes that occur in the tumor proteome upon treatment can be the first step towards the development of better therapies, as alternatives or in combination with current regimens. Examination of the differences between each of the EWS progression states identified 1013 significantly changing proteins (FDR < 0.05, Fig. 2a, Source Data file). The largest differences were observed between primary tumors and post-treatment tumors, representing the dramatic change induced by chemotherapy. Among the down-regulated proteins upon treatment (Cluster 4) we found a large network of RNA processing proteins, epigenetic regulators, DNA replication and repair factors, and mitotic proteins, including topoisomerases, MSH2,3, KAT6, 7,2, etc. (Fig. 2b and Supplementary Fig. S2a). Most of these proteins were elevated upon relapse and progression, suggesting that their downregulation upon treatment

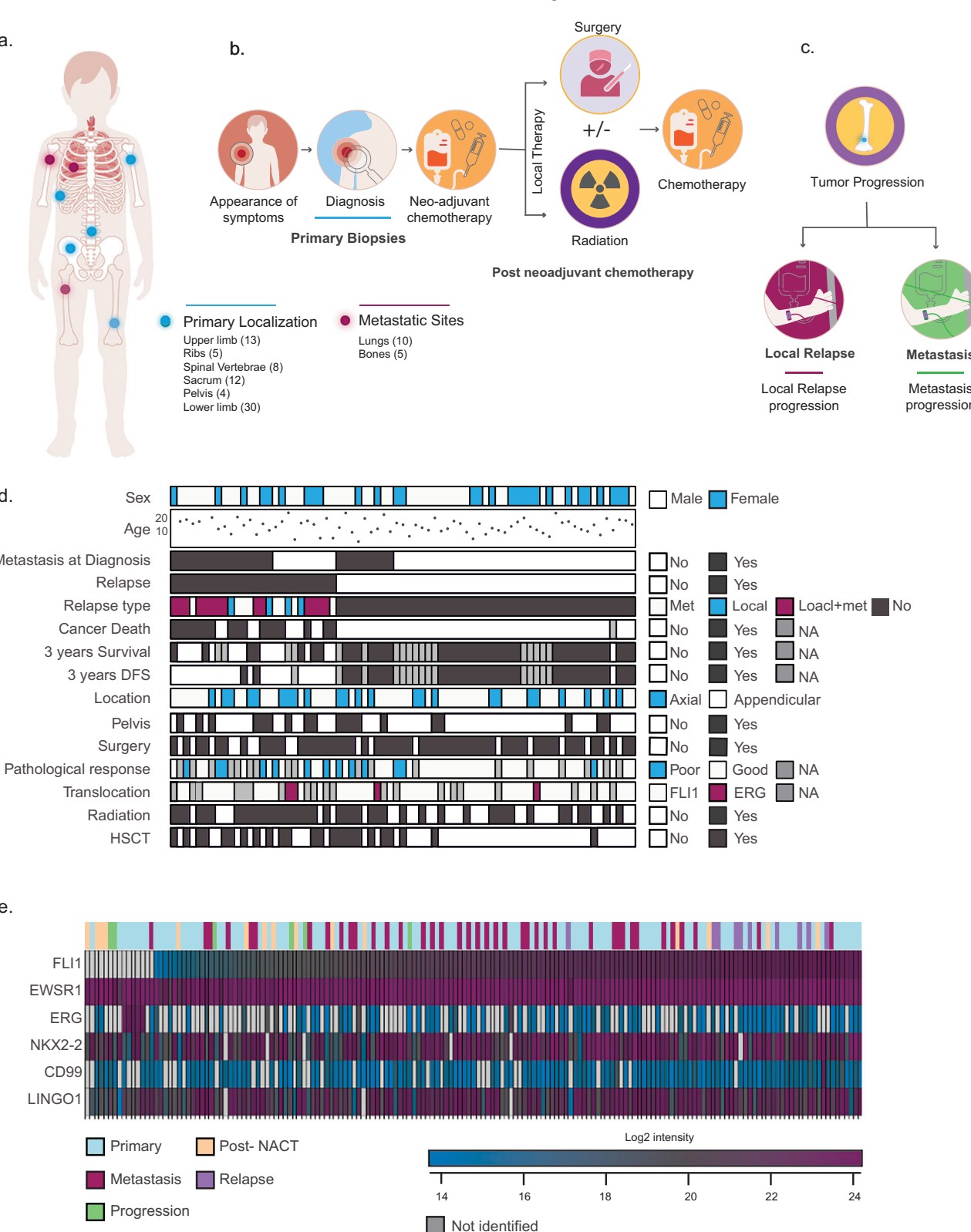

was directly impacted by the treatment, but insufficient for complete tumor eradication. Two additional clusters (2 and 3) of protein upregulated in advanced cancer stages highlighted the involvement of proteins related to DNA replication, ribosome biogenesis, protein translation, and degradation, following the more proliferative and aggressive phenotype of advanced cancer stages. Orthogonal

validation by immunohistochemistry (IHC) showed higher expression of RRS1, a ribosome biogenesis regulator, in relapse samples (cluster 2; Fig. 2c, d, $p = 0.047$). Cluster 1 included NOTCH3 and HDAC1, proteins known to mediate EWS-ETS signaling[35–37] (Fig. 2b and Supplementary Fig. S2b). Interestingly, examination of the opposite trend, of upregulated proteins upon treatment (clusters 7–8), showed a network of

**Fig. 1 | Clinical and molecular features of EWS Cohort. a** Anatomical distribution of EWS tumors in the study cohort. **b** A timeline of EWS disease stages and treatment, including surgical biopsies, pre-treatment, and post-neoadjuvant chemotherapy-treated surgical specimens. **c** Stages of post-treatment samples include local relapse samples and metastases. **d** Clinical parameters of 74 EWS patients. Clinical parameters are indicated. A good pathological response is defined as more than 90% necrosis after treatment. Missing values (NA) in 3 year survival and disease-free survival panels indicate that patients are alive but have less than 3 years of follow-up. Missing values in pathological response indicate that the patient did

not undergo surgery and therefore lacks a pathological evaluation post-treatment. HSCT, Hematopoietic stem cell transplantation. **e** The heatmap shows the intensity of six EWS-associated proteins across the entire EWS cohort ($n = 162$ tissue samples, including biological replicates). Samples are ordered by FLI1 intensity. The top color bar indicates tumor stage: Primary (light blue), Post-NACT (post-neoadjuvant chemotherapy treatment, orange), Metastasis (maroon), Relapse (purple), and Progression (green). Heatmap colors represent protein abundance ($\log_2$ intensity), ranging from blue (low) to dark purple (high); gray indicates proteins not identified. Source data are provided as a Source Data file.

immune-related proteins, such as STAT1, interferon-related proteins (IFIT1 and IFIT5), apoptotic proteins (CASP1 and PYCARD), and a group of iron-related and redox enzymes, including ferritin (FTH1 and FTL), FDXR, GPX3, and MAOB, as well as glycolytic enzymes and hypoxia-inducible factor (HIF1a) targets, including PKM, hexokinase 3, and transaldolase (Fig. 2e).

In agreement with the identified protein clusters, enrichment analysis on the ratios between the primary tumors and each one of the other stages showed downregulation of DNA replication processes, DNA damage response proteins, and ribosome biogenesis proteins upon treatment, and elevation of most of these pathways upon relapse and progression (Fig. 2f, Supplementary Fig. S2c). Mitochondrial metabolism proteins were downregulated in all stages relative to the primary tumor. Additionally, post-NACT samples presented significant enrichments of multiple immune-related pathways, including MHC class II and type I interferon (Fig. 2f, Supplementary Fig. S2c, and Source Data file).

The proteomics data showed high expression levels of iron-related proteins in post-NACT samples. We validated the high expression by IHC of FTH1 (ferritin heavy chain 1), which showed significantly higher expression in the cancer cells post-NACT (Fig. 3a, b, $p = 0.0014$). Similarly, IHC analysis showed higher expression of the transferrin receptor (TFRC) in post-NACT samples (Fig. 3c, d, $p = 0.0163$). TFRC is a key mediator of cellular iron uptake, facilitating the transport of transferrin-bound iron into the cell. FTH1 typically acts as a counter-regulatory protein by sequestering intracellular iron, limiting ROS generation and lipid peroxidation availability. Iron-related proteins and redox proteins are known to inhibit ferroptotic cell death, which was shown to be associated with response to chemotherapy. For example, chemotherapy has been shown to elevate HIF1a, which in turn upregulates GPX3, FDXR, and ferritin[38,39]. These factors might then inhibit ferroptosis and confer cell survival despite high cell stress. Additionally, previous reports suggested that iron chelation and anti-oxidant activity in tumors can increase interferon signaling and immune response[40–44], thereby connecting ferroptosis inhibition and drug resistance to tumor immunogenicity. In accordance with the hypothesized involvement of iron metabolism in chemotherapy resistance, iron staining of selected tumor samples showed a marked increase in the non-heme-bound iron in the post-treatment samples (Fig. 3e, f, $p = 0.0127$). These results testify that the ferroptosis-related signal does not merely reflect tissue hemorrhage in the post-treatment samples. Altogether, we propose that these proteins may confer treatment resistance via inhibition of chemotherapy-induced ferroptosis and induction of an immune response.

To further explore the expression of ferroptotic proteins in EWS clinical samples, we focused on known ferroptosis regulators from FerrDb[45] and leveraged the unique paired nature of some patient samples in our study to examine how these proteins change upon treatment within single patients. This analysis increased the number of proteins that were proposed to be involved in ferroptosis inhibition and induction of autophagy, including GPX4, BRD4, HSPB1, NFS1, and ATG16L1 (Supplementary Fig. S3a; $p < 0.05$). Additionally, using an independent mRNA expression dataset[46], we validated the clinical relevance of these proteins. Consistent with our clinical proteomics

data, high mRNA expression of *GPX4, FTH*, and *FTL* was associated with shorter event-free survival of EWS patients, and a similar trend was found for overall survival (Supplementary Fig. S3b, c).

Next, we wished to investigate the functional importance of ferroptosis to chemotherapy resistance. We compared the sensitivity of EWS cells to chemotherapy with and without the induction of ferroptosis. To this end, we examined the impact of two ferroptosis inducers, RSL3, a GPX4 inhibitor, and erastin, an inhibitor of the cystine/glutamate antiporter, on the response to doxorubicin and etoposide chemotherapies. We used the cell line model of A673 cells and its chemo-resistant counterpart[14], and in addition, generated a resistant cell line model from the RDES cell line.

Generation of RDES cells with reduced chemosensitivity was achieved by treating the cells for five cycles of combination treatments with four chemotherapeutic drugs, doxorubicin, etoposide, vincristine, and cyclophosphamide (see "Methods"). To evaluate chemotherapy sensitivity, we treated parental and long-term drug-exposed A673 and RDES cells with increasing concentrations of doxorubicin or etoposide in 3D spheroid models. As expected, prolonged drug exposure led to an elevation in $IC_{50}$ values for both drugs in A673 and RDES cells (Fig. 3g). In RDES spheroids, co-treatment with the ferroptosis inducers RSL3 or erastin reduced etoposide $IC_{50}$ values in both parental and adapted cells. Doxorubicin co-treatment showed that the parental cells were more sensitive to RSL3, and the adapted ones were more sensitive to erastin (Fig. 3h–k). Similar experiments in A673 cells showed a stronger signal in the adapted cells, primarily upon doxorubicin treatment (Supplementary Fig. S4a, b). These findings support the notion that ferroptosis-related mechanisms may contribute to the variable drug responses observed in patient tumors, warranting further validation in future studies.

## Proteomic investigation of prognostic proteins suggests immune involvement in patient prognosis

Routine clinical diagnosis examines primary tumors to identify the exact translocation and EWS molecular markers. Prognosis is primarily determined by the tumor's location and presence of metastasis. Pelvic tumors are known to present with a poorer prognosis than appendicular tumors in the limbs[5]. These are also often diagnosed later, as they are asymptomatic for longer periods. Tumor metastatic stage at diagnosis is the most important prognostic factor; however, even non-metastatic tumors often relapse and metastasize[5]. Harnessing the rich data from our cohort of primary tumors and the clinical data of each patient, we aimed to identify potential prognostic features in the proteomic profiles of the primary tumors. To this end, we directly compared the proteomic differences between primary tumors from patients with good and poor prognoses, based on three clinical parameters: metastasis at diagnosis, three-year disease-free survival (DFS), and tumor relapse (Fig. 4a, b; FDR < 0.1). We found 100 significantly changing proteins between primary tumors with or without metastasis at diagnosis. Nineteen proteins differed based on the relapse, and only one protein differed based on disease-free survival, probably due to the small number of patients with more than 3-years follow-up. Enrichment analysis on the difference between the means of the two prognostic groups revealed that patients with good prognosis (no

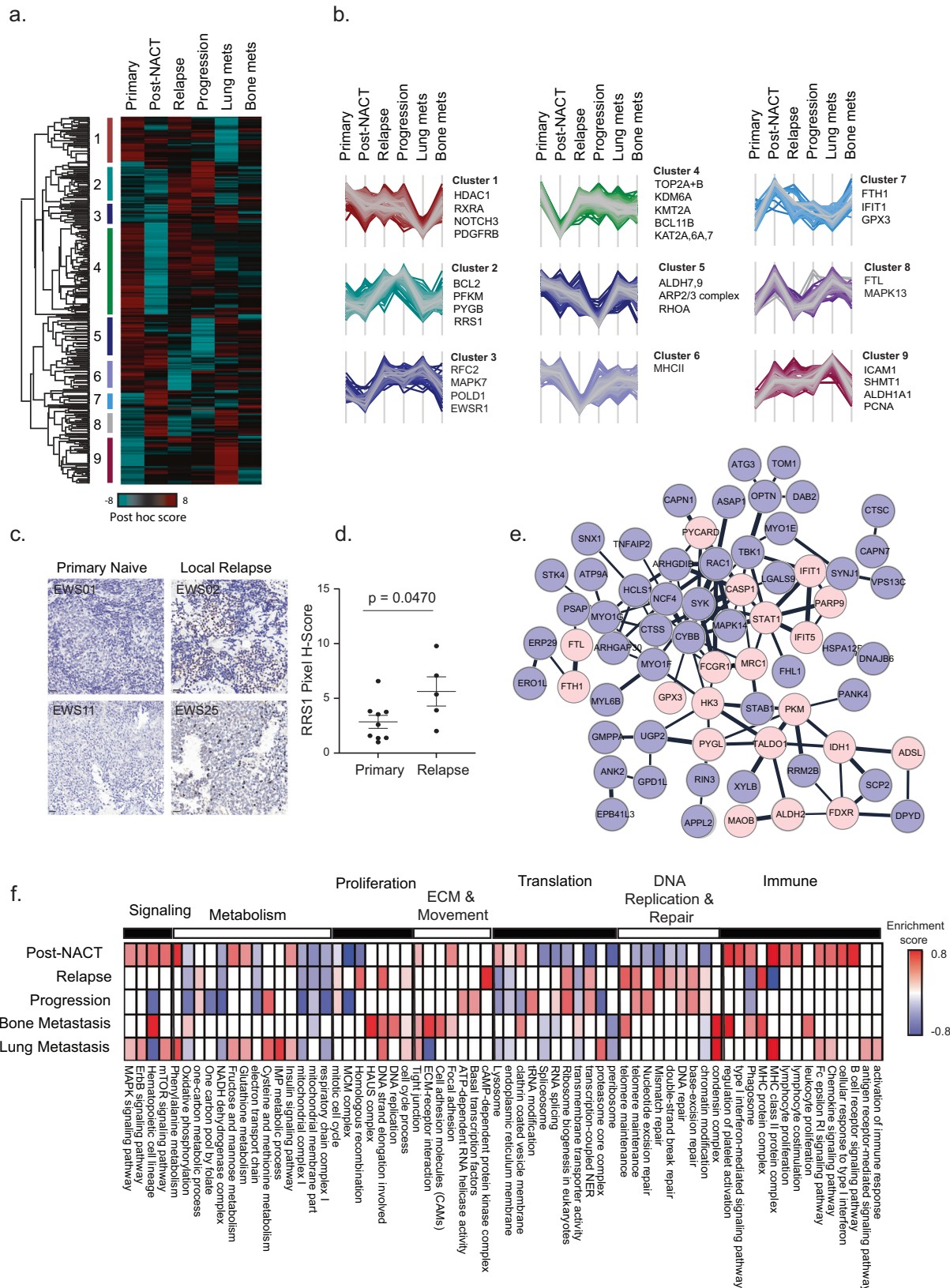

relapse, no metastasis, and more than 3 years of disease-free survival) showed enrichment of HLA proteins and type 1 interferon signaling, as well as proteasome activity. On the other hand, patients with poor prognosis exhibited enrichment of oxidative phosphorylation, cell cycle, and DNA replication (Fig. 4c–e, FDR < 0.02, Source Data file). Among the proteins that were commonly highly expressed in patients

with poor prognosis (at least 2-fold change), we found known neutrophil degranulation markers such as AZU1, MPO, PRTN3, and CAMP (Fig. 4f, g). A similar network of neutrophil proteins was also highly expressed in non-metastatic tumors that showed relapse (Supplementary Fig. 4b), raising the possibility that neutrophil abundance in primary tumors could be an early indicator of relapse risk,

**Fig. 2 | Proteomic analysis of EWS stages. a** One-way ANOVA analysis was conducted to compare between tumor stages ($n = 162$ samples; 6 groups including biological replicates from some of the patients), followed by two-sided Tukey's post-hoc testing (FDR < 0.05), and revealed 1013 proteins that changed significantly across samples (full list in Source Data). Hierarchical clustering identified nine clusters. Heatmap colors represent standardized protein expression (post-hoc score), ranging from turquoise (low) to red (high). **b** Profile plots of protein expression patterns (post-hoc score) in each cluster defined in 2a, across six cancer stages (Primary, Post-NACT, Relapse, Progression, Lung mets, Bone mets). Selected proteins of interest are indicated. **c** Representative IHC images of RRS1 in primary and local relapse samples. Scale bar, 50 μm. Images are representative of $n = 14$ biologically independent samples. **d** Quantitative analysis of RRS1 in primary tumors ($n = 9$ biologically independent samples) and local relapse samples ($n = 5$ biologically independent samples). Data are presented as mean ± SEM. Statistical analysis was performed using a two-sided Student's $t$ test ($t$ (12) = 2.21, $p = 0.047$, 95% CI [0.04299, 5.518], Cohen's $d = 1.23$). Quantification was performed on proteomics ROIs. **e** A network of proteins from clusters 7 and 8 shows high expression in post-chemotherapy samples. The network interactions were taken from the STRING database. Ferroptotic proteins (FTL, FTH1, GPX3, and MAOB), immune-related proteins (STAT1, IFIT1/5, MRC1), and metabolic proteins (HK3, PKM, IDH1) are highlighted in light pink. **f** 1D annotation enrichment analysis (FDR $q < 0.02$, two-sided Mann–Whitney test) was performed on the median fold change for each tumor stage relative to the primary tumor stage. The heatmap shows the enrichment scores for selected functional annotations, ranging from blue (low) to red (high). The complete enrichment list is provided in the Source Data. Source data are provided as a Source Data file.

independent of metastatic status. Finally, we confirmed the existence of neutrophils by immunofluorescence staining of MPO and CD15 (Fig. 4h).

## Unsupervised clustering of primary tumors reveals three patient groups associated with prognosis

Molecular classification of primary tumors can be critical for treatment decision-making and the basis for drug development. We wished to investigate whether unsupervised analysis of the proteomics data of the primary tumors can highlight protein networks that associate with the patients' clinical manifestations. Using the Consensus Clustering algorithm[47], tumors were separated into three clusters, which include 34, 7, and 17 patients (Fig. 5a, Supplementary Fig. S5a, b). These clusters reflected prognostic differences between clusters, with cluster 2 having a poorer prognosis than cluster 1 and cluster 3 (Fig. 5b). Cox regression analysis also showed a significantly higher hazard ratio of cluster 2 patients, independent of other clinical parameters, such as the tumor location and surgical intervention (Fig. 5c, $p = 0.044$). In agreement with previous reports, the metastatic stage at diagnosis also presented a significantly higher hazard ratio ($p = 0.019$). Similarly, more non-metastatic patients were found in cluster 3 and more relapsing patients were found in cluster 2 (Fisher's exact test, $p < 0.05$; Supplementary Fig. 5c, d). These results show that the unsupervised clustering of the proteomic data could reveal prognostic proteins already in the primary tumors.

Analysis of the significantly changing proteins between these clusters identified 1,263 proteins that differ between clusters (Fig. 5d, Source Data file, FDR < 0.05). A two-dimensional enrichment analysis showed that mitochondrial metabolism was lower in cluster 2 relative to clusters 1 and 3 (Fig. 5e, f, Source Data file, FDR < 0.02). MHC complex was enriched in cluster 3 relative to clusters 1 and 2, while cluster 2 was highly enriched for cell cycle proteins (e.g., MCM complex proteins) and ubiquitin-mediated proteolysis, and de-enriched for proteasome core complex (Fig. 5e, f). This conundrum prompted us to closely examine the protein expression level of the ubiquitin-proteasome system. Clustering of 72 significantly changing proteins that belong to these pathways showed separation into two main protein clusters, and almost perfect separation of samples according to the consensus clusters (Supplementary Fig. S5e). Consistent with the enrichment results, cluster 2 samples exhibited low expression of 20S core proteasome proteins and high expression of 19S proteasomal proteins. This high expression of 19S proteins coincided with increased expression of multiple deubiquitinases, notably including ten ubiquitin-specific peptidases (USPs). Moreover, multiple ubiquitin-conjugating enzymes (UBE2s) were highly expressed in the same group of samples (consensus cluster 2). UBE2A, UBE2B, UBE2D1, UBE2D1, and UBE2H were all implicated in the positive regulation of DNA-damage response proteins and are also positively correlated with these processes in our data (Supplementary Fig. S5f). In contrast to the expression of ubiquitin-proteasome related proteins in cluster 2, cluster 3 presented high expression of the 20S proteasome, and key

mediator of proteasomal degradation and antigen presentation to the immune system. We speculated that higher proteasomal degradation processes in cluster 3 may be associated with antigen presentation on MHC class I in these tumors[48,49]. Assessment of antigen processing and presentation proteins showed higher levels of HLA proteins (MHC class I and MHC class II) specifically in cluster 3 (Supplementary Fig. S5g). Overall, these results show two main phenotypes; the first, in cluster 2 samples, which suggests low overall ubiquitin-proteasome mediated degradation, and the second one, in cluster 3 samples, suggests high proteasomal activity and antigen presentation.

## Immunological characterization of EWS consensus clusters

Identification of networks of proteins associated with antigen presentation suggested higher immunogenicity of cluster 3 tumors and lower immunogenicity of cluster 2 tumors. The significant enrichment of immune processes prompted us to generate an immune score for each tumor, based on the average expression level of immune signature proteins from the ESTIMATE algorithm[31]. Interestingly, despite having significantly different prognoses, both clusters 2 and 3 presented higher immune scores than cluster 1 (Fig. 6a, $p = 0.0014$, $p = 5.4 \times 10^{-4}$, respectively). Most prominently, cluster 2, which presented a poor prognosis, and cluster 3, which presented a good prognosis, had a high involvement of immune processes (Fig. 6a). However, the proteomic data showed marked differences between the immune processes in the two clusters. While cluster 2 showed high expression of anti-inflammatory signals, primarily dominated by neutrophil and checkpoint proteins (AZU1, S100A8 and 10, CD276, etc.; Fig. 6b), the immune processes in cluster 3 included HLA proteins, implying potential activation of anti-tumor signals by the adaptive immune system. In agreement, deconvolution of the proteomic data using the xCell algorithm predicted higher numbers of neutrophils, but not M2 macrophages, in cluster 2 (Fig. 6c).

To validate the proteomic-based predictions, we profiled the main immune cell populations using multiplexed immunofluorescence imaging of neutrophils (MPO and CD15), macrophages (CD68, HLADR, and MRC1), T-cells (CD4 and CD8), and the EWS cancer cells (NKX2.2) on a subset of the primary tumors (Source Data file). In agreement with the proteomic results that showed higher MHC expression and enrichment of antigen processing and presentation in cluster 3, multiplexed imaging analysis of the proteomic ROIs showed a higher abundance of CD68+ macrophages, higher abundance of HLADR+ cells and higher abundance of CD68 + HLADR+ cells in cluster 3 relative to clusters 1 and 2 (Fig. 6d, e). In addition, we found a higher abundance of CD4 + T-cells in cluster 3 (Fig. 6f, g). Per the proteomics data, we found a higher percentage of neutrophils (CD15 + MPO+, Supplementary Fig. S6i, j) in primary tumors from cluster 2, which aligns with the poorer survival of these patients (Figs. 5b and 4h). Comparison between good and poor prognosis patients (relapse vs. non-relapse, and 3-year disease-free survival) tumors showed that elevated macrophages and HLA+ cells are associated with better prognosis (Supplementary Fig. S6). Altogether, our results show the

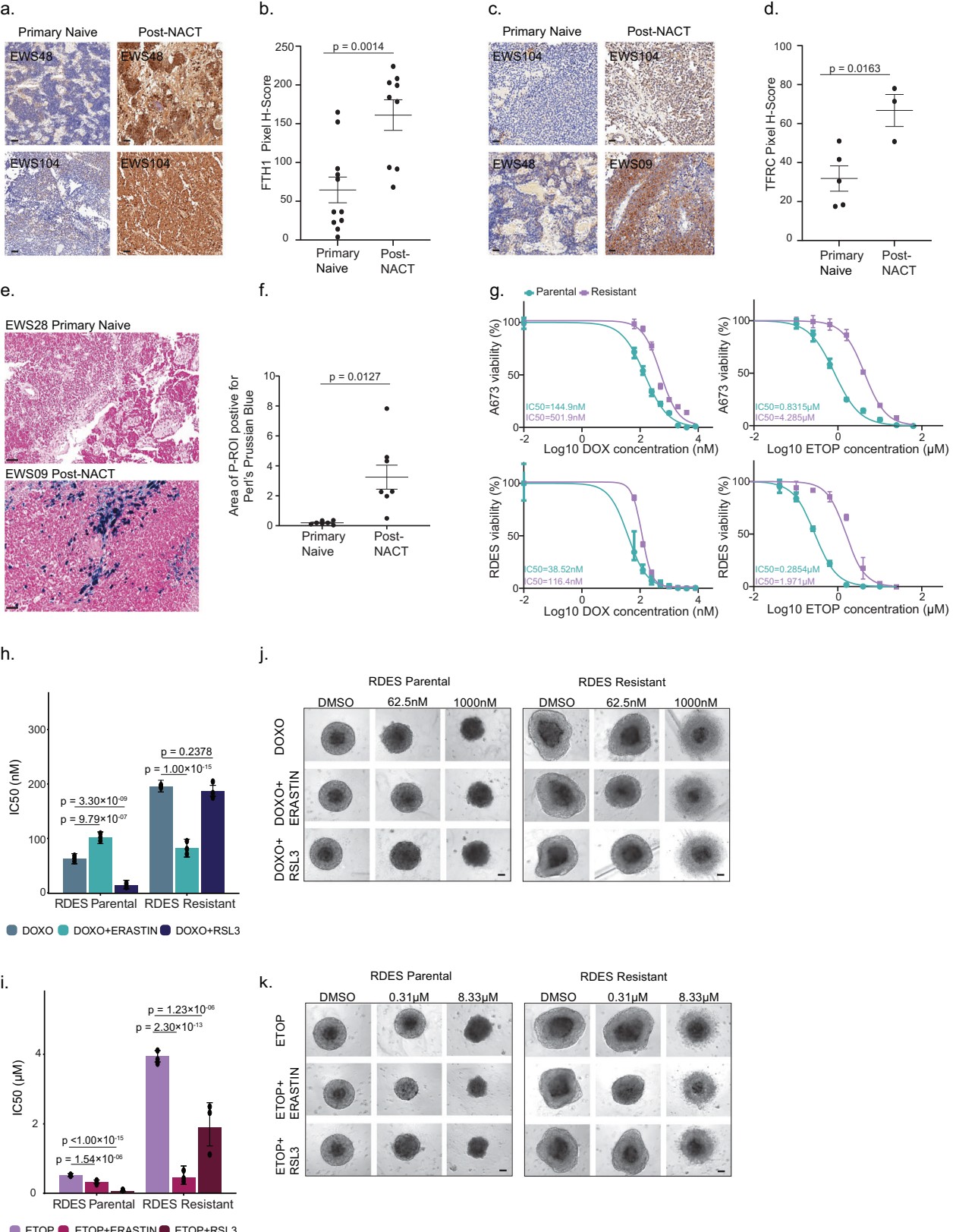

potential involvement of neutrophils in the more aggressive tumors and the potential involvement of macrophages and T-cells in less aggressive tumors. These results suggest that the immune system could be harnessed to increase anti-tumor activity in a subset of the patients, and the substantial proteomic differences between clusters suggest that protein-based biomarkers could predict immune involvement.

## Discussion

The understanding of Ewing sarcoma (EWS) has evolved from a simplistic view of single gene translocation to an intricate biology influenced by diverse cellular mechanisms driving aggressiveness and progression. Understanding the importance of the tumor microenvironment and its impact on the cancer phenotype prompted us to analyze a cohort of clinical samples that represent the inter-patient

**Fig. 3 | Ferroptosis in chemo-resistant EWS. a** IHC images of FTH1 in matched primary and post-NACT samples. Scale bar, 50 μm. Images are representative of $n = 20$ biologically independent samples, including a subset of matched samples from the same patients. **b** FTH1 quantification in primary tumors ($n = 11$ biologically independent samples) and post-NACT samples ($n = 9$ biologically independent samples), including a subset of matched samples from the same patients. Data are presented as mean ± SEM. Statistical analysis was performed using a two-sided unpaired t-test ($t (18) = 3.771$, $p = 0.0014$, 95% CI [42.78, 150.4], Cohen's $d = 1.69$). Quantification was based on proteomics ROIs. **c** IHC images of TFRC in primary and post-NACT samples. Scale bar, 50 μm. Images are representative of $n = 8$ biologically independent samples, including a subset of matched samples from the same patients. **d** TFRC quantification in primary tumors ($n = 5$ biologically independent samples) and post-NACT samples ($n = 3$ biologically independent samples), including a subset of matched samples from the same patients. Data are presented as mean ± SEM. Statistical analysis was performed using a two-sided Student's $t$ test ($t (6) = 3.3$, $p = 0.016$, 95% CI [9.055, 60.77], Cohen's $d = 2.41$). Quantification was based on the proteomic ROI. **e** Representative IHC images of Perl's Prussian blue in primary and post-NACT samples. Scale bar, 50 μm. Images are representative of $n = 14$ biologically independent samples. **f** Quantification of Perl's Prussian blue-positive area within proteomic ROIs in primary ($n = 7$ biologically independent samples) and post-NACT ($n = 7$ biologically independent samples) samples. Data are presented as mean ± SEM. Statistical analysis was performed using a two-sided Welch's t-test ($t (6.03) = 3.5$, $p = 0.012$, 95% CI [0.9627, 5.422], Glass' $\Delta = -1.324$). **g** Chemotherapy sensitivity profiling in 3D spheroid models of EWS. Dose-response curves for doxorubicin and etoposide in parental and resistant RDES and A673 spheroids. The $x$-axis shows $\log_{10}$ drug concentration; the $y$-axis shows cell viability relative to untreated controls (GraphPad Prism). Data points represent the mean ± SD of $n = 4–6$ independent spheroid replicates. Spheroid viability was quantified using CellTiter-Glo® 3D. Resistant cells refer to cells after prolonged culture with a chemotherapy combination. **h, i** Ferroptosis-based combination therapies enhance chemotherapy response. Bar plots show chemotherapy $IC_{50}$ values in RDES spheroids treated with doxorubicin (**h**) or etoposide (**i**) alone, or in combination with RSL3 or erastin (fixed concentrations: 20 nM or 0.33 μM, respectively). Data represent the best-fit $IC_{50}$ values derived from non-linear regression analysis (GraphPad Prism) of $n = 3$ independent spheroid replicates. Error bars indicate the 95% confidence interval (profile likelihood) of the fit. The sum-of-squares F test was used to determine statistical significance; $P$-values are indicated in the figure. **j, k** Representative images of RDES spheroids treated with chemotherapy alone or in combination with RSL3 or erastin. Scale bar, 200 μm. Images are representative of $n = 3$ independent spheroid replicates at each drug concentration. Source data are provided as a Source Data file.

variability, the differences among progression stages, and cancer anatomical locations. Our cohort of 74 EWS patients and 170 tumor samples, combined with the MS-based proteomic approach, revealed the EWS proteomic landscapes related to cancer prognosis, chemo-resistance, and progression. Moreover, the unbiased proteomic view established a robust foundation for future drug development and diagnostics. This work highlights multiple networks that may have potential implications on EWS progression.

While our study provides insights into Ewing sarcoma biology through high-resolution proteomics and spatial profiling, several limitations should be acknowledged. First, although the cohort is relatively large and includes primary, relapse, metastasis, and post-NACT samples, it was derived from a single institution. Broader, multi-center studies with larger and more diverse patient populations will be required to validate and generalize our findings. Second, functional validation of the implicated pathways, particularly those related to mitochondrial metabolism, proteasome activity, and DNA damage response, is needed to confirm mechanistic relevance. In addition, while our multiplexed imaging analysis provides spatial resolution of the tumor immune microenvironment, the functional assessment of immune cell states (e.g., T-cell exhaustion, macrophage polarization) was not performed. Lastly, despite the enrichment of cancer cells in the proteomic ROIs, the analyses are still considered bulk measurements and include some non-cancer cells. The use of the ESTIMATE algorithm provided a rough estimate of tumor purity. However, this was based on a partial gene-set coverage, and in general, this tool was not generated specifically for this cancer type. We therefore refer to these results as general assessments of purity, and we did not use them for downstream data correction.

Our findings suggest that inhibiting ferroptosis might contribute to EWS chemo-resistance. Recent studies highlight the susceptibility of various cancers, both hematological and solid tumors to ferroptosis induction[50–53]. This can be achieved in multiple ways, including inhibition of key antioxidant pathways (e.g., via GPX4), manipulation of iron metabolism to augment the intracellular labile iron pool, enhancement of lipid peroxidation, and disruption of cystine/glutamate antiporter activity[54]. For example, ferroptosis susceptibility and iron metabolism were proposed as potential therapeutic vulnerabilities in *MYCN*-driven neuroblastoma and osteosarcoma[35]. Studies in pancreatic and ovarian cancer demonstrated targeted ferroptosis induction through drug inhibition, and synergistic effects with combination therapies, emphasizing the promise of this strategy in maximizing

therapeutic impact[56,57]. Our results point toward ferroptosis modulation as a potential therapeutic avenue in EWS, suggesting that activation of these pathways/mechanisms may increase tumor response in aggressive tumors. Further studies are warranted to better characterize the relationship between ferroptosis modulation and chemotherapy response in EWS.

Examination of primary EWS tumors showed their separation into three clusters, based on their proteomic profiles. These clusters revealed the link between DDR, proteasomal and ubiquitin-related proteins, tumor immunogenicity, and patient prognosis. DDR proteins were found to be regulated by the EWS fusion gene[58]. However, their variable expression in tumors suggests additional regulators beyond the fusion gene itself. Similarly, studies highlighting USP6's influence on IFN response and USP19's role in EWS::FLI1 stability emphasize the significance of USPs in EWS development and the connection between the fusion gene, the proteasome, and immune response[59]. This aligns with our findings, where cluster 2, exhibiting the poorest prognosis, showed the highest expression of USP19 alongside seven other USPs. Targeting specific USPs demonstrated substantial growth inhibition in EWS cells, while treatment with VLX1570, a 19S proteasome inhibitor, decreased tumor growth in an EWS xenograft model, proposing 19S proteasome inhibitors as a promising therapeutic target for EWS[60]. Despite DDR inhibitors showing limited efficacy in EWS[61], the positive correlation between 19S proteasome proteins and DDR indicates a potential for combination therapy involving both the 19S proteasome and DDR pathways.

Our results showed a significant association between a favorable patient prognosis and the presence of macrophages and T-cells (cluster 3). These results suggest that there is still a need for a better understanding of the immune cell dynamics within EWS TME to harness the immune system for a subset of patients. In addition, we associated between poor patient prognosis and expression of neutrophil proteins. Tumor-associated neutrophils (TANs) exhibit multi-layered interactions within the tumor microenvironment (TME). Their involvement in angiogenesis, their influence on T-cells, tumor-associated macrophages (TAMs) and B cells, along with the role of neutrophil extracellular traps (NETs), highlight their complex impact on cancer progression[19,62]. These findings are consistent with our prior observations linking neutrophil activity to EWS tumor biology[63]. Nowadays, various neutrophil-targeted strategies involve enhancing neutrophil function (e.g., G-CSF, CXCR4 inhibitors), modulating their responses (e.g., SIRPα-CD47 interaction), or inhibiting pro-tumor phenotypes (e.g., selectin antagonists, SYK inhibitors), aiming to fine-tune their role in cancer therapy[64].

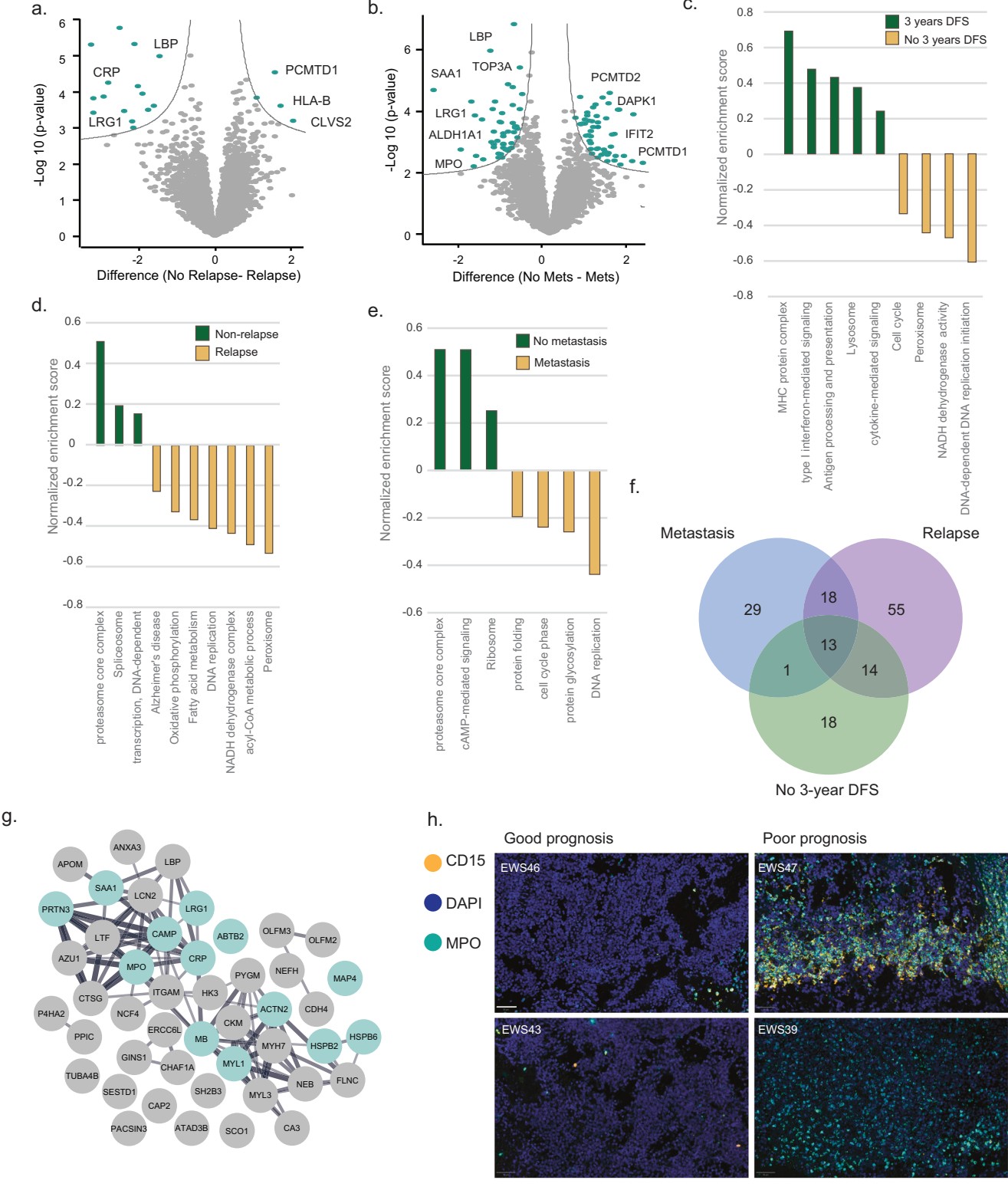

## Methods

### Ethical statement

This study complies with all relevant ethical regulations. The use of human specimens was approved by the Institutional Review Board and Ethics Committee of the Sourasky Medical Center (Approval No. 0122-11). All samples are archived samples from the Pathology Unit of the hospital. Due to the retrospective nature of the analysis utilizing archived pathological remnants, the IRB granted a waiver of informed consent. Consequently, no written consent from patients or legal guardians was required, and no compensation was provided.

### Cohort assembly

Formalin-fixed paraffin-embedded (FFPE) Ewing sarcoma tumors were collected from 74 patients. Several samples were collected from some of these patients, reaching a total of 170 EWS samples, from primary tumors, post-neoadjuvant chemotherapy treatment (NACT) samples, relapse, progression and metastases from bones and lungs. The study

**Fig. 4 | Supervised analysis of prognostic protein markers. a** Volcano plot shows significantly changing proteins between primary tumors ($n = 92$ samples, including biological replicates of some of the patients) with ($n = 17$) and without ($n = 75$) relapse (two-sided Student's $t$-test, permutation-based FDR < 0.1). **b** Volcano plot shows significantly changing proteins between primary tumors ($n = 92$ samples, including biological replicates of some of the patients) that were diagnosed with ($n = 27$) or without ($n = 65$) metastases (two-sided Student's $t$ test, permutation-based FDR < 0.1). **c** One-dimensional annotation enrichment analysis (FDR $q < 0.02$, two-sided Mann-Whitney test) performed on the median fold change between 3 years of disease-free survival (DFS) and no disease-free survival shows selected functional annotations enriched in each group. **d** One-dimension annotation enrichment analysis (FDR $q < 0.02$, two-sided Mann–Whitney test) performed on the median fold change between relapse and non-relapse shows selected functional annotations enriched in each group. **e** One-dimension annotation enrichment

analysis (FDR $q < 0.02$, two-sided Mann–Whitney test) performed on the median fold change between metastatic patients at diagnosis and non-metastatic patients, shows selected functional annotations enriched in each group. **f** A Venn diagram shows the overlap of proteins correlating with poor prognosis (relapse, metastasis at diagnosis and 3 years disease-free survival). We calculated protein expression differences between good and poor prognosis groups for each factor, selecting proteins with at least 2-fold change. **g** A protein network includes proteins identified in at least two poor prognosis factors. The network was assembled using the STRING database and subsequently visualized in Cytoscape. Proteins related to neutrophil degranulation are turquoise colored. **h** Representative immunofluorescent images of primary tumor tissue stained for MPO and CD15 neutrophils. Nuclei are stained with DAPI. Scale bar, 50 μm. Images are representative of $n = 26$ biologically independent patient-derived samples from the good and poor prognosis groups. Source data are provided as a Source Data file.

was approved by the Institutional Review Board of Sourasky Medical Center (Approval No. 0122-11). All samples were anonymized as defined in the study protocol. All patients in the cohort were under 24 years of age. EWSR1::FLI1/ERG translocation was validated by genetic testing in 75% of the tumors. For all samples, diagnosis was confirmed by a specialized sarcoma pathologist based on classical morphology, clinical context, and an immunohistochemical profile including CD99 and NKX2.2, with exclusion of alternative entities. The clinical and pathological characteristics include biological sex (determined based on clinical records), age, metastasis at diagnosis, relapse, pathological response (based on percent necrosis), translocation type, other treatment modalities, and 3-year disease-free survival and overall survival (when available). All patients underwent NACT to reduce tumor burden with the COG AEWS0031 or EURO-E.W.I.N.G. 99 protocols. To protect patient privacy, age is reported as an aggregated summary statistics (median and range) rather than as individual-level data points. A summary of these clinicopathological features at the cohort level is provided in Supplementary Data 3. All samples analyzed by proteomics are consumed for sample preparation. Remaining FFPE blocks are archived in the Sourasky Medical Center Pathology Department.

We divided our samples into different categories based on tumor stage and origin. Primary samples were treatment-naïve surgical biopsies; post-chemotherapy samples were tumors that have undergone a NAT treatment and defined as poor responders with less than 90% tumor necrosis; local relapse refers to relapse in the area surrounding the primary tumor; progression refers to a relapse stage that does not respond to treatment; metastasis includes the spread of cancer to distant sites, such as the lungs or bones. Seventeen patients in our cohort had samples from multiple disease stages (Supplementary Fig. S1a and Supplementary Data 1, 2). Tissue sections were examined by an expert pathologist specializing in bone tumors to delineate tumor areas, assess necrosis, and evaluate treatment response in post-treatment samples.

### Clinical sample preparation
Each FFPE tumor sample was deparaffinized with two xylene washes, followed by rehydration through a series of ethanol washes: twice in 100% ethanol, once in 96%, and once in 70%. The samples were then placed in distilled water (DDW) until scraping. Using hematoxylin and eosin-stained tissue samples as a template, tumors were macrodissected to contain at least 80% cancer cells. Pathological analysis of each tumor ensured that we avoid mainly fibrotic and necrotic areas, as well as regions with intense lymphocyte infiltration and adipose tissue. For each sample, we macrodissected cancer cell regions from 1 to 4 FFPE sections (~0.5–12 mm² tissue area in each section) of 6 μm thickness. To improve representativeness and minimize bias, multiple ROIs from different parts of each tumor were selected. The scraped tissues were lysed in 4% SDS in 25 mM HEPES buffer, followed by de-

crosslinking by heating for 1.5 h at 95 °C, followed by sonication for 20 cycles (30 s on, 30 s off) in a Pico-Bioruptor sonicator (Diagenode). Protein trypsin digestion was performed following the semi-automated SP3 protocol[65,66] using the Bravo liquid handler (Agilent). Proteins were reduced with 10 mM TCEP and alkylated with 40 mM CAA followed by overnight digestion at 37 °C with LysC-Trypsin mix (Promega, 1:50 enzyme: protein ratio) and sequencing grade-modified trypsin (Promega, 1:25 enzyme: protein ratio), respectively. The second day of the SP3 protocol was performed manually, and the resulting peptide concentrations were determined using fluorescamine assay by fluorescence measurement on the 3300 Nanodrop.

### LC-MS-based proteomics
LC-MS analyses were performed on the Dionex UltiMate RSLC 3000 nano-HPLC system (Thermo Fisher Scientific) coupled to the Exploris480 mass spectrometer, using gas phase fractionation and data-independent acquisition (DIA). Peptides were loaded onto a PepMap precolumn, and followed by separation on EASY-spray Pep-Map columns (75 μm i.d., 75 cm length), using a 145-min water-acetonitrile gradient at a flow rate of 200 nL/min. The gradient was composed of two steps, 12–28% solvent B (99.9% [v/v] acetonitrile, 0.1% [v/v] formic acid) over 120 min, followed by 28–44% solvent B over 25 min. MS analyses followed the DIA acquisition approach with a mass range of 400–1000 m/z and 24 isolation windows of 25 Th. MS1 resolution was 120 K, with the standard AGC target set to 300%. MS2 scans were acquired with 30 K resolution and an AGC target of 100%. Normalized HCD collision energy was 32. Gas phase fractionation was performed by high-field asymmetric waveform Ion Mobility Spectrometry (FAIMS) and we performed two injections of 0.5 μg peptides from each sample. The peptides were analyzed using two different combinations of compensation voltages (CVs), specifically −40 and −60, and −50 and −65.

### Spectral library and DIA-MS data processing
Raw MS files were divided into subfiles based on their CV values and transformed to .d files before data analysis using the DIA-NN software[67] (version 1.8). In-silico spectral library was generated in DIA-NN based on the canonical human proteome (Uniprot Release, 2018_03; 20,612 sequences) and peptide retention. We configured the following settings for database search: Trypsin/P was specified as the protease with a maximum of 1 missed cleavage. Precursor mass ranges of 300 to 1800 m/z, fragment mass ranges of 200 to 1800 m/z, with a scan window of 15, and retention of precursors with charges +1–4 and lengths between 7 and 30 amino acids. Carbamidomethylation of cysteine residues was defined as a fixed modification, and Oxidation (M) and N-terminal Acetylation were set as variable modifications. Interfering precursor peaks were eliminated, and a robust LC (high accuracy) quantification strategy was done. Precursors were filtered based on a $q$-value of 0.01. Protein grouping was performed at the

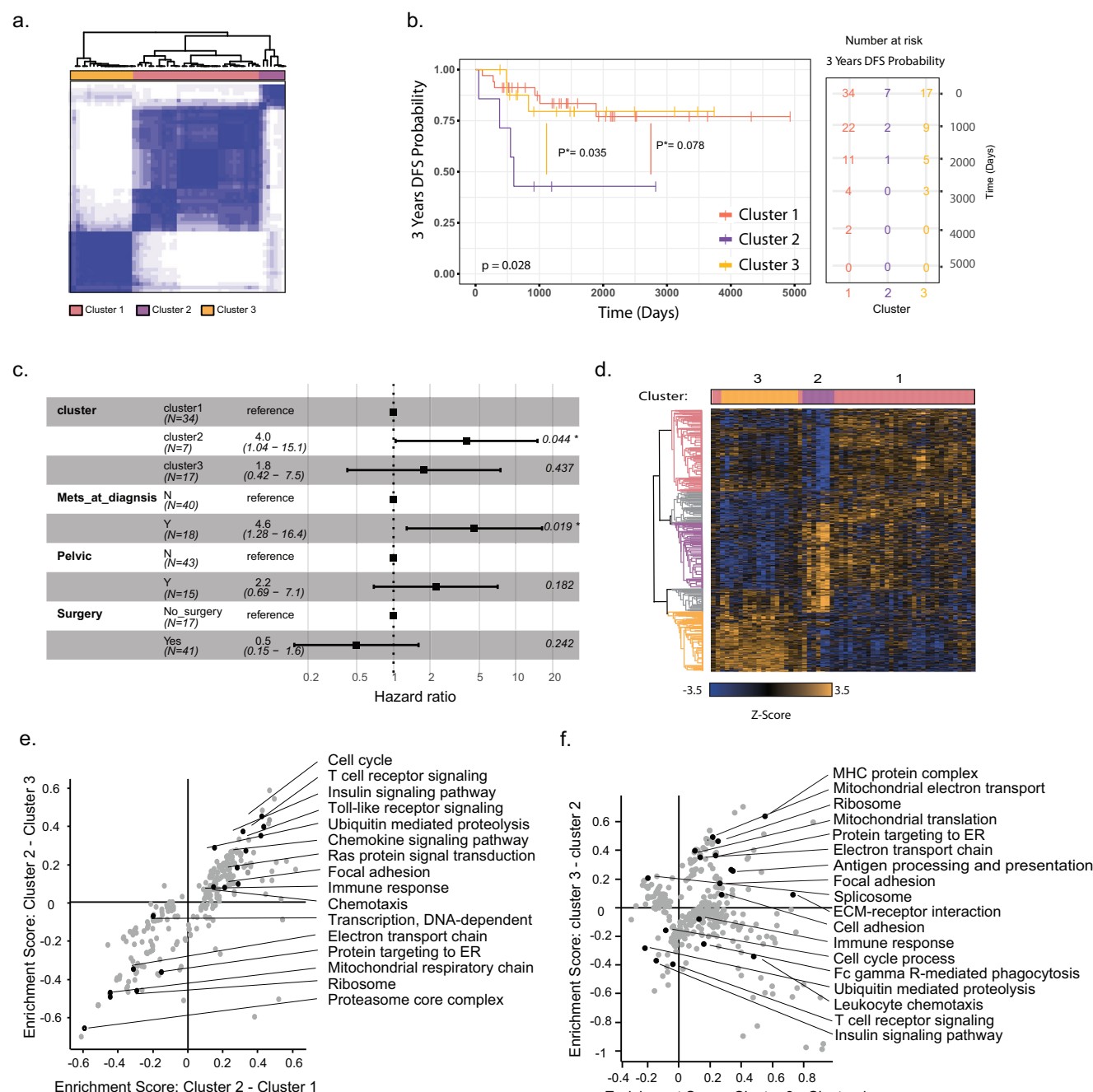

**Fig. 5 | Unsupervised clustering of primary EWS tumors. a** The consensus matrix of EWS primary samples (*n* = 58 biologically independent samples; data were averaged for patients with multiple samples) separates primary tumors into three clusters. The color scale represents the consensus index, ranging from 0 (white) to 1 (dark blue), indicating the frequency with which sample pairs cluster together. **b** Kaplan–Meier survival curve for three clusters shows significant differences in 3 years DFS (*n* = 58 patients, data were averaged for patients with multiple samples, two-sided Log-rank test, $\chi^2$ (2) = 7.2, *p* = 0.028). Cluster 2 shows the lowest survival probability. Additionally, pairwise comparisons using the two-sided Log-Rank test with BH adjustment revealed a significant difference between cluster 1 vs. cluster 2 (*p* = 0.035), while comparisons between cluster 1 vs. cluster 3 (*p* = 0.927), and cluster 2 vs. cluster 3 (*p* = 0.078) were not significant. **c** Multivariable Cox proportional hazards regression analysis including cluster subtype, metastasis at diagnosis, pelvic tumor location, and surgery status as covariates (*n* = 58 patients; data were averaged for patients with multiple samples). Forest plot shows that cluster 2 vs. cluster 1 (*HR*: 4, 95% CI = 1.04–15.1, *p* = 0.043) and metastasis at diagnosis vs. no

metastasis (*HR*: 4.6, 95% CI = 1.26–16.4, *p* = 0.019) are independently associated with poorer DFS survival. Pelvic disease and surgery status were not statistically significant. *P*-values were derived using the Wald test. **d** Hierarchical clustering of significantly changing proteins between three primary EWS clusters *n* = 58 biologically independent samples; data were averaged for patients with multiple; one-way ANOVA Benjamini–Hochberg FDR < 0.05). The color bar indicates the cluster number. Heatmap colors represent standardized protein expression (*Z*-score), ranging from blue (low) to yellow (high). **e** Two-dimensional annotation enrichment analysis shows enriched pathways in each cluster based on the median fold change between cluster 2 and clusters 1 and 3. Every dot represents a significantly enriched pathway. Significance was determined using a two-dimensional generalization of the two-sided Wilcoxon–Mann–Whitney test with Benjamini–Hochberg correction for multiple hypotheses (FDR < 0.02). **f** Same as (**e**) but for the ratios of cluster 3 and clusters 1 and 2. The complete enrichment list is provided in the Source Data. Source data are provided as a Source Data file.

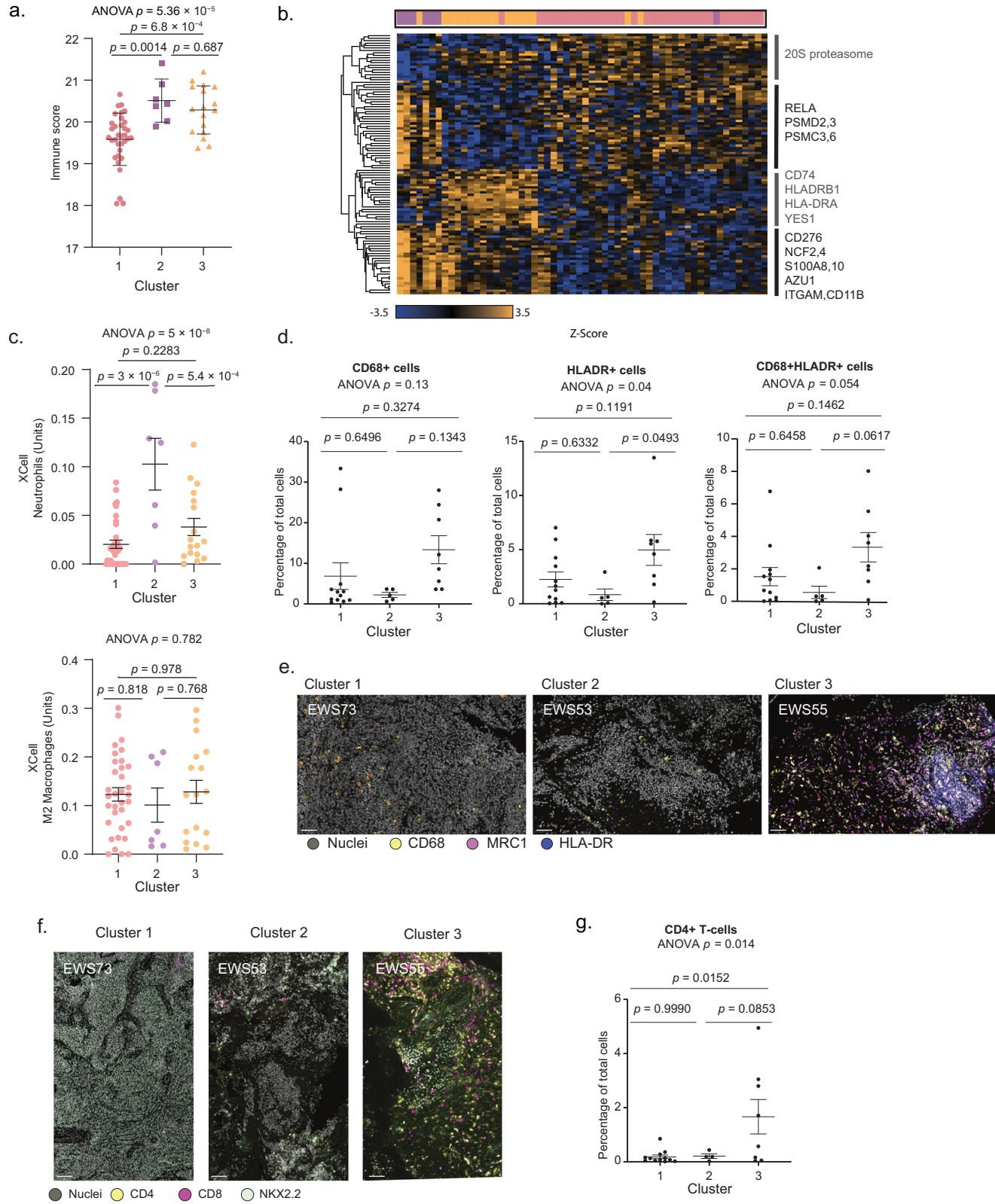

canonical gene level. The neural network classifier operated in double-pass mode to enhance classification performance. Downstream data analysis was performed on the protein group table from the DIA-NN output, which combined all CV sub-files of each sample into a single intensity value for each protein. Raw protein intensity values were log$_2$-transformed and normalized using Quantile normalization in R (R package preprocessCore v1.64.0). Four samples were excluded from further analysis: one BCOR sarcoma and three low-quality samples with fewer than 5700 identified proteins. In addition, we averaged two other samples with their technical replicates from the same FFPE slide and excluded two post-neoadjuvant chemotherapy samples that didn't receive the complete chemotherapy regimen.

**Fig. 6 | Multiplexed imaging of EWS consensus clusters. a** Immune scores in EWS consensus clusters ($n = 58$ biologically independent samples; data were averaged for patients with multiple samples). Immune scores were calculated as the median expression levels of 56 immune-related proteins comprising the ESTIMATE signature found in our dataset. Significance was calculated using one-way ANOVA ($F$ (2, 55) = 11.82, $p = 5.4 \times 10^{-5}$, $\eta^2 = 0.3$) followed by two-sided Tukey's post-hoc pairwise comparisons between clusters. Data are presented as mean ± SEM. *P*-values are indicated in the figure; full statistical details, including test statistic, degrees of freedom, CI and effect sizes are provided in the Source Data. **b** Hierarchical clustering of immune-related proteins that significantly differ between consensus clusters. Heatmap colors represent standardized protein expression (*Z*-score), ranging from blue (low) to yellow (high). **c** xCell-based estimation of Neutrophil and M2 macrophage count. Each dot represents an independent primary patient ($n = 58$ biologically independent samples; data were averaged for patients with multiple samples) Significance was calculated using one-way ANOVA ($F$ (2, 55) = 15.31, $p = 5 \times 10^{-6}$, $\eta^2 = 0.357$) and for M2 macrophages ($F$ (2, 55) = 0.247, $p = 0.782$, $\eta^2 = 0.008$), followed by two-sided Tukey's post-hoc pairwise comparisons between clusters. Data are presented as mean ± SEM. *P*-values are indicated in the figure; full statistical details, including test statistics, degrees of freedom, 95% CI, and effect sizes, are provided in the Source Data. **d** The proportion of CD68+, HLA-DR + CD68+, and HLA-DR+ cells in three clusters were calculated as a percentage of the total cell population in the proteomic ROIs. Data are shown as mean ± SEM ($n = 25$ biologically independent samples: cluster 1, $n = 12$; cluster 2, $n = 5$; cluster 3, $n = 8$). Statistical analysis was performed using one-way ANOVA to assess overall differences for CD68+ cells ($F$ (2, 22) = 2.175, $p = 0.1375$, $\eta^2 = 0.165$), for HLA-DR+ cells ($F$ (2, 22) = 3.654, $p = 0.0426$, $\eta^2 = 0.25$), and for HLA-DR + CD68+ cells ($F$ (2, 22) = 3.320, $p = 0.0549$, $\eta^2 = 0.23$), followed by two-sided Tukey's post-hoc pairwise comparisons between clusters. *P*-values are indicated in the figure; full statistical details, including t-statistics, degrees of freedom, 95% CI, and effect sizes, are provided in the Source Data. **e** Representative immunofluorescent images of CD68+, HLA-DR+, and MRC1+ cells across the three EWS consensus clusters. Scale bar, 50 µm. Images are representative of $n = 25$ biologically independent patient-derived samples. **f** Representative immunofluorescent images of CD8+ and CD4 + T-cells, and NKX2.2 EWS cells. Scale bar, 50 µm. Images are representative of $n = 24$ biologically independent samples. **g** The proportion of CD4 + T-cells in three clusters were calculated as a percentage of the total cell population in the proteomic ROIs. Data are shown as mean ± SEM ($n = 24$ biologically independent patient-derived samples: cluster 1, $n = 12$; cluster 2, $n = 4$; cluster 3, $n = 8$). Statistical analysis was performed using one-way ANOVA to assess overall differences ($F$ (2, 21) = 5.21, $p = 0.0145$, $\eta^2 = 0.33$), followed by Tukey's post-hoc pairwise comparisons between clusters. *P*-values are indicated in the figure. Source data are provided as a Source Data file.

## Data preprocessing and statistical analysis

Data analysis was performed using Perseus[68,69] software (versions 1.6.13.0 and 1.6.10.43), the R environment, and GraphPad Prism. The initial protein table included a total of 10,220 proteins. For downstream analyses, we applied a focused filtering step, excluding proteins annotated as blood-derived or extracellular matrix (ECM)-associated, to emphasize tumor cell-intrinsic proteomic features. Removal of ECM proteins was performed to minimize uncertainty arising from the mixed cellular origin of ECM proteins in bulk tissue proteomics, as their contribution could reflect either tumor or stromal compartments. This filtration does not suggest that these proteins are irrelevant to this tumor type. To confirm that ECM removal did not bias our overall conclusions, we compared principal component analysis (PCA) distributions of the dataset before and after filtering. We observed no significant changes in global sample distribution (Supplementary Fig. S7a, b). The complete, unfiltered protein dataset, including all quantified proteins and ECM proteins, is available in Supplementary Data 4 to support future studies addressing Ewing Sarcoma and its tumor microenvironment, including ECM. Tumor purity was assessed using the ESTIMATE (R package "estimate" v1.0.13) algorithm to evaluate whether bulk proteomic samples were suitable for downstream analyses. Given the partial gene-set coverage in our dataset and the specificity of stromal components in distinct tumor types, these scores should be regarded as approximate indicators of purity rather than exact quantifications of tumor, stroma, or immune composition, and they were not integrated into our downstream analyses. For the progression analysis of 162 samples (including primary, post-chemotherapy, relapse, progression, and metastasis samples), we performed additional filtration to ensure valid values in at least 70% of all samples. Missing values were imputed by generating a normal distribution with a width of 0.5 of the original intensity distribution and a downshift of 1.8 standard deviations. This resulted in a dataset of 6312 proteins, which we used to perform downstream analyses. ANOVA test to compare between stages was performed with permutation-based FDR correction of 0.05, followed by a two-sided Tukey's post hoc test.

In the primary tumor consensus clustering, we averaged biological replicates from each tumor, resulting in a dataset of 58 samples. Subsequently, we filtered these data to retain proteins with at least 70% valid values, which resulted in a dataset of 6517 proteins. Missing values were imputed as mentioned above. Student's *t* tests were used for comparisons between groups of poor and good prognosis patients, with permutation-based FDR < 0.1 and S0 = 0.1. One-dimensional (1D) annotation enrichment analyses, two-dimensional (2D) annotation enrichment analyses (based on the Wilcoxon–Mann–Whitney test), and Fisher's exact test enrichment analyses were performed using the Perseus software, with a Benjamini–Hochberg FDR < 0.02. Protein networks were generated with the STRING database[70,71] and then edited in Cytoscape[72]. Biological annotations, including Gene Ontology (GO) and Kyoto Encyclopedia of Genes and Genomes (KEGG), were taken from UniProt.

For the unsupervised clustering of the primary EWS tumors, we applied the Consensus Clustering algorithm using the R package ConsensusClusterPlus (v1.66.0)[47,73]. Parameters used were: maximal number of clusters, six; number of iterations, 1000; subsampling fraction, 0.9; clustering algorithm, hierarchical; and distance matrix, Pearson correlation. Before applying the algorithm, ratios across proteins and samples were *Z*-score normalized. Kaplan–Meier analysis and Cox regression were performed using R's "survival" (v3.8-3), "ggsurvfit" and "survminer" (v0.5.1) packages to compare consensus clusters based on 3-year disease-free survival. The Cox regression model was further adjusted for additional variables, including metastasis at diagnosis, pelvic disease, and surgery following neoadjuvant chemotherapy. The proportional hazards assumption was verified using Schoenfeld residuals, and no significant violations were observed (Global test $p = 0.20$).

Kaplan–Meier plots based on mRNA expression data were based on the Dirksen database[46] and used the R2 Genomics Analysis and Visualization platform (http://r2.amc.nl). The Kaplan Scan feature on the R2 platform was used to create Kaplan–Meier curves based on the log-rank test.

## Cell culture

A673 parental and chemotherapy-resistant cell lines (A-673; human Ewing sarcoma; kindly provided by Jacob Scott's Laboratory, Cleveland Clinic) were cultured in DMEM high-glucose (Biological Industries, #010521A) supplemented with 10% FBS, 1 mM sodium pyruvate, and 1 mM penicillin-streptomycin at 37 °C in 5% $CO_2$. RDES parental (RD-ES; human Ewing sarcoma of bone; kindly provided by Yosef Yarden's Laboratory, Weizmann Institute of Science) and resistant clones were maintained in RPMI (GIBCO, #21875034) containing 15% FBS, and 1 mM penicillin–streptomycin under identical conditions. Cultures were routinely confirmed as mycoplasma-free using a PCR-based detection kit (EZ PCR Mycoplasma Test Kit, Sartorius #20-700-20). Internal mass spectrometry-based proteomic profiling demonstrated high abundance of FLI1- and EWSR1-derived peptides, consistent with the expected Ewing sarcoma proteomic signature.

### Generation of RDES chemo resistant cell lines

To model the clinical standard of care for Ewing sarcoma, RDES cells were subjected to five cycles of alternating vincristine-doxorubicin-cyclophosphamide (VDC) and etoposide-cyclophosphamide (EC), recapitulating the VDC/IE protocol. Since ifosfamide in the IE regimen requires hepatic activation and no metabolically active form is commercially available, it was substituted with 4-hydroperoxycyclophosphamide, a chemically analogous compound, as previously done[14].

Experimental groups received either VDC-vincristine (#2068-78-2, Cayman Chemical), doxorubicin (#225316-40-9, Cayman Chemical), and 4-hydroperoxycyclophosphamide (CAS: H714675, TRC Canada)-or EC-etoposide (#33419-42-0, Cayman Chemical) and 4-hydroperoxycyclophosphamide at concentrations approximating each drug's $IC_{30}$–$IC_{40}$. Each treatment cycle consisted of 4 days of drug exposure followed by recovery in maintenance medium, with recovery intervals progressively reduced from 10 to 5 days across cycles. Controls were exposed to DMSO vehicle only throughout the experiment. The resistant cell lines generated in this study are available from the corresponding author upon request.

### 3D spheroid generation and drug sensitivity assays

Spheroid experiments were performed following published protocols[74,75] with some modifications, as detailed below. Cultures were generated in HPLM medium (Rhenium, #A4899101) supplemented with FBS (10% for A673, 15% for RDES) and 1 mM penicillin at 37 °C in a humidified 5% $CO_2$ incubator. Prior to spheroid generation, A673 and RDES parental and resistant cells were expanded as 2D monolayers for 2 days, not exceeding 70% confluence. Spheroids were generated in 96-well round-bottom, ultra-low attachment plates (Corning/Sigma-Aldrich, CLS7007-24EA) by seeding 1500 A673 or 2000 RDES cells per well in HPLM. After 24 h of aggregation, spheroids were supplemented with 10 μL fresh medium and cultured for an additional 24 h before treatment initiation.

To determine single-agent $IC_{50}$ values, spheroids were exposed to doxorubicin, etoposide, RSL3, or erastin across an 8-point concentration series, with DMSO as a constant vehicle control. Treatments were applied for additional 60 h.

Drug interaction studies included vehicle, chemotherapy alone (doxorubicin or etoposide) and in combination with the ferroptosis inducers (RSL3 or erastin). Treatments were applied for 60 h on preformed spheroids under identical conditions. All experiments were performed in three independent biological replicates.

Spheroid viability was quantified using the CellTiter-Glo® 3D assay (Promega, #G6982) according to manufacturer instructions. Luminescence signals were background-subtracted and normalized to vehicle controls to calculate relative viability. Dose-response curves and $IC_{50}$ values were derived in GraphPad Prism using nonlinear regression (log[inhibitor] vs. normalized response). Significant shifts in $IC_{50}$ between parental and resistant lines were assessed by an extra-sum-of-squares F-test.

### Immunohistochemistry staining and multiplexed immuno-fluorescence staining and imaging

Opal multiplex staining was performed using the BOND RXm research detection platform (Leica, DS9455). Tissue sections of 28 patients were deparaffinized and rehydrated using BOND dewaxing solution (Supplementary Data 5). Endogenous peroxide blocking was done with 3% H2O2 and 1% HCL in Methanol, followed by heat-induced antigen retrieval in 10 mM citric acid (PH = 6). Blocking was performed with 20% NHS and 0.1–0.5% triton according to target. Primary antibodies were used as described below. In case a secondary biotinylated antibody was used, an additional step of the biotin blocking kit was performed. Primary antibodies, diluted in 2% normal horse serum (NHS) and 0.1%-0.5% triton, were incubated overnight as detailed below.

Secondary HRP or biotin-conjugated antibodies incubation was followed by fluorescently labeled OPAL reagents or streptavidin. NKX2.2 staining was double amplified by using biotinylated secondary antibody, followed by ABC kit and OPAL reagent. Antibody removal was performed by 10-minute microwave treatment with citric acid (PH = 6), and then the protocol was repeated from the blocking step. Nuclei were stained with Hoechst. Imaging was performed by whole slide scanning using the Phenoimager system (Akoya Biosciences Fusion 2.0) with 10X/0.8 objective−pixel size of 0.5 micron and compatible Opal dedicated filter set and Auto Fluorescent background channel. Final image output of 8 bit. Three staining panels were used for immunofluorescence: (i) Panel I with anti-MPO (PA516672, Thermo Fisher Scientific, 1:100), anti-NKX2.2 (BSB3110, Bio SB, 1:75), anti-CD15 (MAS-11789, Invitrogen, 1:50), anti-CD68 (CST-76437S, Cell Signaling Technology, 1:200); (ii) Panel II with anti-NKX2.2 (as above), anti-CD4 (AB-ab133616-100, Abcam, 1:100), anti-CD8 (MA514548, Thermo Fisher Scientific, 1:100); (iii) Panel 3 with anti-NKX2.2 (as above), anti-CD68 (as above), anti-HLAII (14-9956-82, Thermo Fisher Scientific, 1:75), anti-MRC1 (HPA004114, Atlas Antibodies, 1:200), anti-PSMA3 (HPA000905, Atlas Antibodies, 1:100).

DAB staining was performed using the BOND RXm polymer refine detection system (Leica, DS9800) and standard protocol. Primary antibodies were used as follows: anti-RRS1 (HPA060937, Atlas Antibodies, 1:1000), anti-NK2.2 (as above), anti-TFRC (136800, Thermo Fisher Scientific, 1:1200), anti-FTH1 (PA5257500, Thermo Fisher Scientific, 1:2000). Imaging was performed by whole slide scanning using Pannoramic scan II (3DHistech) with a 20× objective.

Perls' Prussian Blue staining was performed to detect ferric iron deposits in tissue sections. Briefly, slides were deparaffinized, rehydrated, and incubated in freshly prepared staining solution consisting of a 1:1 mixture of 2% (w/v) potassium ferrocyanide and 2% (v/v) hydrochloric acid for 30 min at room temperature. Following incubation, sections were rinsed in distilled water, counterstained with nuclear fast red, dehydrated through graded alcohol, cleared in xylene, and mounted with coverslips. Stained slides were scanned using Pannoramic scan II (3DHistech) with a 20× objective.

### Image analysis

Spectral unmixing and background subtraction of multiplexed QPTIFF images were performed using unstained reference slides in InFormV2.0.2, Akoya Biosciences. Tiled images were stitched in Qupath[76] v0.4.3. Nuclear segmentation was performed using the Nuclei model of Cellpose 2.0[77] followed by expansion (estimated diameter of 14 pixels, expansion of up to 5 pixels) as implemented in the Qupath Cellpose extension v0.9.6. A Random Trees machine learning object classifier was trained in Qupath v0.5.1 on a training panel of representative stitched regions of 500 × 500 μm from multiple slides. Proteomic regions of interest (P-ROIs) were manually annotated on the Opal slides with guidance from consecutive H&E sections. All downstream analysis were performed with percentage of cells expressing the marker in these P-ROIs. Statistical analyses were performed using GraphPad Prism. Differences among the three proteomic clusters were evaluated by one-way ANOVA with Tukey's post hoc test for multiple comparisons. Clinical associations, including metastasis at diagnosis, relapse status, and 3-year disease-free survival (3Y DFS), were assessed using Welch's t-test. Data are reported as mean ± SEM.

Quantitative analysis of DAB staining was performed using the pixelwise H-score method[78] in QuPath v0.5.1. Proteomic regions of interest (ROIs) were manually annotated on the DAB slides as described earlier. Quantification was performed with thresholds for 1+, 2+, and 3+ staining intensities set at 0.15, 0.5, and 1.0, respectively. Comparisons between primary and relapse samples, as well as between primary and post-NACT samples, were performed in GraphPad Prism using unpaired t-tests.

## General study design

All in vitro experiments were conducted in at least 3 independent biological replicates ($n = 3$) to ensure reproducibility. Technical stability in proteomic analysis was maintained through routine performance checks using standard HeLa reference samples at regular intervals. Sample acquisition order on the mass spectrometer was randomized. Investigators were not blinded to group allocation during clinical sample selection. However, proteomic data acquisition was automated, and subsequent clustering analyses were performed using unsupervised algorithms to ensure unbiased interpretation of the data.

## Reporting summary

Further information on research design is available in the Nature Portfolio Reporting Summary linked to this article.

## Data availability

The raw mass spectrometry data generated in this study have been deposited in the PRIDE ProteomExchange database under accession code PXD050234. The processed proteomics data generated in this study are provided in the Supplementary data. The Image data used in this study are available in the BioImage Archive database under accession code S-BIAD1597. Source data are provided with this paper.

## Code availability

The code for image processing and the Github MIT license document are available via the following GitHub[79]: [https://github.com/WIS-MICC-CellObservatory/Ewing-Sarcoma-Proteomics-and-Immune-Landscape].

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

## Acknowledgements

We thank the members of the Geiger lab for fruitful discussions and technical assistance. Opal slide imaging was made possible thanks to the support of the de Picciotto Cancer Cell Observatory in Memory of Wolfgang and Ruth Lesser, a research grant from the Quinquin Foundation, and a research grant from the Morris Kahn Institute for Human Immunology and the Fabrikant-Morse Families Research Fund for Humanity. This work was funded by the Israel Science Foundation grant #3106/21 and the European Council ERC-consolidator grant #101044574. This research was also generously supported by the

Applebaum Foundation and the Vera and John Schwartz Family Center for Metabolic Biology.

## Author contributions

S.G. performed the experiments, analyzed the data and drafted the manuscript; R.S., S.G., R.E., and T.G. conceptualized, designed the research and interpreted the results; S.G., R.S., O.S., and M.M. assembled samples and clinical information; V.M., S.G., R.O., and L.F.A. performed the multiplexed imaging; V.M., O.G., and Y.A. acquired the multiplexed imaging data, developed the analytical pipelines and performed the image analyses; V.M., S.G., and S.M. performed the spheroid and ferroptosis analysis; L.B. established the proteomic sample preparation methods; V.M. contributed to figure preparation and the writing of the manuscript; B.D. initialized the research; T.G. supervised the research, analyzed the data and wrote the manuscript.

## Competing interests

The authors declare no competing interests.
