## [Peer Review File · Nature Communications]

Proteomic landscape of Ewing sarcoma primary tumors and metastases

Corresponding Author: Professor Tamar Geiger

Version 0:

Reviewer comments:

Reviewer #1

(Remarks to the Author)

In this study, the authors undertake mass spectrometry profiling of a large cohort of EWS samples. They perform descriptive analyses of the proteomic data and identify some interesting findings. The strength of the study is the large number of cases analysed for this rare entity. The weaknesses are detailed in comments below.

Major comments

1. Sample and preparation details are lacking

A. Bone sarcoma samples are often decalcified, a process which is known to impact the quality/quantity of biomolecules. Can you authors confirm if the samples were decalcified? If so, can the authors need to provide data detailing the impact of protein yields, protein IDs and protein quantification of decalcification compared to frozen or non-decalcified matched specimens in a subset of the cohort. Without this information it is difficult to assess how much of the data reflect artefacts introduced by the decalcification process. And therefore, how much of the generated proteomic data is actually representative of the disease.

B. In the methods, the authors state that samples were dewaxed based on "standard protocol" - what is this standard protocol please provide details. Also it seems there is no reversal of crosslinks step, is this the case?

C. Authors state that samples were macrodissected for 80% tumour cell content? Was this done for the tiny biopsies? Is it possible to macrodissect from tiny biopsies? Does the 1-4 FFPE sections of 6um thickness apply to biopsies as well?

2. In general for every single section, there is a lot of descriptive analysis comparing different groups or clusters. For each of these sections and comparisons, there is a need to undertake orthogonal validation of the findings by immunohistochemistry. This includes Figure 2 different timepoints, in particular pre and post treatment, primary vs mets, Figure 3, different proteins expressed in different consensus clusters, Figure 4 different proteins associated with clinical parameters. Without IHC validation, it is not possible to ascertain if the proteomics findings are not just artefacts, particularly in the context of decalcified tissue.

3. Figure 1d, authors state that these proteins and markers are highly expressed. Highly expressed is a relative term. Compared to what? This should be compared to normal bone rather than just within an EWS cohort to demonstrate that the markers are highly expressed?

4. The ESTIMATE algorithm was used several times in the manuscript, once to estimate stroma content and the other for immune cells. Can the author comment on the coverage of their protein data and the markers used by ESTIMATE. if the overlap is low, is it appropriate to use ESTIMATE for the proteomics data?

5. Comparisons between primary and post-chemo surgical specimens. Because there are no surgery alone specimens (without chemo treatment), the authors cannot exclude the possibility that some of the protein changes seen are the result of surgery rather than chemotherapy. It is known that surgery leads to the activation of inflammation and immune related pathways some of which are similar to those seen in the authors dataset. The authors should consider this possibility.

6. The authors make multiple claims which is not supported by the data, particularly lack of functional data to establish causation. These statements are pure conjecture so please remove them.

"Altogether, we propose that these proteins may confer treatment resistance via inhibition of chemotherapy-induced ferroptosis, and induction of immune response."

"Our findings suggest that inhibiting ferroptosis contributes to EWS chemo-resistance, and suggests the potential impact of the proteasome and ubiquitin proteolysis on antigen presentation and DNA damage response (DDR)."

7. All survival analyses, such as figure 3 and the analysis of the RNA data needs to be subjected to multivariable analysis which includes known prognostic/confounding factors to assess if the consensus clusters or proteins are independent prognostic variables.

8. It is surprising that there is no discussion of the limitations of this study. This is a very descriptive, hypothesis generating study with no validation of findings in an independent cohort. Single institution studies are subject to selection bias. This is just one of several limitations that need to be considered in the discussion.

9. Can the authors make clear how missing data was dealt with in the dataset? they report >10K proteins but each condition has ~7K, so how was missing data handled?

Reviewer #2

(Remarks to the Author)

The manuscript by Gordon et al. entitled 'Proteomic landscapes of Ewing sarcoma unravel immunological regulation of tumor progression' reports on a relatively large series of Ewing sarcoma (EwS) tumor samples that have been subjected to mass-spectrometry based proteomics analyses. The authors conduct a variety of subsequent bioinformatics analyses and compared (mainly unmatched) samples from treatment-naïve primary tumors and samples from metastatic lesions or relapses that are mainly post-treatment samples. These comparative and purely correlative analyses led the authors to conclude that ferroptosis might play a role in chemo-resistance. Further, they identified by various clustering analyses subgroups of primary tumors that may be associated with diverse signaling pathways and immune profiles. Some validation studies using various immunohistological markers suggested associations with metastasis at diagnosis, which led the authors to conclude that infiltration of the tumors by neutrophils may be associated with a worse patient outcome. While this study reports on a large series of samples with proteomic data, the study remains purely descriptive in nature and most conclusions are speculative and not underpinned by any functional experiment. Some of the conclusions appear overstated and the fact that some major prognostic subgroups could be also found in a published cohort based on bulk-level mRNA expression data somewhat undermines one of the major arguments of the authors to claim novelty and importance of proteomics for advancement of EwS research.

Apart from these general concerns, there are many major concerns that limit the enthusiasm of this Reviewer for this study:

1) It is simply unacceptable that the supplementary tables contain patient-specific identifying criteria, such as the precise date of diagnosis, date of birth, date of death etc. These data, if published in that form would make it very easy to unambiguously identify patients, which is not in the interest of our patients. It is hard to believe that the authors have an IRB approval for this and this Reviewer strongly recommends to de-identify the patients and remove such (unnecessary) data from the tables. Then, the statement in the methods part of "all samples were anonymized as defined in the study protocol" would be valid. The patient-specific data given in the current table is the more so critical since Israel is a relatively small country (population-wise) and since EwS is a very rare disease, patient-specific information regarding date of birth, date of death, and Israel as a likely country of residence and the diagnosis of EwS, each and every patient can be easily identified (not only by close relatives). It is the first time this Reviewer sees such a data leak in a submitted paper, and frankly speaking, this is NOT ok.

2) For almost all correlative associations of proteomics subgroups with patient parameters and other phenomenological findings it is unclear whether these events are the drivers or passengers of the observed phenotypes. For instance, the presence of neutrophils could be a mere expression of more necrotic tumor tissue within a given tumor due to higher proliferation rates that overwhelm the tumor's capacity to form new vessels, which may lead then to higher rates of metastasis. As such, neutrophils are just an epiphenomenon of biological mechanisms that are as yet not defined or discovered. Also, in this instance, multiple comparisons of immune markers with phenotypes have been done. If corrected for multiple testing, the above-mentioned association is likely rendered non-significant. Hence, subsequent conclusions are not underpinned by solid data.

3) Similarly, many downstream pathway analyses yield only generic pathways potentially associated with metastasis. For instance, what is the exact meaning of a downregulation of 'DNA-dependent DNA replication initiation' in association with disease-free survival? What does that mean? How is this validated? And what do we truly learn from these associations? Are they driving the phenotype (DFS) or just epiphenomena?

3) The identification of 13 proteins in overlap of "metastasis", "relapse", and poor "DFS" would have been a good start for follow-up, but again no meaning can be drawn from this (Fig. 4h).

4) The association of immune scores with different proteomics subclusters of which one (cluster 2) is associated with a trend for worse outcome is interesting. Yet, as the result is not nominally significant, it remains unclear whether this is a trustworthy

result and whether all subsequent explorations/speculations are worth being discussed in the current extension.

5) As shown in Figures 1a/b, the tumor samples came from mostly unmatched primaries/metastases of different individuals with very different localizations. Some sites were only represented by a very few cases (such as pelvis and bone mets). Hence, some of the findings reported in this paper could be still subject of changes once larger sample series would be available. It is also unclear to what extent the findings on immune infiltrates etc. are a mere reflection of the anatomical site and/or location of the tumor (soft-tissue versus bone).

6) The statement that ferroptosis inhibition may be an event to overcome chemo-resistance (see abstract) appears as a strong overstatement and is solely based on a correlative observation that proteins involved in ferroptosis show high expression in post-chemotherapy samples. For these samples, it should be noted that the mere finding that the samples are from a post-chemotherapy condition is insufficient to conclude that these samples were chemo-resistant. In many patients, chemotherapy needs to be interrupted (e.g. due to cumulative toxicity) and is not per se a reflection of resistance. Also, all chemotherapies applied in EwS lead to cell death in a non-ferroptosis-dependent pathway. Hence, the upregulation of ferroptosis-related genes is not per se a reflection of any biology related to relevant cell death. Again, functional analyses are completely lacking.

Reviewer #3

(Remarks to the Author)

This is a very exciting study looking at the proteome of over 100 Ewing sarcoma tumor samples from 72 patients. To my knowledge this is the first study doing MassSpec-based analysis on Ewing sarcoma tumor tissues and hence it deserves attention. Ewing sarcoma although is a disease with high clinical heterogeneity, Ewing sarcoma tumors are genetically homogeneous. Transcriptome/epigenome studies have not yet revealed strong biomarkers that could explain clinical differences. Examining the proteome is imperative.

While I am enthusiastic about this study, I do have a few concerns outlined below (not in a particular order).

1. The treatment protocol of Ewing sarcoma is not accurately presented. Patients undergo neo-adjuvant chemotherapy, followed by surgery (in most cases) and/or radiotherapy (in some cases), which is followed by maintenance chemotherapy. As such Figure 1b is not accurate (therapy cycles indicated with a radiotherapy icon is wrong). Metastasis also occurs at initial diagnosis. Relapse and metastasis are not independent events (relapse can be localized and metastatic). Figures 1a,b at their current version they do not add anything to the story and could be omitted as well. If kept, they should be designed with higher precision and with some informative content.
Calling surgery samples post-chemotherapy (and in some cases post-treatment) samples is misleading, given that most of these patients continued to receive chemo after surgery.
2. Surgery samples are known to be highly necrotic. Did the authors manage to dissect live cancer cells from these samples? How do they check for live cancer cells? Is the ESTIMATE algorithm run on dissected samples (MassSpec/protein level data)? It may be important to specifically indicate the purity of surgery samples on Supplementary Figure 1e.
3. Lines 96-98: it is not a surprise that surgery samples show the lowest correlation with other time points given that these samples are highly necrotic. Could it be that most cells they profile are non-cancer cells (i.e. hematopoietic)?
4. According to supplemental table 1b all surgery samples come from bad responders with 40% to 90% necrosis. It may be good that this point is discussed in the manuscript.
5. EWSR1 protein is expressed in most cell types. Perhaps not a good marker for Ewing sarcoma. Can the fusion oncogene be detected with MassSpec?
6. Similarly to the above comment, FLI1 is not expressed in Ewing sarcoma cancer cells. Does the FLI1 protein detected correspond to the fusion protein? Could it be that FLI1 protein detection is due to non-cancer cells in the tumor tissue (i.e. hematopoietic and/or endothelial cells)?
7. Figure 1c. Age plot – dots do not need to be connected.
8. Figure 1d and Supplementary Figure 1f (Lines 106-111). Could it be that the variation in the expression of the selected 'markers' could be explained by differences in tumor tissue purity?
9. Lines 114, 121: '...post-treatment samples....' Do the authors mean surgery samples (samples collected after neo-adjuvant chemo)?
10. Lines 125-126: Cluster 4 genes may be downregulated at surgery due to necrosis (please see comments above).
11. The presence of the EWSR1::ETS fusion oncogene is only confirmed in 75% of the patients included in this study. Given that Ewing sarcoma tumors have been misclassified in the past, have the authors confirmed by other means that the rest 25% are Ewing sarcoma tumors? This is important especially for the clustering shown in Figure 3a. Are these cases evenly distributed across the three clusters?
12. The authors have conducted a number of different analyses:
Figure 2: comparing samples from different time points
Figure 3: unsupervised clustering of primary tumors. This analysis could be of relevance/interesting, as it promises to identify disease biomarkers beyond presence/absence of metastasis at initial diagnosis. However, the authors do not follow this up. The number of samples is also low.
Figure 4: Supervised analysis of prognostic biomarkers
Figure 5: Multiplexed imaging revealing neutrophil involvement and lymphoid cell exclusion in metastatic tumors.
Each finding could be potentially interesting if followed up. Also, how these different analyses/findings could be linked?

Reviewer #4

(Remarks to the Author)

The authors here present a large proteomics dataset comprising primary, relapsed, and metastatic Ewing sarcoma samples. This is an important resource for the field and the data will be of high interest and useful for many investigators. A strength of this study is the large number of samples and inclusion of multiple samples per patient, and the novel contribution of clinically annotated, protein-level datasets for Ewing sarcoma patient samples that are scarce. While the correlative data and unbiased clustering are intriguing, the lack of functional or mechanistic data make this paper useful as a resource but the purely correlative observations preclude mechanistic conclusions about progression.

- The enrichment of ferroptosis related genes post-therapy is interesting, as is the hypothesis that this would sensitize EwS cells to chemotherapy (Line 162). Functional experiments confirming this hypothesis that would make the conclusion more compelling, e.g. ferroptosis agonists/inhibitors, overexpression or knockdown of FTH1, +/- chemotherapy treatment. Likewise the relationship between increased proteasomal protein expression and antigen presentation (Line 204) is intriguing, but not shown functionally. But again statements like Line 264 “Our findings suggest that inhibiting ferroptosis contributes to EWS chemo-resistance, and suggests the potential impact of the proteasome and ubiquitin proteolysis on antigen presentation and DNA damage response (DDR)” seem preliminary if based solely on these data.

- Line 175: The authors state “Interestingly, these clusters reflected prognostic differences between clusters, with cluster 2 having a poorer prognosis than cluster 1 and cluster 3 (Fig. 3b). Kaplan-Meier survival analysis also agreed with other clinical parameters, indicating a significantly higher number of patients diagnosed with non metastatic disease in cluster 3, and highest number of metastatic patients, and relapse-presenting patients in cluster 2 (Fisher enrichment analysis, p -val <0.05 ; Supplementary Fig. 3c-d”. They state that those findings are significant due to a $P<0.1$, ($p=0.0768$) logrank test, it is not clear why $p<0.1$ is used as opposed to the traditional $p<0.05$ and if this was determined pre- or post-testing.

o Likewise Supplementary Fig. 2D only one curve has a Bonferroni adjusted p value <0.05 for EFS, but all are referred to as significant. Overall survival should probably be shown as well, which is not significant.

- Line 377: The authors say that extracellular matrix proteins were removed for data analysis, as well as blood proteins. Can they comment on whether this is a technical necessity based on the processing pipeline, as ECM proteins can be expressed by Ewing sarcoma cells and in fact may correlate with important features such as EWS::FLI1 “low” states?

- The frequent use of Y axes that do not begin at 0 (Fig 2D, 3F, S1D, S1E) sometimes over-inflates differences visually.

- Line 98: Tumor purity was measured using the ESTIMATE method, but those authors comment that its use in sarcomas is problematic: Yoshihara et al. 2013 <https://doi.org/10.1038/ncomms3612>– “The dependency of ESTIMATE on infiltrating stromal and immune cells resulted in some limitations, such as the inability to accurately infer tumour cellularity of hematopoietic or stromal tumours (for example, leukaemia, sarcoma and gastrointestinal stromal tumours) because of the high and tumour-intrinsic expression of stromal- or immune-related genes.” Can the authors quantify tumor purity using a more Ewings tailored approach?

- Line 384: for primary tumor consensus clustering the authors “averaged biological replicates”. Were these replicates taken from adjacent slides of the same region, or do they represent distinct geographic regions of the tumor? If so it would be interesting to see an analysis of the spatial heterogeneity of the primary tumor samples, as opposed to collapsing all regions together.

- Figure 5B: Were all CD15+ cells called as neutrophils during multiplex immunofluorescence? It appears that other cell types, like MDSCs, can sometimes express CD15. Additional markers for each cell type could increase confidence in cell type calling. i.e by staining an adjacent section with additional markers. How confident are the authors that CD15 staining is exclusive to neutrophils in these samples? Likewise CD68 can mark osteoclasts, if any remained in decalcified bone tumor samples.

Reviewer #5

(Remarks to the Author)

Version 1:

Reviewer comments:

Reviewer #1

(Remarks to the Author)

In this revised version of the manuscript, the authors have addressed some of the points previously raised but not others.

Points that have been addressed

1. Calcified vs decalcified samples - the authors have adequately addressed this issue

2. Details of sample prep and data analysis methodology - the authors have now provided details of sample prep and data analysis methods.

There are some responses that are inadequate or problematic

1. Immune and stromal deconvolution - The authors provide information in their rebuttal that only half to two-thirds of the genes that comprise the immune deconvolution of ESTIMATE is represented in their dataset. They then go on to say that they are confident that the results are real because the use of xCell and ConsensusTME give similar results. I'm afraid this logic is flawed. Consistency of the different algorithms is meaningless if they all have poor coverage, i.e. they are all giving similarly incorrect deconvolution estimates especially since there is high overlap in the markers used in each of the algorithms. I also agree with Reviewer 4 and question the suitability of these algorithms for cells of stromal origin. The authors did attempt to validate some of the findings by IHC or multiplex IF, but this is problematic too, see next point.

2. IHC or IF "validation" - I want to thank the authors for taking the revisions seriously by doing a range of different IHC and multiplexed IF validation. However, many of the findings are actually not significant and therefore not consistent with the mass spec data. What is more concerning is that the authors make several claims in the text that the findings are indeed validated even when the quantitative differences are not statistically significant, e.g the PMSA quantification (Figure 5) and many of the immune cell quants (Figure 6 and several of the supp figures) are not statistically different between the groups.

3. Functional validation - In response to the my and other reviewers comments about the lack of functional work on ferroptosis and resistance, the authors have done some drug testing in a paired cell line experiment to demonstrate that combination with GPX4 inhibitor is able to overcome drug resistance. These experiments are unfortunately really superficial and lacking in depth. They raise more questions than answers, for instance, does the cell line pair have similar proteomic changes in the ferroptosis pathways as the tissues, what are GPX4 levels in the cell lines, biochemical analysis that ferroptosis has been inhibited was not done, and doing it in one cell line and one drug is really insufficient to draw conclusions in contemporary cancer research.

4. Removal of ECM proteins - In providing more detail to the data analysis section, the authors revealed that ECM proteins were removed before analysis. In response to reviewer 4, they said "they often vary due to the technical variability in tissue dissection and often do not reflect the cancer cell phenotype itself. This allows normalization of the entire data to ensure more accurate sample ratio calculation". This is very unusual, especially given that the authors have enriched for 80% tumour cell content. It is not the norm to remove ECM proteins before analysis, can the authors provide some data to show what happens to the data if you include ECM proteins? Could the ECM proteins be differentially impacted by decalcification. It has been shown in soft tissue sarcomas that ECM proteins are consistent and representative of histological subtypes and there is similar evidence in proteomic data from other bone tumours like chordoma as well.

5. Specificity of NKX2.2 - in response to reviewer 3, the authors said that for those cases without fusion status, staining for NKX2.2 was done and it was positive for all cases. Unfortunately this stain is not specific for EWS as shown by Jason Hornick and Chris Fletcher (see Hung et al., Modern Pathology 2016). Therefore it is still uncertain if these cases are truly EWS.

6. I still cannot find an explicit limitation section in the discussion for this study.

Minor issues

1. The authors in their intro and discussion highlight EWS as a paediatric disease. However quite a number of their patients are in the AYA age range, so it may be good to reflect this in the text.

2. The authors should provide a table summarising the clinicopath features of the cohort. It is very difficult to ascertain cohort level features from the figures and the supplemental table of individual cases.

3. Who is the pathologist for this study? Is he/she a named author. I only ask this because the diagnosis as well as the assessment of necrosis should be done by a expert sarcoma pathologist.

Reviewer #2

(Remarks to the Author)

The revised manuscript by Gordon et al. entitled 'Proteomic landscapes of Ewing sarcoma unravel immunological regulation of tumor progression' reports on a relatively large series of Ewing sarcoma (EwS) tumor samples that have been subjected to mass-spectrometry based proteomics analyses. While the manuscript has been slightly improved, it remains a rather exploratory analysis that largely stays at the phenotypic description as the authors now admit in the rebuttal letter, and some of the provided new data are still not convincing as outlined below:

1) The new data on the so-called "A673-resistant" cell lines is problematic. First, A673 is a complicated model that shows a relatively high degree of genomic drift (Kasan M et al. 2025 Nat Commun) and that has an unusual second driver mutation (BRAF V600E). This model is alone not suitable to address the research question. Second, the shown "resistance" is defined by a moderate increase (not even a doubling) of the relative IC50 towards doxorubicin. The "resistant" cell lines still respond at lower nanomolar concentrations to this drug. Thus, the used cell line is not suitable for modeling chemoresistance as they are still by and large sensitive toward this drug. The authors should use more suitable and truly resistant models grown in 3D and also include other chemotherapeutics used in EwS. The hitherto presented data are not

convincing to support any mechanistic conclusion.

2) The authors now provide data on stromal contamination (ESTIMATE). Plausibly, the number of identified immune cells correlates with the estimated amount of stroma in each sample. How can the authors exclude a selection bias, that is that the samples taken from the EWS tumors that underwent MS analysis did not have different amounts of stroma per se? How representative is each sample for a given tumor?

3) The authors claim that the highly significant correlation between neutrophils and necrosis is merely driven by one outlier sample. How would the statistics look alike if this outlier sample would be removed (Reviewer Figure 10)? If it would stay significant, then the speculations on the possible biological relevance of neutrophils in the paper should be removed as this is likely driven by just catching up more necrotic tissue in the samples. From the new histological images (Reviewer Figures 3-5) one can actually see a variable amount of necrosis.

4) The histological images showing FTH1 and TFRC immunoreactivity show very dramatic staining intensities that do not correspond to the moderately different mean or median (not clear from the figure and legends) H-scores. The authors should show images that are representative for the medians and not extreme phenotypes. Can the authors show a simple iron stain. It is plausible that upregulation of both proteins is a mere epiphenomenon of hemorrhage and bleeding into the tumor tissue post-NACT. Also, both immunostains show huge background smears.

5) The applied statistical tests should be clearly stated in the Figure Legends. It is currently very difficult to judge which bar graph has been assessed with which test (and post-hoc test)

Reviewer #3

(Remarks to the Author)

The authors have responded adequately to my comments.

I still have a few points to be addressed:

1. Patients with Ewing sarcoma normally undergo surgery only after neo-adjuvant therapy. At initial diagnosis they are normally subjected to core needle biopsy. The authors should double check if the tumor samples used in this manuscript and correspond to initially diagnosis (primary tumor) they were indeed surgical samples. If this is not the case, then their response to Reviewer #1-Comment C (.....Of note, EWS biopsies are typically surgical biopsies,.....) is not accurate.

2. Study S-BIAD1597 cannot be found in the BioImage Archive (<http://www.ebi.ac.uk/bioimage-archive>).

3. Supplementary Fig. 3b-c. Why the patient numbers do not much for EFS and OS analysis?

4. Supplementary Fig. 3c. FLT plot – you may want to show FLT low in red.

5. Reviewer still finds Figure 1e misleading – FLI1, EWSR1, and ERG and not Ewing sarcoma markers. Ewing sarcoma tumors are driven by EWSR1::FLI1/EWSR1:ERG fusion oncoproteins. CD99 is not a very specific marker. The authors may want to provide further explanation in the figure legend. DO the authors assume the FLI1 protein measurements correspond to the fusion protein for example? the same applies to EWSR1 and ERG.

Reviewer #4

(Remarks to the Author)

The authors have responded to prior comments: new data have been added to address ferroptosis resistance and conclusions tempered appropriately. Two minor corrections are suggested: The EWS::FLI1 fusion should be labeled with :: per latest nomenclature standard and it is better called a "fusion gene" rather than a "transgene".

Reviewer #5

(Remarks to the Author)

Version 2:

Reviewer comments:

Reviewer #1

(Remarks to the Author)

I thank the authors for undertaking an extensive revision of the manuscript and have largely address most of my comments.

Given that the authors now show that the removal of ECM proteins has no impact on the analysis and interpretation of the data, I would request that this step be removed from the pre-processing in the final manuscript. As this reviewer suspected, the authors have confirmed that this step is unnecessary - it is important that the sarcoma field is aware of this and is not misled that this pre-processing step is required to remove such "contaminants". Furthermore, the authors should also report any ECM proteins that were quantified in this study as part of the final proteomic dataset as this is important for the EWS field moving forward given that the TME is an increasingly key area of research as a potential source of targets and biomarkers.

Reviewer #2

(Remarks to the Author)

The further revised manuscript by Gordon et al. entitled 'Proteomic landscapes of Ewing sarcoma unravel immunological regulation of tumor progression' reports on a relatively large series of Ewing sarcoma (EwS) tumor samples that have been subjected to mass-spectrometry based proteomics analyses. While this reviewer acknowledges the additional effort made by the authors that partially addressed significant concerns, the manuscript remains a rather exploratory analysis that largely stays at the phenotypic description. The strength of the paper lies in the unique and large proteomics dataset with matched clinical annotation, which would – if (and only if) made freely available to the scientific community – constitute a strong argument for publication. However, the authors still overinterpret some of their findings and this Reviewer can only again recommend to tone down or even omit some of the functional conclusions as they are not fully supported by the data. In the opinion of this Reviewer, the paper would be suitable for publication as a resources article and be much better by omitting some of the overstatements as outlined below. The paper would be actually stronger as a pure resources article, and perhaps some of the functional/mechanistical conclusions would fit better in follow-up papers once more supportive data is available.

1) This Reviewer thanks the authors for their efforts to generate 3D spheroid models, which is an improvement per se, and recognizes their efforts to generate a second model for "chemoresistance" in EwS (derived from RDES cells). Yet, the minimal differences in sensitivity towards Doxorubicin and the modest differences in sensitivity towards Etoposide are not convincing to declare these new models as "chemoresistant". The situation is even worse for the previously reported A673 model. This whole set of experiments, while going in the right direction, is an example for an overstatement. The paper would be better if these weak data are omitted from the otherwise strong resources paper and would better fit in a solid follow-up paper with truly convincing chemoresistance models. It is simply not convincing that if even the "better" model can still be killed by Doxorubicin in doses at the nanomolar range, to declare them as "resistance models". Accordingly, conclusions (e.g. regarding ferroptosis) from these so-called "resistance" models are not valid. The observed differences in ferroptosis could be due to multiple other factors apart of the so-called "resistance", e.g. by metabolic selection bias during the long-term culture (see Methods). Similarly, the minimal differences in protein abundance of "ferroptosis-related" proteins in "resistant" versus parental cells become only visible if the Y-axis of the shown graph is cropped (data between 2 and 16 AU not shown). The abundance might be statistically significant but the actual difference is minimal and the biological relevance of these differences is unclear. The strong resources aspect of this paper is actually diluted by the attempt to "validate" potential findings from the clinical cohort in preliminary experiments. This Reviewer strongly recommends to take these data out of this paper and to validate them thoroughly in follow-up studies.

2) Similar to point #1, the role of neutrophils and necrosis is a weakening aspect of this paper. In their rebuttal letter, the authors refer to a previous paper of the same group (Shukrun et al., *Cancer Sci.*, 2024; doi:10.1111/cas.15992) that postulated that neutrophil extracellular traps (NETs) are associated with poor outcome. It should be noted that this previous study was based on a very small cohort (n=46) of EwS patients of which only n=6 died from disease. Some of the main conclusions of this paper are based on graphs such as this presented in Figure 3b and c that show highly overlapping data points across groups and only minimal changes overall. These figures of the paper of Shukrun et al. (*Cancer Sci.*, 2024) shows that patients with relapse or death of disease have actually more or less the same levels of NETs. The result may be statistically significant (which appears questionable given the massive overlap of data points), but are they biologically and/or clinically relevant? What is the clinical meaning if a patient has e.g. a "NETs release (%)" at diagnosis of 50%? The patient could easily fit in either group (relapse vs no relapse; live vs dead). This Reviewer is not convinced that the abundance of neutrophils and/or NETs is of any clinical meaningful relevance for patient prognosis, risk-stratification, and decision-making. Hence, also the findings of the current study, which are likely based on an extension of the previous cohort, are likely not clinically relevant. Again, the paper would be better off by omitting overstatements on the prognostic/clinical role of these findings and by presenting the current paper as a pure but honest resources article. Given the prior data (Shukrun et al. 2024) and data of the current study, this Reviewer again recommends to either strongly tone down on the relevance of neutrophils and NETs in EwS or even better to take these data out and thoroughly validate them in an independent follow-up study.

Reviewer #3

(Remarks to the Author)

The authors have adequately addressed my previous comments.

Version 3:

Reviewer comments:

Reviewer #1

(Remarks to the Author)

The authors have addressed my comments in this latest version of the manuscript.

Reviewer #2

(Remarks to the Author)

Thank you for the thoughtful revisions. The main text is now appropriately tempered; however, the abstract still reads more assertively than the data support.

In particular, the abstract retains bold claims about chemo-resistance and includes the sentence:

> “We found ferroptosis inhibition as a potential mediator of EWS chemo-resistance and identified novel subclasses of EWS that link the tumor immune landscape with DNA damage repair, ubiquitin-related proteins, and patient prognosis. Multiplexed immunofluorescence imaging supported the association between poor patient prognosis and tumor neutrophils, and the association of macrophages and T-cells with a favorable prognosis.”

As currently written, this phrasing implies causality that I do not see demonstrated in the manuscript. If you wish to keep these points in the abstract, I recommend reframing them as correlations or hypotheses and aligning the strength of language with the evidence shown. For example, consider wording such as:

> “Our analyses suggest that ferroptosis pathways may be associated with chemotherapy response in EWS, and we delineate molecular subclasses that correlate the tumor immune landscape with DNA damage repair, ubiquitin-related proteins, and patient outcome. Multiplexed immunofluorescence indicate possible associations between neutrophils and poorer prognosis, and between macrophages/T cells and more favorable prognosis.”

Relatedly, the title remains unchanged and, in my view, is potentially misleading. The term “immunological regulation of tumor progression” implies a demonstrated causal role; the data presented support association rather than regulation. Please adjust the title to avoid implying causality. A more accurate and appropriately cautious title might be:

“Proteomic landscape of Ewing sarcoma primary tumors and metastases,”

To summarize: the dataset is sound and valuable, and of clear interest. My recommendation is to harmonize the abstract (and title) with the more measured conclusions in the main text by avoiding causal language where only associations are shown, and by defining terms precisely. This will strengthen the manuscript and ensure readers take away conclusions that are fully supported by the data.

Point by point response to reviewers

Reviewer #1:

In this study, the authors undertake mass spectrometry profiling of a large cohort of EWS samples. They perform descriptive analyses of the proteomic data and identify some interesting findings. The strength of the study is the large number of cases analysed for this rare entity. The weaknesses are detailed in comments below.

We thank the reviewer for the overall positive assessment of the work and the detailed evaluation of the manuscript. In the revised version we answered all the reviewer's concerns, as detailed below.

Major comments

1. Sample and preparation details are lacking-

A. Bone sarcoma samples are often decalcified, a process which is known to impact the quality/quantity of biomolecules. Can you authors confirm if the samples were decalcified? If so, can the authors need to provide data detailing the impact of protein yields, protein IDs and protein quantification of decalcification compared to frozen or non-decalcified matched specimens in a subset of the cohort. Without this information it is difficult to assess how much of the data reflect artefacts introduced by the decalcification process. And therefore, how much of the generated proteomic data is actually representative of the disease.

We thank the reviewer for this comment. As indicated by the reviewer, some of the tissues in our cohort were decalcified in the clinic before their embedding in paraffin. The decision regarding the decalcification depends on the extent of bone tissue involvement. Previous studies have shown negligible impact of decalcification on the proteome (e.g. Iglesias-Gato 2018), which is also in line with the fact that this procedure is routinely followed by standard IHC. Potentially other biomolecules, such as nucleic acids, metabolites and lipids are more sensitive. Nevertheless, following this reviewer's comment we investigated the potential impact of decalcification using several experimental and computational analyses.

*1. To examine the calcification impact **experimentally**, we excised mouse sternum and took one part of the tissue for a decalcification procedure (24 h incubation in 4% formalin and 18hr incubation at 37C with a plate on stirrer in EDTA (MOLDECAL), as performed in the clinic) and another part was directly fixed. We followed with sample preparation using the SP3 protocol, as described in the manuscript, and continued with MS-based proteomic analysis. Overall, we identified a similar number of proteins and peptides in all samples. The quantitative comparison of the protein expression levels showed an average correlation of 0.96 between decalcified and non-decalcified tissue. A T-test to compare these two sample groups showed a statistically significant difference in only 148 proteins, out of a total of 8562 identified proteins, which correspond to only 1.78% of the identified proteome. Importantly, since these samples do not originate from the same tissue block (since one of them was decalcified), we cannot attribute these changes to the decalcification procedure. These results are presented in **Reviewer Figure 1**.*

2. Re-analysis of the patient derived proteomic data showed no difference in the number of identified proteins between decalcified and non-decalcified tumor samples focusing exclusively on primary samples

to eliminate confounding effects from other tumor progression stages. (Reviewer Figure 2). Panel a is also added to the revised manuscript as **Supplementary Figure 1c**.

3. Examination of the proteomic differences between decalcified and non-decalcified samples showed no separation associated with the decalcification procedure.

Based on all these analyses we are confident that there is no risk that our results were confounded by this technical difference between samples.

Reviewer Figure 1: Experimental assessment of decalcification effects. a) H&E images of decalcified (left) and non-decalcified mouse sternum. b) Dot plot compares the number of proteins identified in non-decalcified and decalcified mouse tissues. c) Pearson correlations of mouse samples show a range from 0.93 to 0.98. d) Volcano plot shows significantly changing proteins between non-decalcified and decalcified sternum mouse tissue (permutation-based FDR <0.05 s_0 = 0.1).

Reviewer Figure 2: Comparing decalcified and non-decalcified samples. a) Dot plot shows the number of proteins identified from decalcified and non-decalcified primary EWS samples. b) Pearson correlations of all primary samples are depicted in the color bar, representing a range from 0.7 to 0.96.

B. In the methods, the authors state that samples were dewaxed based on "standard protocol" - what is this standard protocol please provide details. Also it seems there is no reversal of crosslinks step, is this the case?

We apologize for the missing information. The detailed protocol is now added to the revised version. Each FFPE tumor sample was deparaffinized with two xylene washes, followed by rehydration through a series of ethanol washes: twice in 100% ethanol, once in 96%, and once in 70%. The samples were then placed in distilled water (DDW) until scraping.

In addition, we added the de-crosslinking step which includes tissue lysate incubation for 90 min in 95°C.

C. Authors state that samples were macrodissected for 80% tumour cell content? Was this done for the tiny biopsies? Is it possible to macrodissect from tiny biopsies? Does the 1-4 FFPE sections of 6um thickness apply to biopsies as well?

Macrodissection was performed to isolate regions of 1 mm² or larger. In our experience, we can extract ~0.5 µg protein from ~1 mm² tissue (of ~6 µm thickness), and we only need 2 µg for protein digestion to reach the presented analytical depth. Therefore, the size of the biopsy does not limit our ability to perform the analysis. Of note, EWS biopsies are typically surgical biopsies, due to their location and the fact that these are pediatric patients. Finally, the main limitation is not the size of the biopsy, but its cellularity, as we carefully dissected only tumor regions.

2. In general for every single section, there is a lot of descriptive analysis comparing different groups or clusters. For each of these sections and comparisons, there is a need to undertake orthogonal validation of the findings by immunohistochemistry. This includes Figure 2 different timepoints, in particular pre and post treatment, primary vs mets, Figure 3, different proteins expressed in different consensus clusters, Figure 4 different proteins associated with clinical parameters. Without IHC validation, it is not possible to ascertain if the proteomics findings are not just artefacts, particularly in the context of decalcified tissue.

We thank the reviewer for this important comment. We invested major efforts to validate our findings using orthogonal approaches, including immunohistochemistry, immunofluorescence and functional validation.

a) IHC of FTH1 shows significantly higher expression in post neoadjuvant-treatment (NACT) samples compared to pre-treatment ones, in agreement with the proteomics data. Quantification of the IHC signal shows higher expression in the proteomic region of interest (ROI) and when quantified across all cancer cell (NKX2.2+) regions and across the entire slide (**Reviewer Figure 3 and new Figure 3a-b**).

Reviewer Figure 3: IHC of FTH1 comparing primary and post neoadjuvant chemotherapy samples. a) Representative IHC images of FTH1 in matched primary and post-neoadjuvant chemotherapy samples. Dot plots show the quantification of FTH1 in primary and post-neoadjuvant chemotherapy samples. Error bars represent the SEM values. Quantification was based on proteomic ROI, NKX2.2 positive cells, and whole tissue.

b) IHC of TFRC (transferrin receptor 1) shows significantly higher expression in post-NACT treatment samples compared to pre-treatment ones. Quantification of the IHC signal shows higher expression in the proteomic ROIs, when quantified across all cancer cell regions, and across the entire slide (**Reviewer Figure 4 and new Figure 3c-d**). TFRC is a key mediator of cellular iron uptake, facilitating the transport of transferrin-bound iron into the cell. FTH1 acts as a counter-regulatory protein by sequestering intracellular iron, limiting ROS generation and lipid peroxidation availability. By staining for both TFRC and FTH1, we aimed to capture the balance between iron uptake and storage, which is critical in determining ferroptosis sensitivity or resistance.

Reviewer Figure 4: IHC of TFRC comparing primary and post-NACT samples. a) Representative IHC images of TFRC in primary and post neoadjuvant chemotherapy samples. Dot plots present the quantification of TFRC in primary and post neoadjuvant chemotherapy samples. Error bars represent the SEM values. Quantification was based on the proteomic ROI, NKX2.2 positive cells, and whole tissue.

c) IHC of RRS1, which was found to be more highly expressed in the relapsed tumors relative to the primary tumors, showed higher nuclear staining in the proteomic ROIs of relapsed tumors. These results are presented in **Reviewer Figure 5** and **new Figure 2c-d**.

Reviewer Figure 5: IHC of RRS1 comparing primary and local relapse samples. a) Representative IHC images of RRS1 in primary and local relapse samples. b) Quantification of RRS1 in primary and local relapse samples. Error bars represent the SEM values. An unpaired T-test was used, and quantification was based on proteomic ROI.

d) Multiplexed imaging was also performed to validate specific proteins in the consensus clusters and in relation to patient prognosis. These validations were performed in the context of the immunological changes between clusters, and included PSMA3 and HLA-DR, which showed higher expression in cluster 3, in addition to neutrophil markers (MPO and CD15, cluster 2), and macrophage and T-cell markers (CD68, MRC1, CD8 and CD4 cluster 3).

Altogether, these experiments validated the expression changes in all tested proteins. In addition, they showed the protein localization in the tissue, thereby providing additional information.

These results are partially presented in **Reviewer Figure 6** and in the revised manuscript in **Figure 4h**, **Figure 5g**, **Figure 6d-i**.

Reviewer Figure 6: Multiplexed immunofluorescence of immune-related proteins. a) The proportion of all neutrophils (only MPO+ and both MPO+CD15+) in three clusters was calculated as a percentage of the total cell population. b) PSMA3 staining in the three EWS clusters of primary tumors shows highest expression in cluster3. Images and statistics are added in Figure 5. c) Macrophage staining with anti-CD68, anti-HLADR, and anti-MRC1. Quantification of cell counts is added for each cell subset based on co-staining of the markers at the single cell level. Data are shown as mean \pm SEM. d) T-cell staining in EWS consensus clusters. e) Neutrophil staining with anti-CD15 and MPO.

e) Beyond the staining validation we also further investigated the functional importance of ferroptosis to chemotherapy response in EWS. To this end, we tested the impact of the ferroptosis inducer RSL3 (GPX4 inhibitor). We used the recently published EWS chemotherapy-resistance cell line model based on A673 cells (Scarborough et al, 2020). Initial analysis showed that A673-resistant cells are less sensitive to treatment with doxorubicin (as expected) and RSL3. Next, we treated parental and chemo-resistant cells with increasing concentrations of doxorubicin and increasing concentrations of RSL3. Modeling the effects of these drugs using SynergyFinder (Ianevski et al. 2022) showed strong synergism between these drugs, suggesting the importance of ferroptosis in mediating chemo-resistance. These new results are shown in Reviewer Figure 7 and added to the revised manuscript in new Figure 3e-h.

Reviewer Figure 7: Enhancing doxorubicin sensitivity through ferroptosis induction. a) Dose-response curve for doxorubicin in A673 parental and resistant cells. IC50 values were calculated using PRISM software with nonlinear regression curve fitting. b) Comparison of IC50 values for doxorubicin when treated alone and with RSL3 in A673 parental and resistant cells. c-d) Doxorubicin and RSL3 synergy models for A673 resistant (c) and parental (d) cell lines.

3. Figure 1d, authors state that these proteins and markers are highly expressed. Highly expressed is a relative term. Compared to what? This should be compared to normal bone rather than just within an EWS cohort to demonstrate that the markers are highly expressed?

Throughout the paper the reference to the expression level is relative to other samples in the cohort. Specifically in figure 1d we refer to the proteins' expression within the sample, relative to other proteins in the same sample. The intensity itself does not provide an accurate measure of absolute abundance, but it

provides a good estimate. To remove any potential confusion, we removed these statements from the revised version.

Notably, in the analysis of clinical samples of EWS it is impossible to use normal control. The bone is not true control since the cancer did not initiate in the bone. The cell of origin is still controversial, but the most accepted assumption is that these are mesenchymal stem cells. Analyzing these cells from tissue sections is not possible, and a comparison to cultured MSCs is expected to show mostly technical artifacts due to the sample state.

4. The ESTIMATE algorithm was used several times in the manuscript, once to estimate stroma content and the other for immune cells. Can the author comment on the coverage of their protein data and the markers used by ESTIMATE. if the overlap is low, is it appropriate to use ESTIMATE for the proteomics data?

The proteomic data covered 105/141 and 85/141 proteins of the immune and stromal ESTIMATE signatures, respectively. To ensure that ESTIMATE analyses are reliable, we also compared the immune scores determined by ESTIMATE to those determined by two additional algorithms, ConsensusTME and xCell. We found a high correlation between these algorithms (Reviewer Figure 8) and therefore retained the previous results.

Reviewer Figure 8: Correlation analysis of the TME scores derived from xCell, ConsensusTME and ESTIMATE algorithms. Scatter plots compare the xCell TME scores (left) and ConsensusTME scores (right) with the ESTIMATE scores across tumor samples.

5. Comparisons between primary and post-chemo surgical specimens. Because there are no surgery alone specimens (without chemo treatment), the authors cannot exclude the possibility that some of the protein changes seen are the result of surgery rather than chemotherapy. It is known that surgery leads to the activation of inflammation and immune related pathways some of which are similar to those seen in the authors dataset. The authors should consider this possibility.

We thank the reviewer for this question. As indicated above, in the case of EWS the biopsies are also taken under surgery, we therefore do not expect any such artefacts in the data.

6. The authors make multiple claims which is not supported by the data, particularly lack of functional data to establish causation. These statements are pure conjecture so please remove them.

"Altogether, we propose that these proteins may confer treatment resistance via inhibition of chemotherapy-induced ferroptosis, and induction of immune response."

"Our findings suggest that inhibiting ferroptosis contributes to EWS chemo-resistance, and suggests the potential impact of the proteasome and ubiquitin proteolysis on antigen presentation and DNA damage response (DDR)."

We apologize for some over-stated claims. These are removed from the text. We adapted the writing to the addition of functional validation of the ferroptosis results.

7. All survival analyses, such as figure 3 and the analysis of the RNA data needs to be subjected to multivariable analysis which includes known prognostic/confounding factors to assess if the consensus clusters or proteins are independent prognostic variables.

*We thank the reviewer for this important comment. Following this reviewer’s comment, we performed an additional multivariate analysis to examine the contribution of other clinical confounders to the cluster separation. These analyses included the metastatic state (metastases at diagnosis), the tumor location in the pelvis, and the surgical treatment. The only significant factor was the metastatic state at diagnosis. These new analyses are presented in **Reviewer Figure 9 and Figure 5** of the revised manuscript.*

Cox regression could not be performed with the publicly available RNA data since other clinical parameters were not provided.

Reviewer Figure 9: Cox regression analysis.

8. It is surprising that there is no discussion of the limitations of this study. This is a very descriptive, hypothesis generating study with no validation of findings in an independent cohort. Single institution studies are subject to selection bias. This is just one of several limitations that need to be considered in the discussion.

We added a discussion of the study limitations to the revised version.

9. Can the authors make clear how missing data was dealt with in the dataset? they report >10K proteins but each condition has ~7K, so how was missing data handled?

The Methods section includes a detailed description of the data processing and the imputation. From the data of more than 10K proteins, we removed ECM proteins and removed proteins that did not appear in at least 70% of the samples. This filtration ensured that we do not over-impute the data and have a relatively low number of missing values per sample. Next, we imputed these values by generating an artificial normal distribution with a downshift of 1.8 standard deviations from the overall distribution, and a width of 0.5 of the distribution. This is a very common way to impute missing values when missingness is a result of low expression, as in our case. This information is provided in the Methods section "Data preprocessing and statistical analysis".

Reviewer #2:

The manuscript by Gordon et al. entitled 'Proteomic landscapes of Ewing sarcoma unravel immunological regulation of tumor progression' reports on a relatively large series of Ewing sarcoma (EWS) tumor samples that have been subjected to mass-spectrometry based proteomics analyses. The authors conduct a variety of subsequent bioinformatics analyses and compared (mainly unmatched) samples from treatment-naïve primary tumors and samples from metastatic lesions or relapses that are mainly post-treatment samples. These comparative and purely correlative analyses led the authors to conclude that ferroptosis might play a role in chemo-resistance. Further, they identified by various clustering analyses subgroups of primary tumors that may be associated with diverse signaling pathways and immune profiles. Some validation studies using various immunohistological markers suggested associations with metastasis at diagnosis, which led the authors to conclude that infiltration of the tumors by neutrophils may be associated with a worse patient outcome.

While this study reports on a large series of samples with proteomic data, the study remains purely descriptive in nature and most conclusions are speculative and not underpinned by any functional experiment. Some of the conclusions appear overstated and the fact that some major prognostic subgroups could be also found in a published cohort based on bulk-level mRNA expression data somewhat undermines one of the major arguments of the authors to claim novelty and importance of proteomics for advancement of EwS research.

Apart from these general concerns, there are many major concerns that limit the enthusiasm of this Reviewer for this study:

We thank the reviewer for the thorough review of the manuscript. In the revised version we added more validations, including a functional validation of the importance of ferroptosis to chemoresistance and addressed all other comments.

1) It is simply unacceptable that the supplementary tables contain patient-specific identifying criteria, such as the precise date of diagnosis, date of birth, date of death etc. These data, if published in that form would make it very easy to unambiguously identify patients, which is not in the interest of our patients. It is hard to believe that the authors have an IRB approval for this and this Reviewer strongly recommends to de-identify the patients and remove such (unnecessary) data from the tables. Then, the statement in the methods part of “all samples were anonymized as defined in the study protocol” would be valid. The patient-specific data given in the current table is the more so critical since Israel is a relatively small country (population-wise) and since EwS is a very rare disease, patient-specific information regarding date of birth, date of death, and Israel as a likely country of residence and the diagnosis of EwS, each and every patient can be easily identified (not only by close relatives). It is the first time this Reviewer sees such a data leak in a submitted paper, and frankly speaking, this is NOT ok.

We apologize for this mistake. All patient data is anonymized and approved by the IRB committee. However, our error was to include the dates and age in the table. These were removed from the supplementary tables. We agree that in Israel this error is more critical, and therefore thank the reviewer for this comment. However, we should add that not all patients are Israeli, but some of them arrive here for the treatment; and not all patients were diagnosed or treated in one center throughout the entire course of their disease.

2) For almost all correlative associations of proteomics subgroups with patient parameters and other phenomenological findings it is unclear whether these events are the drivers or passengers of the observed phenotypes. For instance, the presence of neutrophils could be a mere expression of more necrotic tumor tissue within a given tumor due to higher proliferation rates that overwhelm the tumor’s capacity to form new vessels, which may lead then to higher rates of metastasis. As such, neutrophils are just an epiphenomenon of biological mechanisms that are as yet not defined or discovered. Also, in this instance, multiple comparisons of immune markers with phenotypes have been done. If corrected for multiple testing, the above-mentioned associated is likely rendered non-significant. Hence, subsequent conclusions are not underpinned by solid data.

In this manuscript we do not claim that the observed results drive the phenotype, but in many of these cases, we are the first to show the phenotype. We are the first to show the patient clusters, the first to show their association with the immune profiles and the connection between the immune profiles and the proteome. Notably, this is all done in patient samples, which represent the true clinical variability. Understanding the causal relations between all these results will be the subject of many more years of research and should not be expected within one manuscript.

*To address this reviewer’s comment about the association between neutrophils and necrosis, we performed a pathological analysis of necrosis in the cancer regions of the tumors included in the imaging, as typically done in the clinical routine. We correlated the percentage of necrosis in the tumors with the overall neutrophil counts by staining (**Reviewer Fig. 10**). Overall, we found a positive correlation, which is driven*

by one sample with very high neutrophils and very high necrosis. Based on these results we cannot determine that neutrophil infiltration is driven by tumor necrosis.

Reviewer Figure 10: Correlation between tumor necrosis and neutrophil counts. Tumor necrosis was determined by pathological examination and neutrophil counts were determined by tissue staining.

3) Similarly, many downstream pathway analyses yield only generic pathways potentially associated with metastasis. For instance, what is the exact meaning of a downregulation of ‘DNA-dependent DNA replication initiation’ in association with disease-free survival? What does that mean? How is this validated? And what do we truly learn from these associations? Are they driving the phenotype (DFS) or just epiphenomena?

*Enrichment analyses typically show the overall expression of specific pathways, as defined by external, well-accepted databases (e.g. gene ontology and KEGG). They are not expected to prove causal relations but show that these pathways are more highly expressed and **suggest** that they **may** be more active. Beyond that, causal relations can be suggested based on previously published literature. To specifically address this reviewer’s example, “DNA-dependent DNA replication initiation” suggests higher cell proliferation. Association between higher cell proliferation and low disease-free survival is a very-well established link in most cancer types. In this type of research, which provides a large data resource, it is impossible to follow-up on every point. In our work we decided to emphasize the ferroptosis and immune-related results but did not follow-up on other results. Nevertheless, we believe that presenting the entire dataset is valuable for the community and could be a good starting point for other studies.*

3) The identification of 13 proteins in overlap of “metastasis”, “relapse”, and poor “DFS” would have been a good start for follow-up, but again no meaning can be drawn from this (Fig. 4h).

The protein network presented in Figure 4 includes multiple neutrophil proteins, among them MPO, AZU1, LCN2, PRTN3, ITGAM, LRG1. Indeed, we follow up on the neutrophils. In the revised version we added staining of neutrophils. As mentioned above, additional follow-up studies are ongoing but are beyond the scope of this research.

4) The association of immune scores with different proteomics subclusters of which one (cluster 2) is associated with a trend for worse outcome is interesting. Yet, as the result is not nominally significant, it

remains unclear whether this is a trustworthy result and whether all subsequent explorations/speculations are worth being discussed in the current extension.

*We appreciate the reviewer's interest in these results. Following the reviewers' requests we repeated these analyses on additional tumors, and added staining of additional immune markers, including MPO, CD4, CD8, HLA-DR, in addition to CD15 and CD68 that were also included in the previous submission. The results show some significant differences that reflect a clear difference between clusters. These results are also reinforced by Cox regression analysis that was added in the revised version. Analyzing such a rare type of cancer is challenging due to the limited size of available cohorts. Nevertheless, the analyses show clear differences. The neutrophil results are added to the revised manuscript in **Figure 4 and Supplementary Figure 4**. The macrophage and T-cell results are added as **Figure 6 and Supplementary Figure 6**. These results are also presented here as **Reviewer Figure 6** (above).*

*Beyond the staining, we also used the xCell algorithm to deconvolute the proteomic data and predict the number of immune cells in each sample, based on the proteomics data. These results suggest higher levels of neutrophils in cluster 2 compared to the two other clusters (**Reviewer Figure 11 and Figure 6c-d**).*

Reviewer Figure 6: Neutrophil and macrophage estimation using xCell. Dot plots of cell type xCell's scores for neutrophils and M2 macrophages. Each dot represents a patient. Significance was calculated with ANOVA and Bonferroni multiple comparisons test.

5) As shown in Figures 1a/b, the tumor samples came from mostly unmatched primaries/metastases of different individuals with very different localizations. Some sites were only represented by a very few cases (such as pelvis and bone mets). Hence, some of the findings reported in this paper could be still subject of changes once larger sample series would be available. It is also unclear to what extent the findings on immune infiltrates etc. are a mere reflection of the anatomical site and or location of the tumor (soft-tissue versus bone).

*We would like to emphasize that the entire study of immune infiltrates was performed on the primary tumors before any treatment. We agree that drawing conclusions from the different progression states would be difficult, and therefore we did not do that. Overall, the tumors in the presented cohort originate from different bone locations, as detailed below (**Reviewer Figure 12**).*

Reviewer Figure 7: Tumor locations across clusters. The stacked chart shows the distribution of tumor locations across three consensus clusters. Each cluster is divided into the primary tumor anatomical regions: lower limb, upper limb, pelvis, ribs, vertebra, skull, and sacrum.

6) The statement that ferroptosis inhibition may be an event to overcome chemo-resistance (see abstract) appears as a strong overstatement and is sole based on a correlative observation that proteins involved in ferroptosis show high expression in post-chemotherapy samples. For these samples, it should be noted that the mere finding that the samples are from a post-chemotherapy condition is insufficient to conclude that these samples were chemo-resistant. In many patients, chemotherapy needs to be interrupted (e.g. due to cumulative toxicity) and is not per se a reflection of resistance. Also, all chemotherapies applied in EwS lead to cell death in a non-ferroptosis-dependent pathway. Hence, the upregulation of ferroptosis-related genes is not per se a reflection of any biology related to relevant cell death. Again, functional analyses are completely lacking.

We thank the reviewer for this comment. We acknowledge that there are cases in which patients stop chemotherapy prematurely. However, in our cohort we included only patients who received the full regimen and all post-NACT tumors had a poor response to neoadjuvant chemotherapy (<90% necrosis). This information is now added to the revised Methods section.

*In the revised version we also increased the number of validations, with a specific focus on ferroptosis. First, we validated the increased expression level of ferroptosis related proteins using orthogonal IHC staining. Second, we used an EWS chemo-resistance cell line model and showed that inductions of ferroptosis with RSL3, a GPX4 inhibitor has a synergistic effect with doxorubicin (DOXO). These results show the functional importance of ferroptosis in chemoresistance. The results are presented here in **Reviewer Figure 7 (above)** and presented in the manuscript in New **Figure 3**.*

Reviewer #3 :

This is a very exciting study looking at the proteome of over 100 Ewing sarcoma tumor samples from 72 patients. To my knowledge this is the first study doing MassSpec-based analysis on Ewing sarcoma tumor tissues and hence it deserves attention. Ewing sarcoma although is a disease with high clinical heterogeneity, Ewing sarcoma tumors are genetically homogeneous. Transcriptome/epigenome studies have not yet revealed strong biomarkers that could explain clinical differences. Examining the proteome is imperative.

We thank the reviewer for the positive evaluation of the work. In the revised version, we answered all questions and comments and added a substantial number of validations.

While I am enthusiastic about this study, I do have a few concerns outlined below (not in a particular order).

1. The treatment protocol of Ewing sarcoma is not accurately presented. Patients undergo neo-adjuvant chemotherapy, followed by surgery (in most cases) and/or radiotherapy (in some cases), which is followed by maintenance chemotherapy. As such Figure 1b is not accurate (therapy cycles indicated with a radiotherapy icon is wrong). Metastasis also occurs at initial diagnosis. Relapse and metastasis are not independent events (relapse can be localized and metastatic). Figures 1a,b at their current version they do not add anything to the story and could be omitted as well. If kept, they should be designed with higher precision and with some informative content.

Calling surgery samples post-chemotherapy (and in some cases post-treatment) samples is misleading, given that most of these patients continued to receive chemo after surgery.

*We appreciate this comment, and we corrected the illustration to better reflect the treatment course. The “post-treatment” samples are indeed surgical samples after neoadjuvant chemotherapy (NACT), and the patient receive additional cycles after surgery. To clarify this point, in the revised manuscript we corrected the illustration in **Figure 1 (Reviewer Figure 13)**, we added the necessary explanation in the cohort description and changed the writing throughout the manuscript from “post-treatment” to “post-NACT”.*

Figure 8: Schematic illustration of patient treatment and tumor locations.

2. Surgery samples are known to be highly necrotic. Did the authors manage to dissect live cancer cells from these samples? How do they check for live cancer cells? Is the ESTIMATE algorithm run on dissected samples (MassSpec/protein level data)? It may be important to specifically indicate the purity of surgery samples on Supplementary Figure 1e.

*We agree that necrotic regions are much more substantial in surgical samples after chemotherapy. We therefore performed H&E staining of all regions and followed with a careful pathological examination. Next, we carefully dissected these regions. Following the reviewer suggestion, we color coded (orange) the ESTIMATE tumor purity of post-neoadjuvant chemotherapy samples in **Supplementary Figure 1f (Reviewer Figure 14)**. Tumor purity is lower in some of them, but the differences are not substantial.*

Reviewer Figure 14: Tumor purity score of EWS samples. The dot plot presents the Tumor-PurityScore for each sample, assessing the cancer percentage in each tumor sample based on ESTIMATE. Post-neoadjuvant chemotherapy samples are colored in orange.

3. Lines 96-98: it is not a surprise that surgery samples show the lowest correlation with other time points given that these samples are highly necrotic. Could it be that most cells they profile are non-cancer cells (i.e. hematopoietic)?

*We agree that the marked changes upon treatment are not surprising. Following the previous comment about the ESTIMATE tumor score of the surgical samples, the pathological examination of the tumors, and the fact that we see similar expression level of known EWS markers (e.g. LINGO1, EWSR1, FLI1, CD99, NKX2.2), we have no reason to think that we do not analyze EWS cells. In the revised version we also included several IHC images that are used to compare pre-treatment samples and surgical post-treatment samples (**Reviewer Figures 3 and 4, above, and Figure 3a, c**). We also stained for the EWS marker NKX2.2 the same regions in consecutive slides, to ensure that we only included EWS cancer regions in our analyses **Reviewer Figure 15** shows NKX2.2 staining of all post-neoadjuvant chemotherapy samples.*

Reviewer Figure 15: IHC images show NKX2.2 expression in post-NACT samples.

4. According to supplemental table 1b all surgery samples come from bad responders with 40% to 90% necrosis. It may be good that this point is discussed in the manuscript.

We thank the reviewer for this comment. We added the following explanation to the results section line 75: "Among those, the cohort included 62 primary biopsy samples used initially for tumor diagnosis, nine post-neoadjuvant chemotherapy surgical samples with poor response (<90% necrosis)."

5. EWSR1 protein is expressed in most cell types. Perhaps not a good marker for Ewing sarcoma. Can the fusion oncogene be detected with MassSpec?

We agree that EWSR1 is not a specific EWS marker. Unfortunately, we could not search for the fusion protein, since every patient has a slightly different translocation, and the exact site is not routinely tested in the clinic. Since this is not a major focus of this manuscript, we did not think it is useful to sequence all patients at this point. Furthermore, we used several other clinically approved markers to ensure that there are no classification errors.

6. Similarly to the above comment, FLI1 is not expressed in Ewing sarcoma cancer cells. Does the FLI1 protein detected correspond to the fusion protein? Could it be that FLI1 protein detection is due to non-cancer cells in the tumor tissue (i.e. hematopoietic and/or endothelial cells)?

FLI1 is not expressed as highly as other markers, such as NKX2.2, and the expression level also varies substantially across samples. Since we always selected the dissected regions under the microscope, we are confident that we are analyzing EWS regions. We cannot exclude having some contamination of immune cells and blood vessels, as we also report 80% purity. Following this reviewer's comment, we examined the

correlation between FLI1 expression level and tumor purity and found no correlation (Reviewer Figure 16). This result is added in Supplementary Figure 1h.

Reviewer Figure 16: Correlation between FLI1 expression and

7. Figure 1c . Age plot – dots do not need to be connected.

We removed the lines from this plot.

8. Figure 1d and Supplementary Figure 1f (Lines 106-111). Could it be that the variation in the expression of the selected ‘markers’ could be explained by differences in tumor tissue purity?

Following this reviewer’s question, and as indicated above, we know that the tissues are ~80% pure, according to the ESTIMATE results. Examination of the correlation between the expression level and the tumor purity score shows no correlation with purity, and in some cases very low correlation is found (Reviewer Figure 17). Overall, we think that the variable expression level that we see is not directly linked to purity but is also affected by other factors that impact their expression level. Notably, variable expression of these markers is also often seen by IHC staining, suggesting that there are various levels of regulation, which are not the focus of this research. These results are added in Supplementary Figure 1.

Reviewer Figure 9: Pearson correlation between EWS markers and tumor purity score. Each plot displays the Pearson correlation of one marker with the ESTIMATE tumor purity score. Each dot represents an EWS sample.

9. Lines 114, 121: ‘...post-treatment samples...’ Do the authors mean surgery samples (samples collected after neo-adjuvant chemo)?

Yes, post-treatment samples are surgical samples after neo-adjuvant chemotherapy. To clarify this point we changed the description to post-NACT samples throughout the manuscript.

10. Lines 125-126: Cluster 4 genes may be downregulated at surgery due to necrosis (please see comments above).

The surgical samples post-neoadjuvant chemotherapy were dissected from cancer cell regions and not from necrotic areas. As indicated above, we carefully isolated these regions, and the proteomic data shows the expression of the EWS markers and the calculated tumor purity.

11. The presence of the EWSR1::ETS fusion oncogene is only confirmed in 75% of the patients included in this study. Given that Ewing sarcoma tumors have been misclassified in the past, have the authors confirmed by other means that the rest 25% are Ewing sarcoma tumors? This is important especially for the clustering shown in Figure 3a. Are these cases evenly distributed across the three clusters?

All cases were confirmed by NKX2.2 staining. We validated all patients with missing clinical information, to be sure that there are no misclassifications (Reviewer Figure 18). We could not repeat genomic analyses due to ethical constraints.

Reviewer Figure 18: EWS validation by NKX2.2 validation. IHC images showing NKX2.2 expression in primary tumor samples (first two rows) and lung metastasis samples (rows 3-4) from patients with unknown EWSR1-FLI1 translocation status.

12. The authors have conducted a number of different analysis:

Figure 2: comparing samples from different time points

Figure 3: unsupervised clustering of primary tumors. This analysis could be of relevance/interesting, as it promises to identify disease biomarkers beyond presence/absence of metastasis at initial diagnosis. However, the authors do not follow this up. The number of samples is also low.

Figure 4: Supervised analysis of prognostic biomarkers

Figure 5: Multiplexed imaging revealing neutrophil involvement and lymphoid cell exclusion in metastatic tumors.

Each finding could be potentially interesting if followed up. Also, how these different analysis/findings could be linked?

We appreciate this reviewer's comment. Typically, in studies that present large amounts of data, it is impossible to follow-up on every result, and it might also make the manuscript more difficult to grasp. Here, we decided to focus on two main aspects, the ferroptosis and the immune characterization. In the revised manuscript, we added substantial validation to each part, to strengthen our claims.

1. **Ferroptosis-** *we validated the expression of selected proteins highly expressed in post-NACT samples (FTH and TFCR) using IHC staining. These results confirm our findings and show the expression in the cancer cell regions (Reviewer Figures 3 and 4, above). These results are also added to the revised manuscript in Figure 3.*

2. **Ferroptosis-** *we tested the impact of the ferroptosis inducer RSL3 (GPX4 inhibitor) on cell viability of the recently published EWS chemotherapy-resistance cell line model based on A673 cells. Initial analysis showed that A673-resistant cells are less sensitive to treatment with doxorubicin (as expected) and RSL3. Next, we treated parental and chemo-resistant cells with increasing concentrations of doxorubicin and increasing concentrations of RSL3. Modeling the effects of these drugs using SynergyFinder software showed strong synergism between these drugs, showing the importance of ferroptosis in mediating chemo-resistance (Reviewer Figure 7, above). These results are also added to the revised manuscript in Figure 3.*

3. **EWS immune landscape-** *We enlarged our immune marker panel to be more confident about the cell identity. To this end, we included CD4, CD8, MPO, HLA-DR beyond the previous staining of CD3, CD68, and CD15 (Reviewer Figure 6, above, and Manuscript Figures 4 and 6 and Supplementary Figures 4 and 6).*

We also changed a bit the flow of the manuscript to improve the connection between the prognostic markers and the tumor cluster.

Reviewer #4:

The authors here present a large proteomics dataset comprising primary, relapsed, and metastatic Ewing sarcoma samples. This is an important resource for the field and the data will be of high interest and useful for many investigators. A strength of this study is the large number of samples and inclusion of multiple samples per patient, and the novel contribution of clinically annotated, protein-level datasets for Ewing sarcoma patient samples that are scarce. While the correlative data and unbiased clustering are intriguing,

the lack of functional or mechanistic data make this paper useful as a resource but the purely correlative observations preclude mechanistic conclusions about progression.

We thank the reviewer for the positive evaluation of the importance of our work. This manuscript does not aim to provide a mechanism of EWS progression. Mechanistic studies in clinical samples are inherently challenging due to the inability to perturb human systems directly and the limited availability of late-stage tumor samples in clinical settings. One of the key strengths of our study lies in its ability to capture true human variability within a physiological and immunological context, which complements insights gained from in-vitro and in-vivo models. Nevertheless, we made substantial effort to add functional validations and strengthen the orthogonal validations by immunohistochemistry and immunofluorescence imaging.

- The enrichment of ferroptosis related genes post-therapy is interesting, as is the hypothesis that this would sensitize EWS cells to chemotherapy (Line 162). Functional experiments confirming this hypothesis that would make the conclusion more compelling, e.g. ferroptosis agonists/inhibitors, overexpression or knockdown of FTH1, +/- chemotherapy treatment. Likewise, the relationship between increased proteasomal protein expression and antigen presentation (Line 204) is intriguing but not shown functionally. But again, statements like Line 264 “Our findings suggest that inhibiting ferroptosis contributes to EWS chemo-resistance and suggests the potential impact of the proteasome and ubiquitin proteolysis on antigen presentation and DNA damage response (DDR)” seem preliminary if based solely on these data.

*Following this important comment, in the revised manuscript we added substantial validation. Beyond the IHC validations of ferroptosis-related proteins, FTH1 and TFRC, we also further investigated the functional importance of ferroptosis to chemotherapy response in EWS. To this end, we tested the impact of the ferroptosis inducer RSL3 (GPX4 inhibitor) on cell viability using the recently published EWS chemotherapy-resistance cell line model based on A673 cells. Initial analysis showed that A673-resistant cells are less sensitive to treatment with doxorubicin (as expected) and RSL3. Next, we treated parental and chemo-resistant cells with increasing concentrations of doxorubicin and increasing concentrations of RSL3. Modeling the effects of these drugs using SynergyFinder software showed strong synergism between these drugs, showing the importance of ferroptosis in mediating chemo-resistance. These results are added to the revised manuscript as **new Figure 3**. The results are also presented in **Reviewer Figure 4,5, and 7** (above).*

In addition, we toned down some of the speculative statements.

- Line 175: The authors state “Interestingly, these clusters reflected prognostic differences between clusters, with cluster 2 having a poorer prognosis than cluster 1 and cluster 3 (Fig. 3b). Kaplan-Meier survival analysis also agreed with other clinical parameters, indicating a significantly higher number of patients diagnosed with non-metastatic disease in cluster 3, and highest number of metastatic patients, and relapse-presenting patients in cluster 2 (Fisher enrichment analysis, p -val<0.05; Supplementary Fig. 3c-d”. They state that those findings are significant due to a P <0.1, (p =0.0768) logrank test, it is not clear why p <0.1 is used as opposed to the traditional p <0.05 and if this was determined pre- or post-testing.

We apologize for this error. Reanalysis showed a significant p-value. Furthermore, we added Cox regression analysis that shows significant association with survival (Reviewer Figure 19).

Reviewer Figure 19: Survival analyses of EWS consensus clusters. a) Kaplan-Meier survival curve for three clusters shows significant differences in 3 years DFS (p=0.028). Cluster 2 shows the lowest survival probability. Additionally, pairwise comparisons using the Log-Rank test with BH adjustment revealed cluster 1 vs. cluster 2 (p=0.035), cluster 1 vs. cluster 3 (p=0.927), and cluster 2 vs. cluster 3 (p=0.078). b) Cox regression results showing cluster 2 (HR: 3.9562, p=0.0437) and metastasis at diagnosis (HR: 4.5863, p=0.0192) significantly predict poorer survival. Pelvic disease and surgery are not significant.

-Likewise Supplementary Fig. 2D only one curve has a Bonferroni adjusted p value <0.05 for EFS, but all are referred to as significant. Overall survival should probably be shown as well, which is not significant.

Like the previous result, we omitted the word “significant” from the cases that have a higher corrected p-value. Of note, overall survival is largely impacted by the patient follow-up time. Newer patients with short follow-up often reduce the statistical significance of these analyses. Therefore, initially we did not include these analyses. Following this reviewer’s request, we added this result to **Supplementary Figure S3 (Reviewer Figure 20)**.

Reviewer Figure 20: Clinical association of ferroptosis-related genes using mRNA EWS DB. a) Kaplan Meier survival analysis of a publicly available mRNA expression analysis of EWS (Dirksen dataset, n = 85) shows shorter event-free survival for tumors with high GPX4, FTH1, and FTL expression. b) Kaplan Meier survival analysis of a publicly available mRNA expression analysis of EWS (Dirksen dataset, n = 85) shows shorter overall survival for tumors with high GPX4, FTH1, and FTL expression.

- Line 377: The authors say that extracellular matrix proteins were removed for data analysis, as well as blood proteins. Can they comment on whether this is a technical necessity based on the processing pipeline, as ECM proteins can be expressed by Ewing sarcoma cells and in fact may correlate with important features such as EWS: FLI1 “low” states?

We removed the ECM proteins and blood proteins as they often vary due to the technical variability in tissue dissection and often do not reflect the cancer cell phenotype itself. This allows normalization of the entire data to ensure more accurate sample ratio calculation. We agree that ECM proteins may have important roles, but in the case of these complex tissues, we cannot be sure of the source of ECM proteins.

- The frequent use of Y axes that do not begin at 0 (Fig 2D, 3F, S1D, S1E) sometimes over-inflates differences visually.

We disagree with this comment, since the intensity values do not start at zero, but the detection limit is around Log₂ intensity of ~9. Therefore, there is no meaning to starting the y axis at zero.

- Line 98: Tumor purity was measured using the ESTIMATE method, but those authors comment that its use in sarcomas is problematic: Yoshihara et al. 2013 <https://doi.org/10.1038/ncomms3612–> “The dependency of ESTIMATE on infiltrating stromal and immune cells resulted in some limitations, such as the inability to accurately infer tumour cellularity of hematopoietic or stromal tumours (for example, leukaemia, sarcoma and gastrointestinal stromal tumours) because of the high and tumour-intrinsic expression of stromal- or immune-related genes.” Can the authors quantify tumor purity using a more Ewings tailored approach?

We thank the reviewer for this comment. Unfortunately, we did not find an EWS-specific tool. Nevertheless, to address the reviewer’s comment, we compared ESTIMATE to other tools- ConsensusTME and xCell, and obtained similar results in the assessment of immune infiltration (Reviewer Figure 8, above).

Of note, if all these tools overestimate stromal components due to expression of stromal protein expression by the cancer cells, we can speculate that the actual tumor purity is higher than reported. These values are not a part of any downstream analysis.

- Line 384: for primary tumor consensus clustering the authors “averaged biological replicates”. Were these replicates taken from adjacent slides of the same region, or do they represent distinct geographic regions of the tumor? If so it would be interesting to see an analysis of the spatial heterogeneity of the primary tumor samples, as opposed to collapsing all regions together.

Our decision to average samples from the same tumors was intended to be more strict and avoid increasing the clusters with biological repeats of the same patients. This decision was also based on our observation that most repeats from the same patient highly correlated. Nevertheless, following the reviewer’s comment we investigated the consensus clustering including all tested primary tumor samples. Optimal clustering was achieved with four clusters, instead of three, but the main difference was that cluster 1 split into two clusters, and the rest remained almost identical (Reviewer Figure 21).

Reviewer Figure 21: Comparing non-averaged consensus clusters and averaged consensus clusters. Alluvial plots illustrate the correspondence between clusters identified using averaged (*Cluster_Avg*) and non-averaged (*Cluster_nonAvg*) of the primary EWS samples. Each flow represents the distribution of samples between clusters. Most samples maintain their original cluster, while Cluster 1 from the averaged data splits into two subclusters (1a and 1b) in the non-averaged data.

- Figure 5B: Were all CD15+ cells called as neutrophils during multiplex immunofluorescence? It appears that other cell types, like MDSCs, can sometimes express CD15. Additional markers for each cell type could increase confidence in cell type calling. i.e by staining an adjacent section with additional markers. How confident are the authors that CD15 staining is exclusive to neutrophils in these samples? Likewise CD68 can mark osteoclasts, if any remained in decalcified bone tumor samples.

We thank the reviewer for the important comment. In response, we expanded and refined our immune panel to ensure greater confidence in cell identity and specificity. Neutrophils are now labeled using both CD15 and MPO markers and are referred to as MPO-positive, with or without CD15 expression. Macrophages are stained with HLA-DR, MRC1, and CD68, while T-cell staining has been separated into CD8+ and CD4+ populations. This largely expanded the immune panel and increased the specificity. Due to a limited number of sections from previously stained tumor samples, we performed all staining on a new set of samples to ensure robust and consistent results. Using this sample panel, we found a significantly higher number of CD68+ macrophages in cluster 3 to be associated with patient prognosis, and a higher expression level in consensus cluster 3. We also show a higher number of MPO+CD15+ and MPO+ neutrophils in patients with poorer prognosis and in patients from consensus cluster 2.

*All these analyses recapitulated the trends of the results presented in the previous version but added the more accurate cell identity. The results are presented in the revised manuscript in **Figure 4, Figure 6 and Supplementary Figures 4 and 6**. The results are also presented here in **Reviewer Figure 6** (above).*

Reviewer #5:

We thank the reviewer for her/his contribution to our manuscript review.

Point-by-Point Response to Reviewers

Reviewer #1 (Remarks to the Author):

In this revised version of the manuscript, the authors have addressed some of the points previously raised but not others.

Points that have been addressed

1. Calcified vs decalcified samples - the authors have adequately addressed this issue
2. Details of sample prep and data analysis methodology - the authors have now provided details of sample prep and data analysis methods.

We thank the reviewer for the positive evaluation of our analyses.

There are some responses that are inadequate or problematic

1. Immune and stromal deconvolution - The authors provide information in their rebuttal that only half to two-thirds of the genes that comprise the immune deconvolution of ESTIMATE is represented in their dataset. They then go on to say that they are confident that the results are real because the use of xCell and ConsensusTME give similar results. I'm afraid this logic is flawed. Consistency of the different algorithms is meaningless if they all have poor coverage, i.e. they are all giving similarly incorrect deconvolution estimates especially since there is high overlap in the markers used in each of the algorithms. I also agree with Reviewer 4 and question the suitability of these algorithms for cells of stromal origin. The authors did attempt to validate some of the findings by IHC or multiplex IF, but this is problematic too, see next point.

We understand this reviewer's concerns but do not fully agree about the potential impact of these analyses on any of the results, since these analyses only provide a general description of the data and are not used for any downstream analysis. Nevertheless, we addressed this point in additional analytical way that use experimental data, not only computational. The major points are as follows:

- 1. Concordance across orthogonal methods:** This reviewer is concerned that the different analytical tools use the same gene sets, and therefore, their concordance does not reflect on their correctness. While we agree in principle with the reviewer, it is worth noting that the computational frameworks differ significantly. ESTIMATE utilizes ssGSEA-based enrichment of immune/stromal signatures; xCell is based on enrichment of 64 immune/stromal cell types; and ConsensusTME aggregates multiple tools and curated gene sets for consensus scoring. Importantly, the high concordance across these methodologies suggests that these estimates capture robust trends in immune and stromal content, supporting that our bulk EWS proteomic

samples are sufficiently pure for downstream analyses. Moreover, due to the high purity, we did not perform any data correction. Therefore, overall, these results are only intended to provide an overview of the data, but do not have any impact on the analyses.

2. Applicability of ESTIMATE and related tools to proteomic data: We acknowledge that ESTIMATE, ConsensusTME, and xCell were developed for transcriptomic data and require careful interpretation when applied to proteomic datasets. Nonetheless, their use has been extended to mass spectrometry-based studies **in sarcomas**, for instance, Tang, S., Wang, Y., Luo, R., *et al.* *Nat Commun* 2024 (<https://www.nature.com/articles/s41467-024-45306-y>) applied ESTIMATE to infer tumor purity and stromal/immune content in a large-scale pan-sarcoma proteomic cohort, supporting the relevance of such approaches in our study context.

3. Independent evaluation of tumor purity: To examine the ESTIMATE tumor purity derived from our proteomic data, we made use of the immunofluorescence staining data of some of the tumors in our cohort. We correlated the ESTIMATE scores with IHC staining in the same tumor ROIs used for mass spectrometry. Specifically, we assessed tumor cell content by quantifying NKX2.2⁺ cells. We observed a positive correlation between **NKX2.2⁺ cell fraction and estimated tumor purity** (Pearson $r = 0.41$, $p = 0.034$, 28 primary tumors available, **Reviewer Figure 1**). Some of the tumors show lower %NKX2.2⁺ cells, but these are outliers and may be related to the fact that these sections are $\sim 10 \mu\text{m}$ apart.

4. Discussion of the ESTIMATE limitations: To avoid overinterpretation, we have now revised the manuscript in the following ways: i) We explicitly state the limitations of partial gene-set coverage in the Methods section. ii) We emphasize that stromal and immune scores derived from deconvolution are supplementary findings, not used as the basis for core conclusions. iii) We clarify that these tools provide approximate estimates and do not accurately quantify purity or cell-type composition.

Reviewer Figure 1: Purity score correlation with NKX2.2. Scatter plot showing the fraction of NKX2.2⁺ cells (measured by IF in the MS-analyzed ROI) versus the tumor purity score inferred from ESTIMATE. A positive correlation (Pearson $r = 0.41$, $p = 0.03$) supports the use of the ESTIMATE method.

2. IHC or IF "validation" - I want to thank the authors for taking the revisions seriously by doing a range of different IHC and multiplexed IF validation. However, many of the findings are actually not significant and therefore not consistent with the mass spec data. What is more concerning is that the authors make several claims in the text that the findings are indeed validated even when

the quantitative differences are not statistically significant, e.g the PMSA quantification (Figure 5) and many of the immune cell quants (Figure 6 and several of the supp figures) are not statistically different between the groups.

We accept the reviewer's feedback that our validation experiments should be interpreted carefully. To clarify, some of the discrepancy between the proteomics data and the staining analysis is attributed to the fact that staining quantification was performed on the **entire stained slides** rather than the proteomic ROI. This was intended to provide a more comprehensive view of these proteins, expanding beyond the regional data. In addition, validations were performed on consecutive slides, suggesting that some discrepancies might occur. Lastly, staining could not be performed on all slides, but only on a subset of the samples, thus limiting the statistical power of these analyses.

To address the reviewer's concern, we repeated all multiplexed imaging analyses using only the same **regions of interest (ROIs) used for proteomics analysis, rather than whole tumor sections as in the initial version**. This ROI-matched strategy more accurately reflects the biological cases analyzed by mass spectrometry.

Macrophage analysis: Using one-way ANOVA followed by pairwise comparisons, we characterized CD68⁺ macrophage subpopulations. These region-wise analyses show the following results: CD68⁺HLA-DR⁺ macrophages, were more abundant in Cluster 3 compared to Cluster 2 (ANOVA $p = 0.05$, pairwise $p = 0.06$; **Reviewer Figure 2a-c**). All HLA-DR⁺ cells (irrespective of CD68 expression), showed significantly higher levels in Cluster 3 tumors than Cluster 2 (ANOVA $p = 0.04$, pairwise $p = 0.05$; **Reviewer Figure 2a-c**), further supporting the enhanced immunogenic profile of this subgroup. Other macrophage populations, including total CD68⁺, CD68⁺MRC1⁺, and CD68⁺MRC1⁻HLA-DR⁻ did not reach statistical significance, but showed a consistent trend of increased macrophages in Cluster 3. The results of CD68⁺, HLADR⁺, and CD68⁺HLADR⁺ are included in the revised manuscript in **Figure 6d**.

T cell analyses: CD4⁺ T-cell infiltration was elevated in Cluster 3 compared to Cluster 1 and 2 (ANOVA $p = 0.015$; pairwise comparisons $p = 0.0152$ and $p = 0.085$, respectively; **Reviewer Figure 2d**), suggesting a more immunologically active microenvironment.

Neutrophil analysis: Using one-way ANOVA followed by pairwise comparisons, we observed that MPO⁺ CD15⁺ neutrophils were significantly more abundant in Cluster 2 tumors compared to Clusters 1 and 3 (ANOVA $p = 0.013$, Pairwise $p = 0.01$, $p = 0.058$, respectively; **Reviewer Figure 2e**), supporting the proposed myeloid-enriched and immunosuppressive nature of this cluster.

These results are presented in the revised manuscript in **Figure 6d-i**.

Reviewer Figure 2: Multiplexed immunofluorescence validation of immune cell subsets in the proteomic ROIs and their association with EWS consensus clusters. a-c) The proportion of CD68+, HLA-DR+CD68+, and HLA-DR+ cells in three clusters were calculated as a percentage of the total cell population in the pROIs. Statistical analysis was performed using one-way ANOVA to assess overall differences, followed by post-hoc pairwise comparisons between clusters. Data are shown as mean \pm SEM. d) same for CD4+ T-cells. e) same for MPO+CD15+ neutrophils.

Beyond the analysis of the EWS clusters, we repeated the investigation of the associations with clinical outcome. **Similarly, we focused the repeated analysis on the proteomic ROIs rather than the entire slides.** Clinical outcome correlations revealed that CD68⁺ macrophages and their subpopulations were significantly associated with improved prognosis. Total CD68⁺ cell density was significantly higher in non-relapsing tumors and in tumors from patients with favorable 3-year disease free survival (Welch's t-test, $p = 0.015$, $p = 0.02$; **Reviewer Figure 3a-b**). When evaluating all HLA-DR⁺ cells, irrespective of CD68 status, we observed a significant association with both relapse status and 3-year DFS (Welch's t-test, $p = 0.054$, $p = 0.001$; **Reviewer Figure 3c-d**). The CD68⁺HLA-DR⁺ macrophage population was also significantly higher in tumors with 3-year DFS, and the sample trend was also seen for the association with relapse. The CD68⁺MRC1⁺ macrophages population was very small and showed a similar trend (**Reviewer Figure 3e-h**). These associations provide independent, orthogonal support for the immune-related survival signatures identified in our proteomic dataset.

These updated analyses closely mirrored the trends observed in our prior whole-slide analyses but provide a better validation of the proteomic results. We speculate that the lower statistical significance in the whole slide analysis resulted from the limited cohort used for immunostaining. But the more focused analyses provide cleaner and statistically significant signal.

Reviewer Figure 3: Multiplexed immunofluorescence validation of immune cell subsets in the proteomic ROIs and their association with clinical outcomes. a-b) Analysis of total CD68⁺ macrophages in patients with and without relapse, and with and without 3-year disease-free survival (3Y DFS). Welch's t-test; mean \pm SEM. c-d) Same for all HLA-DR⁺ cells. e-f) Same for CD68+HLA-DR⁺ macrophages. g-h) Same for CD68+MRC1⁺ macrophages.

To further assess PSMA3 expression, we quantified it across the proteomics ROIs and the NKX2.2+ regions, marking EWS tumor cells. Across all approaches, we observed a trend toward higher PSMA3 intensity in Cluster 3 tumors compared with Cluster 2, consistent with the proteomic findings. However, statistical significance was not achieved in any individual comparison, possibly due to the limited sample size. We could not validate other 20S proteasomal subunits due to the lack of available antibodies. Given the limited statistical significance of these analyses, we removed these results from the revised manuscript.

3. Functional validation - In response to my and other reviewers' comments about the lack of functional work on ferroptosis and resistance, the authors have done some drug testing in a paired cell line experiment to demonstrate that combination with GPX4 inhibitor is able to overcome drug resistance. These experiments are unfortunately really superficial and lacking in depth. They raise more questions than answers, for instance, does the cell line pair have similar proteomic changes in the ferroptosis pathways as the tissues, what are GPX4 levels in the cell lines, biochemical analysis that ferroptosis has been inhibited was not done, and doing it in one cell line and one drug is really insufficient to draw conclusions in contemporary cancer research.

We regret that the reviewer was not satisfied with our previous functional analyses. The study focused on the proteomics of the clinical cohort, and typically such studies do not include, in addition, very deep investigation of potential regulatory pathways. Furthermore, recapitulating all the ferroptosis-related proteomic results in cultured cells should not be expected as the conditions are markedly different (iron and oxygen levels, growth in culture, etc.), and cell lines never fully mimic tumor samples. Nevertheless, we wished to address the reviewer comments and added the following functional analyses related to the ferroptosis pathway: (i) We included an additional novel chemo-resistant EWS model from the RDES cell line, recently established in our lab (not yet published). (ii) We analyzed chemoresistance and ferroptosis sensitivity in 3D spheroids. (iii) We extended the analyses to include two chemotherapies and two ferroptosis inducers, on two cell line model (four cell lines in total). (iv) We quantified expression of ferroptosis related proteins. (v) We quantified non-heme iron levels in the tissue samples. Unfortunately, we did not succeed in performing BODIPY analysis of ferroptosis, since we obtained highly variable signal depending on the exact treatment duration. Since we moved the functional analyses to 3D models, it is difficult to match the 2D (BODIPY) and 3D (sensitivity) conditions. We realized that these experiments would require extensive optimization, which is beyond the scope of this study.

Our key findings are summarized below:

1. Establishment of a novel RDES chemo-resistance model: To model acquired chemoresistance, we adopted the previously published protocol by Scarborough et al. (2020), which generated a chemotherapy-resistant A673 line. RDES cells were treated with five full cycles of vincristine-doxorubicin-cyclophosphamide (VDC) and etoposide-cyclophosphamide (EC). This procedure models the standard of care given to patients with EWS, which consists of VDC and IE (ifosfamide-etoposide) cycles. Since ifosfamide requires metabolic activation, we substituted ifosfamide with cyclophosphamide (as in the original protocol by Scarborough et al.), since these are analog compounds. The cells were treated at concentrations of IC30/IC40 for each drug in the combination VDC/EC for four days. Recovery time ranged from five to ten days. Control clones

were maintained in the vehicle control medium. The detailed description of the model is included in the Methods section.

2. Establishment of 3D spheroid models based on A673 and RDES cell lines: Following the reviewers' concerns about the physiological relevance of 2D cultures, we established 3D spheroid models of A673 and RDES, including parental and chemo-resistant cells. We followed previously published protocols by Ceranski et al. (2025) and Li et al. (2021) with adaptations to our systems. Spheroids were formed in HPLM media in round-bottom ultra-low attachment microplates and were allowed to grow for 48 hours prior to drug exposure. Cells were treated with etoposide or doxorubicin, and after 60 hours, spheroid viability was quantified using CellTiter-Glo® 3D according to the manufacturer's instructions. Luminescence values were background-subtracted and normalized to vehicle control to calculate percent viability. Dose-response curves and IC₅₀ values were calculated in GraphPad Prism using nonlinear regression. We observed significant shifts in IC₅₀ values between parental and resistant cell lines. In both models, resistance indexes (IC₅₀ resistant/IC₅₀ parental) were higher than 3.2 for both etoposide and doxorubicin (**Reviewer Figure 4**). These results confirm the successful establishment of a chemo-resistant phenotype and the 3D spheroid models. These results are presented in the revised manuscript in **Figure 3g**.

Reviewer Figure 4: Chemotherapy resistance profiling in 3D spheroid models of Ewing sarcoma. Dose-response curves for doxorubicin and etoposide in parental and resistant A673 and RDES spheroids. The x-axis shows log₁₀ drug concentration; the y-axis shows cell viability relative to untreated controls. IC₅₀ values demonstrate a greater than 3.27-fold increase in resistance compared to parental cells, confirming the successful establishment of resistant models. Spheroid viability was quantified using CellTiter-Glo® 3D, and IC₅₀ values were calculated for each condition using nonlinear regression in GraphPad Prism.

3. Combination treatments of chemotherapy and ferroptosis inducers: To strengthen the functional validation and address the reviewers' concerns, we expanded our testing to include **two chemotherapeutic agents (doxorubicin and etoposide)** and **two ferroptosis inducers (RSL3 and erastin)**. We applied fixed concentrations of RSL3 (20nm for RDES, 30nm for A673) or erastin (0.33µm for RDES, 0.4µm for A673), which were determined in preliminary experiments, in combination with a dose range of chemotherapy to examine the potential of combined therapy in overcoming chemotherapy resistance. The spheres were grown for 48 hours, and the combination treatment was applied for additional 60 hours. We determined the IC₅₀ values of

each chemotherapeutic drug and performed a sum of squares F test to assess the statistical differences in IC_{50} values between chemotherapy alone and each chemotherapy plus ferroptosis inducer combination. For A673 cells, both parental and resistant cells showed a significant decrease in **doxorubicin IC_{50} when combined with either RSL3 or erastin**. For RDES cells, both parental and resistant cells demonstrated a significant decrease in **etoposide IC_{50} when combined with either RSL3 or erastin**. Additionally, RDES-resistant cells showed a significant decrease in **doxorubicin IC_{50} when combined with erastin**. These reductions indicate a potentiation of chemotherapy efficacy by ferroptosis inducers, particularly in the resistant cell lines (Reviewer Figure 5a, Figure 3h).

Reviewer Figure 5: Ferroptosis-based combination therapies reduce chemotherapy resistance in RDES and A673 cells. a) Bar plots show chemotherapy IC_{50} values in A673 and RDES spheroids treated with doxorubicin or etoposide alone, or in combination with RSL3 or erastin (fixed concentrations: 20/30 nM or 0.33/0.4 μ M for RDES/A673, respectively). The sum-of-squares F test determined statistical significance; significant decreases in IC_{50} indicate enhanced efficacy. Data are shown for both parental and resistant lines. b+c) Representative images of RDES and A673 parental and resistant spheroids treated with chemotherapy alone or in combination with RSL3 or erastin.

To further strengthen the CellTiter-Glo® 3D findings, we included representative spheroid images for both A673 and RDES parental and resistant cells. Images were taken for DMSO control and for two different concentrations of chemotherapy, either alone or in combination with RSL3 or erastin. This allowed for a direct visual comparison of spheroids treated with the various drug combinations in both models (Reviewer Figure 5b). Consistent with the IC_{50} results,

chemotherapy combined with a ferroptosis inducer produced visibly smaller spheroids compared to chemotherapy alone. The effects were more pronounced in RDES spheroids and were added to **Fig. 3i**. Representative images from A673 spheroids are provided in the **Supplementary Fig. 4a**. These expanded experiments, involving multiple agents and combinations, further reinforce our hypothesis that inducing ferroptosis may increase chemotherapy response.

4. Proteomic analysis of parental and chemo-resistant RDES cells showed significant differences in several ferroptosis-related proteins (from FerrDB). **Reviewer Figure 6** shows the intensities of four proteins that present significant differences between the parental and chemo-resistant cells. The resistant cells expressed lower levels of SLC7A11, the cystine/glutamate antiporter critical for glutathione biosynthesis, thioredoxin reductase (TXNRD1), and peroxiredoxin (PRDX1). All these proteins harbor critical antioxidant activities. The transferrin receptor (TFRC) level increases in the resistant RDES cells, similar to our observation in post-NACT clinical samples, suggesting that the cells may have higher potential for iron uptake. GPX4 did not show significant differences between cells. We speculate that in the 2D in-vitro conditions, the impact of growth conditions with low iron and high oxygen, and the relatively short durations of these treatments, might not accurately represent the real human tumor conditions. To fully understand the relationship between ferroptosis induction and the protein levels of each regulator, it would be necessary to run extensive research of multiple growth conditions (with fully optimized iron and oxygen concentrations), and study multiple experimental durations to link the proteomic changes, lipid peroxidation, and cell death. We believe that full mechanistic elucidation is outside the scope of this manuscript, and the in-depth investigation of the cell line models does not sufficiently contribute to the understanding of the ferroptosis involvement. We therefore did not include these results in the revised manuscript.

Reviewer Figure 6: Expression of ferroptosis related proteins. We performed proteomic analysis of RDES cells (parental and resistant cells) and compared the levels of ferroptosis-related proteins. We crossed the proteomics data with FerrDb and selected the proteins that are directly associated with the iron metabolism and the redox state. Selected results are presented.

5. Quantification of non-heme iron levels in primary and post-NACT samples

To examine the nature and localization of iron accumulation, we performed Perls' Prussian Blue staining, a well-established histochemical technique that specifically detects non-heme ferric iron (Fe^{3+}), typically stored within ferritin and hemosiderin complexes. Importantly, this method does not detect heme-bound iron, such as that found in hemoglobin, and thus does not stain red blood cells. As a result, Perls' staining is particularly effective at distinguishing true intracellular iron storage from hemorrhage-related extracellular iron or heme-bound iron derived from red blood cells. Quantitative analysis of Perls' staining within the proteomic ROIs revealed a significantly higher percentage of positive area in post-NACT tumors compared to primary tumors ($p = 0.0127$).

Reviewer Figure 7: Quantification of non-heme iron levels in primary and post-NACT samples. a) Representative IHC images of Pearl Prussian blue in primary and post-NACT samples. Scale bar, 50 μm . b) Perls' Prussian blue quantification in primary and post-NACT samples. Error bars represent the SEM values. A Welch's T-test was used, and quantification was based on the proteomic ROI.

To support these observations, we have now included representative Perls' Prussian Blue staining images from matched primary and post-NACT tumors (**Reviewer Fig. 7**), with examples chosen to reflect staining intensities near the cohort median rather than extreme phenotypes. While we cannot completely exclude some contribution of intratumoral hemorrhage in post-treatment samples, we note that the upregulation of both FTH1 and TFRC at the protein level was observed consistently across the post-NACT cohort in our MS data and validated by quantitative IHC staining. Together with Perls' staining, these findings strengthen the conclusion that iron regulatory pathways are actively modulated following chemotherapy.

4. Removal of ECM proteins - In providing more detail to the data analysis section, the authors revealed that ECM proteins were removed before analysis. In response to reviewer 4, they said "they often vary due to the technical variability in tissue dissection and often do not reflect the cancer cell phenotype itself. This allows normalization of the entire data to ensure more accurate sample ratio calculation". This is very unusual, especially given that the authors have enriched for

80% tumour cell content. It is not the norm to remove ECM proteins before analysis, can the authors provide some data to show what happens to the data if you include ECM proteins? Could the ECM proteins be differentially impacted by decalcification. It has been shown in soft tissue sarcomas that ECM proteins are consistent and representative of histological subtypes and there is similar evidence in proteomic data from other bone tumours like chordoma as well.

We thank the reviewer for raising this point. The ECM proteins were removed prior to initial normalization because we aim to minimize artifacts introduced by bone and soft tissue ECM and focus on cancer cell-intrinsic biology. We referred to the ECM as a potential artifact because it is largely affected by the technical accuracy of the dissection of the tissue, rather than the inherent extent of ECM secreted by the cancer cells. In addition, we cannot know which cells deposit the ECM, and any claim about the ECM protein expression in these tissues would be inaccurate. Of note, it is very common to eliminate groups of genes/proteins in omics data. For example, removal of various outliers or constant proteins is common in RNA gene expression analyses, and removal of potential contaminants, such as keratins and blood proteins is common in proteomics research. Importantly, the raw, unprocessed data is freely available for additional future analyses, in case readers are interested specifically in the involvement of ECM in EWS. Nevertheless, we addressed this reviewer's request and examined the impact of the removal of ECM proteins on the overall results. We performed several comparisons between the datasets with and without ECM.

1. Correlation of sample-sample distances: We calculated the Euclidean distances between all possible sample pairs (162 samples, resulting in $N(N-1)/2 = 13,041$ pairs) for both the ECM-included and ECM-filtered datasets. The Pearson correlation between the two distance matrices was **R = 0.995**, indicating that sample relationships were nearly identical (**Reviewer Figure 8a**). While most distances were nearly identical, a small number of sample pairs showed minor deviations. This analysis confirms that the removal of ECM proteins did not alter the global data structure and introduced minor, localized changes.

2. Clustering consistency - Adjusted Rand Index (ARI): We performed k-means clustering with **k = 3** on both datasets and compared the resulting assignments using the Adjusted Rand Index (ARI). Comparing the resulting cluster assignments using the Adjusted Rand Index revealed a high degree of similarity with an ARI of **0.89**. This concordance is demonstrated by the fact that out of **162 samples**, only **6 samples (3.7%)** were assigned to different clusters after the removal of ECM proteins (**Reviewer Figure 8b**). This confirms that the removal of ECM proteins did not significantly alter the overall grouping pattern of the data.

3. ANOVA comparison of key protein groups

We repeated the ANOVA analysis of all sample stages (FDR < 0.05) for both datasets to assess whether ECM removal affected the detection of biologically relevant differences. The ECM-included dataset yielded 1,071 significantly changing proteins (145 unique), and the ECM-filtered dataset yielded 1,013 significantly changing proteins (88 unique). Importantly, **924 (91.3%) significantly changing proteins were shared between both groups**, which represents the majority of significant findings (**Reviewer Figure 8c**). These results indicate strong concordance between the datasets with and without ECM. Specifically, ferroptosis inhibitors, which were found to be higher in post-NACT samples were not altered by the filtering process.

5. PCA pattern comparison: PCA plots (**Reviewer Figure 8d-e**) show that the spatial arrangement of samples is preserved between datasets. This visual consistency supports the quantitative results from the correlation and ARI analyses.

Overall, our analyses show minimal impact of ECM removal on the conclusions. Since ECM protein removal aimed to minimize technical variability related to matrix contamination in the macrodissected regions, we did not change these pre-processing steps.

Reviewer Figure 8: Evaluating the impact of ECM removal. a) Scatterplot showing pairwise Euclidean distances between all sample pairs in the ECM-included dataset (x-axis) and the ECM-removed dataset (y-axis). Each point represents a sample pair ($N \times N-1$). The high Pearson correlation coefficient ($R = 0.995$) indicates strong preservation of the sample similarity structure after ECM protein removal. b) K-means ($k=3$) clustering consistency plot between ECM included and ECM-filtered proteome datasets. Each point represents a single sample, with its position determined by its cluster assignment in the ECM included proteomics data (x-axis) and the ECM-filtered data (y-axis). The color of each point corresponds to its cluster in the whole proteomic dataset. The Adjusted Rand Index of 0.89 indicates a high degree of concordance, showing that the vast majority of samples retained their cluster assignment. The few points located off the diagonal represent the minor cluster changes resulting from ECM removal. c) A Venn diagram comparing the significantly changing proteins ($FDR < 0.05$) from the ANOVA analysis of the whole proteome dataset (1,071 proteins) and the ECM-filtered dataset (1,013 proteins). d+e) Principal Component Analysis (PCA) plots of the whole proteome and ECM-filtered datasets, respectively. The overall distribution of the sample points is highly similar in both plots, demonstrating that ECM removal did not cause a significant change in the global structure of the data.

5. Specificity of NKX2.2 - in response to reviewer 3, the authors said that for those cases without fusion status, staining for NKX2.2 was done and it was positive for all cases. Unfortunately this stain is not specific for EWS as shown by Jason Hornick and Chris Fletcher (see Hung et al., *Modern Pathology* 2016). Therefore it is still uncertain if these cases are truly EWS.

We appreciate the reviewer's concern regarding NKX2.2's diagnostic specificity. We would like to clarify that in our study, all diagnoses were confirmed by an experienced sarcoma pathologist based on a comprehensive assessment of morphology, clinical context, and immunohistochemistry. While we acknowledge that NKX2.2 is not entirely specific, it remains among the most sensitive and widely used immunohistochemical markers for Ewing sarcoma, particularly when combined with other diagnostic criteria. NKX2.2 was consistently positive in all fusion-unknown cases, in line with the expected EWS phenotype.

The pathological diagnosis in our cohort (accepted by leaders in the field) was established based on classical morphological features of Ewing sarcoma, namely small, round blue cell morphology with scant cytoplasm and uniform nuclei, in combination with a characteristic immunohistochemical profile. Tumors showed strong membranous CD99 expression and nuclear NKX2.2 staining. In older cases, FLI1 was also used as an additional supportive marker. To ensure specificity, we routinely performed negative staining for non-Ewing markers, including LCA, TdT, Pan-Cytokeratin (PanCK), S100, and Desmin, which were consistently negative, helping to exclude alternative diagnoses. This immunohistochemistry profile, interpreted within the clinical and anatomical context of each patient, forms the diagnostic foundation for EWS in contemporary practice.

Importantly, molecular testing is often not routinely performed in cases with classic histopathological features and strong supportive IHC markers in current clinical practice, including in major sarcoma centers. This approach aligns with accepted diagnostic standards when the

morphology and immunohistochemistry profile are typical and there are no unusual anatomical, clinical, or histological features to suggest alternative diagnoses.

We have clarified this point in the revised Methods of the manuscript.

6. I still cannot find an explicit limitation section in the discussion for this study.

We added an explicit limitations paragraph in the revised manuscript.

“While our study provides novel insights into Ewing sarcoma biology through high-resolution proteomics and spatial profiling, several limitations should be acknowledged. First, although the cohort is relatively large and includes primary, relapse, and post-NACT samples, it was derived from a single institution. Broader, multi-center studies with larger and more diverse patient populations will be required to validate and generalize our findings. Second, functional validation of the implicated pathways, particularly those related to mitochondrial metabolism, proteasome activity, and DNA damage response, is needed to confirm mechanistic relevance. In addition, while our multiplexed imaging analysis provides spatial resolution of the tumor immune microenvironment, the functional assessment of immune cell states (e.g., T-cell exhaustion, macrophage polarization) was not performed. Lastly, despite the enrichment of cancer cells in the proteomic ROIs, the analyses are still considered bulk measurements and include some non-cancer cells. The use of the ESTIMATE algorithm provided a rough estimate of tumor purity. However, this was based on a partial gene-set coverage, and in general, this tool was not generated specifically for this cancer type. We therefore refer to these results as general assessments of purity, and we did not use them for downstream data correction.”

Minor issues

1. The authors in their intro and discussion highlight EWS as a paediatric disease. However quite a number of their patients are in the AYA age range, so it may be good to reflect this in the text.

Indeed, several patients in our cohort fall within the adolescent and young adult (AYA) age range (above 18 and below 24 years). We have revised the manuscript to acknowledge that Ewing sarcoma predominantly affects children and adolescents but also frequently occurs in young adults and may present unique clinical challenges in this population. This has been corrected in the Introduction.

2. The authors should provide a table summarising the clinicopath features of the cohort. It is very difficult to ascertain cohort level features from the figures and the supplemental table of individual cases.

We appreciate the reviewer’s suggestion and have added a summary table detailing the key clinicopathological features of the cohort, including age range, survival, relapse, stage at

diagnosis, and response to therapy. This has been included as Supplementary Table 1c and referenced in the revised *Results* sections.

3. Who is the pathologist for this study? Is he/she a named author. I only ask this because the diagnosis as well as the assessment of necrosis should be done by a expert sarcoma pathologist.

The pathologist involved in this study is Dr. Osant Sher, a named author of the paper, who is a certified pathologist with specific expertise in sarcomas. She assessed tumor necrosis on available primary diagnostic slides and was involved in the diagnostic workup of the patients in the cohort. Her evaluations were integral to both the clinical diagnosis and the histopathological annotation of the samples used in the proteomic analysis.

Reviewer #2 (Remarks to the Author):

The revised manuscript by Gordon et al. entitled ‘Proteomic landscapes of Ewing sarcoma unravel immunological regulation of tumor progression’ reports on a relatively large series of Ewing sarcoma (EwS) tumor samples that have been subjected to mass-spectrometry based proteomics analyses. While the manuscript has been slightly improved, it remains a rather exploratory analysis that largely stays at the phenotypic description as the authors now admit in the rebuttal letter, and some of the provided new data are still not convincing as outlined below:

1) The new data on the so-called “A673-resistant” cell lines is problematic. First, A673 is a complicated model that shows a relatively high degree of genomic drift (Kasan M et al. 2025 Nat Commun) and that has an unusual second driver mutation (BRAF V600E). This model is alone not suitable to address the research question. Second, the shown “resistance” is defined by a moderate increase (not even a doubling) of the relative IC50 towards doxorubicin. The “resistant” cell lines still respond at lower nanomolar concentrations to this drug. Thus, the used cell line is not suitable for modeling chemoresistance as they are still by and large sensitive toward this drug. The authors should use more suitable and truly resistant models grown in 3D and also include other chemotherapeutics used in EwS. The hitherto presented data are not convincing to support any mechanistic conclusion.

We thank the reviewer for the thorough evaluation of our work but regret that the validations were found insufficient. We wish to emphasize that the intention was not to initiate an independent research study, but rather check the hypotheses based on the solid clinical proteomics data. Nevertheless, in the revised version we made substantial efforts and addressed all the reviewer’s requests: (i) We included an additional novel chemo-resistant EWS model from

the **RDES** cell line, recently established in our lab (not yet published). (ii) We analyzed chemoresistance and ferroptosis sensitivity in **3D spheroids**. (iii) we extended the analyses to include **two chemotherapies and two ferroptosis inducers**, on two cell line model (four cell lines).

Our key findings are summarized below:

1. Establishment of a novel RDES chemo-resistance model: To model acquired chemoresistance, we adopted the previously published protocol by Scarborough et al. (2020), which generated a chemotherapy-resistant A673 line. RDES cells were treated with five full cycles of vincristine-doxorubicin-cyclophosphamide (VDC) and etoposide-cyclophosphamide (EC). This procedure models the standard of care given to patients with EWS, which consists of VDC and IE (ifosfamide-etoposide) cycles. Since ifosfamide requires metabolic activation, we substituted ifosfamide with cyclophosphamide (as in the original protocol by Scarborough et al.), since these are analog compounds. The cells were treated at concentrations of IC₃₀/IC₄₀ for each drug in the combination VDC/EC for four days. Recovery time ranged from five to ten days. Control clones were maintained in the vehicle control medium. The detailed description of the model is included in the Methods section.

2. Establishment of 3D spheroid models based on A673 and RDES cell lines: Following the reviewers' concerns about the physiological relevance of 2D cultures, we established 3D spheroid models of A673 and RDES, including parental and chemo-resistant cells. Spheroids were formed in HPLM media in round-bottom ultra-low attachment microplates and were allowed to grow for 48 hours prior to drug exposure. Cells were treated with etoposide or doxorubicin, and after 60 hours, spheroid viability was quantified using CellTiter-Glo® 3D according to the manufacturer's instructions. Luminescence values were background-subtracted and normalized to vehicle control to calculate percent viability. Dose-response curves and IC₅₀ values were calculated in GraphPad Prism using nonlinear regression. We observed significant shifts in IC₅₀ values between parental and resistant cell lines. In both models, Resistance indexes (IC₅₀ resistant/IC₅₀ parental) were higher than 3.2 for both etoposide and doxorubicin (**Reviewer Figure 9**). These results confirm the successful establishment of a chemo-resistant phenotype and the 3D spheroid models. These results are presented in the revised manuscript in **Figure 3g**.

—

Reviewer Figure 9: Chemotherapy resistance profiling in 3D spheroid models of Ewing sarcoma. Dose-response curves for doxorubicin and etoposide in parental and resistant A673 and RDES spheroids. The x-axis shows \log_{10} drug concentration; the y-axis shows cell viability relative to untreated controls. Spheroid viability was quantified using CellTiter-Glo® 3D, and IC₅₀ values were calculated for each condition using GraphPad Prism.

3. Combination treatments of chemotherapy and ferroptosis inducers: To strengthen the functional validation and address the reviewers' concerns, we expanded our testing to include **two chemotherapeutic agents (doxorubicin and etoposide)** and **two ferroptosis inducers (RSL3 and erastin)**. We applied fixed concentrations of RSL3 (20nm for RDES, 30nm for A673) or erastin (0.33µm for RDES, 0.4µm for A673), which were determined in preliminary experiments, in combination with a dose range of chemotherapy to examine the potential of combined therapy in overcoming chemotherapy resistance. The spheres were grown for 48 hours, and the combination treatment was applied for additional 60 hours. We determined the IC₅₀ values of each chemotherapeutic drug and performed a sum of squares F test to assess the statistical differences in IC₅₀ values between chemotherapy alone and each chemotherapy plus ferroptosis inducer combination. For A673 cells, both parental and resistant cells showed a significant decrease in **doxorubicin IC₅₀ when combined with either RSL3 or erastin**. For RDES cells, both parental and resistant cells demonstrated a significant decrease in **etoposide IC₅₀ when combined with either RSL3 or erastin**. Additionally, RDES-resistant cells showed a significant decrease in **doxorubicin IC₅₀ when combined with erastin**. These reductions indicate a potentiation of chemotherapy efficacy by ferroptosis inducers, particularly in the resistant cell lines (**Reviewer Figure 10a, new Figure 3h**).

4. To further strengthen the CellTiter-Glo® 3D findings, we included representative spheroid images for both A673 and RDES parental and resistant cells. Images were taken for DMSO control and for two different concentrations of chemotherapy, either alone or in combination with RSL3 or erastin. This allowed for a direct visual comparison of spheroids treated with the various drug combinations in both models (**Reviewer Figure 10b**). Consistent with the IC₅₀ results, chemotherapy combined with a ferroptosis inducer produced visibly smaller spheroids compared to chemotherapy alone. The effects were more pronounced in RDES spheroids and were added to **new Fig. 3i**. Representative images from A673 spheroids are provided in the **Supplementary**

Fig. 4a. These expanded experiments, involving multiple agents and combinations further reinforce our hypothesis that inducing ferroptosis may increase chemotherapy response. We acknowledge that there are differences between the cell lines in terms of their response to RSL3 and erastin, but this is expected given the differences in the mechanisms of action of the two drugs and the expected difference in protein expression levels of various proteins. As also indicated by the reviewer, the growth conditions in 2D, 3D or the in-vivo environment differ substantially in terms of iron and oxygen availability. Therefore, differences in these pathways are to be expected.

These expanded experiments, involving multiple agents and combinations, further reinforce our hypothesis that inducing ferroptosis may increase chemotherapy response in chemo-resistant cells.

Reviewer Figure 10: Ferroptosis-based combination therapies reduce chemotherapy resistance in RDES and A673 cells. a) Bar plots show chemotherapy IC_{50} values in A673 and RDES spheroids treated with doxorubicin or etoposide alone, or in combination with RSL3 or erastin (fixed concentrations: 20/30 nM or 0.33/0.4 μ M for RDES/A673, respectively). The sum-of-squares F test determined statistical significance; significant decreases in IC_{50} indicate enhanced efficacy. Data are shown for both parental and resistant lines. b+c) Representative images of RDES and A673 parental and resistant spheroids treated with chemotherapy alone or in combination with RSL3 or erastin.

2) The authors now provide data on stromal contamination (ESTIMATE). Plausibly, the number of identified immune cells correlates with the estimated amount of stroma in each sample. How can the authors exclude a selection bias, that is that the samples taken from the EwS tumors that

underwent MS analysis did not have different amounts of stroma per se? How representative is each sample for a given tumor?

In our study, multiple regions were selected for many of the tumors, based on comprehensive histopathological evaluation of the H&E slides, covering different parts of the viable tumor tissue. This strategy was intentionally chosen to minimize bias from any single ROI and provide a composite representation of the tumor proteome. We believe this approach improves the representation of the samples and reduces the likelihood of regional sampling bias. Generally, all ROIs were selected to include highly pure cancer cell populations and exclude as much as possible ECM proteins and immune cells. In this way, we aimed to focus on the cancer cell phenotype. Of note, if we were to select random regions, we would likely see mostly ECM and blood proteins, resulting in low proteome coverage and providing few insights into the cancer cells themselves. The ESTIMATE analysis was only aimed at examining the predicted tumor purity to ensure that we work with sufficiently pure cancer regions, and was not intended to reflect on the tumor cellularity/stromal components altogether. Examination of the correlation between the % alphaSMA in the proteomic ROI and in the whole slide showed a strong correlation (Pearson $r = 0.79$, $p = 5.4e-005$; **Reviewer Figure 11**), but that the stroma in the ROI is much lower.

Reviewer Figure 11: Evaluation of stromal representation in the proteomic ROI. Scatter plot demonstrating the correlation between α SMA⁺ cell density (as a surrogate for stromal content) in multiplex-imaged MS-analyzed ROIs and corresponding whole-slide stromal scores. Each dot represents a tumor.

3) The authors claim that the highly significant correlation between neutrophils and necrosis is merely driven by one outlier sample. How would the statistics look like if this outlier sample would be removed (Reviewer Figure 10)? If it would stay significant, then the speculations on the possible biological relevance of neutrophils in the paper should be removed as this is likely driven by just catching up more necrotic tissue in the samples. From the new histological images (Reviewer Figures 3-5) one can actually see a variable amount of necrosis.

In response to the reviewer's question, we reanalyzed the correlation between neutrophil abundance and histologically assessed tumor necrosis after removing one outlier sample that exhibited 55% necrosis, substantially higher than all other samples in the cohort. After excluding this sample, the correlation was no longer statistically significant ($r=0.29$, p -value= 0.21), indicating that this outlier largely drives the initial association (**Reviewer Figure 12a**).

Neutrophil extracellular traps (NETs) have been implicated in modulating the immune microenvironment, promoting tumor progression, and contributing to therapy resistance. In our previously published work (Shukrun et al., *Cancer Sci.*, 2024; doi:10.1111/cas.15992), we demonstrated that NET formation in EWS tumors is associated with poor prognosis. To further examine the potential association between neutrophils and necrosis, we revisited a subset of 13 EWS tumor samples in which immunofluorescence staining for NETs was performed using H3 and neutrophil elastase (NE). We quantified both neutrophil density and the percentage of neutrophils undergoing NETosis (H3⁺NE⁺ co-localization among all neutrophils) and correlated these with histologically assessed necrosis scores (**Reviewer Figure 12b-c**). We found no statistically significant correlation between neutrophil density and necrosis ($r=0.35$, p -value=0.126) or NETosis (%) and necrosis ($r=0.325$, p -value=0.28).

Together, these findings suggest that the neutrophils' presence and role in patient prognosis is not merely a result of tumor necrosis.

Reviewer Figure 12: Neutrophils and necrosis correlation. a) Scatter plot shows the correlation between neutrophil abundance (proteomic data) and histologically assessed tumor necrosis after exclusion of one outlier sample with 55% necrosis. b+c) Scatter plots show correlations between neutrophil density (center) and percentage of NETosis (right) with tumor necrosis scores in a subset of 13 tumors. No significant associations were observed.

4) The histological images showing FTH1 and TFRC immunoreactivity show very dramatic staining intensities that do not correspond to the moderately different mean or median (not clear from the figure and legends) H-scores. The authors should show images that are representative for the medians and not extreme phenotypes. Can the authors show a simple iron stain. It is plausible that upregulation of both proteins is a mere epiphenomenon of hemorrhage and bleeding into the tumor tissue post-NACT. Also, both immunostains show huge background smears.

We thank the reviewer for the valuable feedback regarding the interpretation and presentation of FTH1 and TFRC immunostaining data. We fully agree that representative images should ideally reflect the median H-scores. However, we specifically selected the current images because they

represent unique matched primary and post-NACT tumor samples. This pairing enabled us to visually illustrate the biological shift in staining intensity following therapy, which adds unique value beyond the representation of a single time point's median. The graph provides the information of the additional unmatched samples. That said, the staining patterns shown are broadly representative and not extreme outliers in terms of H-score distribution. For transparency, we have added additional panels in **Reviewer Figure 13a** showing other examples closer to the cohort-wide median H-scores to capture better the typical staining range observed.

Reviewer figure 13: Quantification of non-heme iron levels in primary and post-NACT samples. a) Representative IHC images of TFRC and FTH1 in primary and post-NACT samples. Scale bar, 50 μ m. b) Representative IHC images of Pearl Prussian blue in primary and post-NACT samples. Scale bar, 50 μ m. c) Perl's Prussian blue quantification in primary and post-NACT samples. Error bars represent the SEM values. A Welch T-test was used, and quantification was based on the proteomic ROI.

To clarify the nature and localization of iron accumulation, we performed Perl's Prussian Blue staining, a well-established histochemical technique that specifically detects non-heme ferric iron (Fe^{3+}), typically stored within ferritin and hemosiderin complexes. This method does not detect

heme-bound iron, such as that found in hemoglobin, and thus does not stain intact red blood cells. As a result, Perls' staining is particularly effective at distinguishing true intracellular iron storage from hemorrhage-related extracellular iron or heme-bound iron derived from red blood cells. To support these observations, we have now included representative Perls' Prussian Blue staining images from both primary and post-NACT tumors (**Reviewer Fig. 13b**). Quantitative analysis of Perls' staining within the proteomic ROIs revealed a significantly higher % positive area in post-NACT tumors compared to primary tumors ($p = 0.012$, **Reviewer Fig. 13c**).

While we cannot completely exclude some contribution of intratumoral hemorrhage in post-treatment samples, we note that the upregulation of both FTH1 and TFRC at the protein level was observed consistently across the post-NACT cohort in our MS data and validated by quantitative IHC scoring (H-score). Together with Perls' staining, these findings strengthen the conclusion that iron regulatory pathways are actively modulated following chemotherapy, and may reflect intrinsic tumor adaptations rather than passive iron accumulation.

5) The applied statistical tests should be clearly stated in the Figure Legends. It is currently very difficult to judge which bar graph has been assessed with which test (and post-hoc test)

The detailed statistical tests were added to the relevant figure legends.

Reviewer #3 (Remarks to the Author):

The authors have responded adequately to my comments.

I still have a few points to be addressed:

1. Patients with Ewing sarcoma normally undergo surgery only after neo-adjuvant therapy. At initial diagnosis they are normally subjected to core needle biopsy. The authors should double check if the tumor samples used in this manuscript and correspond to initially diagnosis (primary tumor) they were indeed surgical samples. If this is not the case, then their response to Reviewer #1-Comment C (.....Of note, EWS biopsies are typically surgical biopsies,.....) is not accurate.

We accept the reviewer's comment regarding the statement about surgical biopsies. Our statement that 'EWS biopsies are typically surgical biopsies' reflected the historical practice at our institution during the relevant period. For the vast majority of cases included in our cohort, open surgical biopsy was the customary method for obtaining a diagnosis. This was often driven by established protocols and the need for sufficient diagnostic material at the time. We acknowledge

that core needle biopsies (CNB) are increasingly the standard according to current guidelines. Importantly, we confirm that all analyzed specimens represent initial diagnostic biopsies obtained from the primary tumor prior to any neoadjuvant therapy.

2. Study S-BIAD1597 cannot be found in the BioImage Archive (<http://www.ebi.ac.uk/bioimage-archive>).

Study S-BIAD1597 is not public in the BioImage Archive until the manuscript is accepted. The data can only be accessed via the link provided:

<https://www.ebi.ac.uk/biostudies/bioimages/studies/S-BIAD1597?key=ef4c9d7b-5b96-4cde-bc09-56123235a609>

3. Supplementary Fig. 3b-c. Why the patient numbers do not match for EFS and OS analysis? The observed differences in patient numbers between OS/ EFS groups stem from using a "Kaplan-Meier scan" analysis. Unlike fixed-threshold methods (e.g., median split), the Kaplan-Meier scan identifies the optimal expression cutoff that yields the most significant separation in survival outcomes for each endpoint. As this threshold is data-driven, it is recalculated independently for OS and EFS, depending on the distribution of events and the gene's prognostic relationship with each outcome. Therefore, the differences in sample sizes between FTH1 high vs. low in OS and EFS reflect the independent optimization performed for each survival endpoint.

Supplementary Fig. 3c. FLT plot – you may want to show FLT low in red. We apologize for this error. It has been corrected in the revised version.

Reviewer still finds Figure 1e misleading – FLI1, EWSR1, and ERG and not Ewing sarcoma markers. Ewing sarcoma tumors are driven by EWSR1::FLI1/EWSR1::ERG fusion oncoproteins. CD99 is not a very specific marker. The authors may want to provide further explanation in the figure legend. DO the authors assume the FLI1 protein measurements correspond to the fusion protein for example? the same applies to EWSR1 and ERG.

We agree that FLI1, ERG, and EWSR1 are not Ewing sarcoma-specific markers when considered as standalone proteins. We included them in Figure 1e because they are key components of the EWSR1::FLI1 and EWSR1::ERG fusion oncoproteins, which drive the biology of Ewing sarcoma. However, we mass spectrometry-based proteomics cannot distinguish between the fusion oncoprotein and the wild-type forms of these proteins without fusion-junction peptides, based on sequencing of each patient, which is not ethically approved at this point.

We have revised the figure legend of Figure 1e to clarify this point and explicitly state that the protein measurements are shown for their biological context, not as diagnostic biomarkers.

These clarifications appear in the revised manuscript and in the figure legend.

Reviewer #4 (Remarks to the Author):

The authors have responded to prior comments: new data have been added to address ferroptosis resistance and conclusions tempered appropriately. Two minor corrections are suggested: The EWS::FLI1 fusion should be labeled with :: per latest nomenclature standard and it is better called a "fusion gene" rather than a "transgene".

We thank the reviewer for this comment and corrected the term throughout the manuscript.

Reviewer #5 (Remarks to the Author):

RESPONSE TO REVIEWERS

Reviewer #1:

I thank the authors for undertaking an extensive revision of the manuscript and have largely address most of my comments.

Given that the authors now show that the removal of ECM proteins has no impact on the analysis and interpretation of the data, I would request that this step be removed from the pre-processing in the final manuscript. As this reviewer suspected, the authors have confirmed that this step is unnecessary - it is important that the sarcoma field is aware of this and is not misled that this pre-processing step is required to remove such "contaminants". Furthermore, the authors should also report any ECM proteins that were quantified in this study as part of the final proteomic dataset as this is important for the EWS field moving forward given that the TME is an increasingly key area of research as a potential source of targets and biomarkers.

We thank the reviewer for appreciating our extensive revision. As our analyses demonstrated, removal of ECM proteins did not affect the main outcomes. All quantified ECM proteins are included in the final proteomic dataset (see Methods and Supplementary Table S1). We have added the following text referring to this analytical step: "For downstream analyses, we applied a focused filtering step, excluding proteins annotated as blood-derived or extracellular matrix (ECM)-associated, to emphasize tumor cell-intrinsic proteomic features. Removal of ECM proteins was performed to minimize uncertainty arising from the mixed cellular origin of ECM proteins in bulk tissue proteomics, as their contribution could reflect either tumor or stromal compartments. This filtration does not suggest that these proteins are irrelevant to this tumor type. To confirm that ECM removal did not bias our overall conclusions, we compared principal component analysis (PCA) distributions of the dataset before and after filtering. We observed no significant changes in global sample distribution (Supplementary Fig. S7a–b). The complete, unfiltered protein dataset, including all quantified proteins and ECM proteins, is available in Supplementary Table S1d to support future studies addressing Ewing Sarcoma and its tumor microenvironment including ECM". We added a new Supplementary Figure S7 showing Principal Component Analysis (PCA) plots of the whole proteome and ECM-filtered datasets. The overall distribution of the sample points is highly similar in both plots, demonstrating that ECM removal did not cause a significant change in the global structure of the data. We wish to avoid changing all figures at this point, as we also believe that for the intracellular cancer proteins, our approach is more accurate.

Reviewer #2:

The further revised manuscript by Gordon et al. entitled ‘Proteomic landscapes of Ewing sarcoma unravel immunological regulation of tumor progression’ reports on a relatively large series of Ewing sarcoma (EwS) tumor samples that have been subjected to mass-spectrometry based proteomics analyses. While this reviewer acknowledges the additional effort made by the authors that partially addressed significant concerns, the manuscript remains a rather exploratory analysis that largely stays at the phenotypic description. The strength of the paper lies in the unique and large proteomics dataset with matched clinical annotation, which would – if (and only if) made freely available to the scientific community – constitute a strong argument for publication. However, the authors still overinterpret some

We thank the reviewer for the thorough review of our work and the appreciation of the value of our proteomic work. We regret that this reviewer did not find the validations sufficient. Nevertheless, we addressed the reviewer’s comments according to the editor’s advice, as detailed below.

1) This Reviewer thanks the authors for their efforts to generate 3D spheroid models, which is an improvement per se, and recognizes their efforts to generate a second model for “chemoresistance” in EwS (derived from RDES cells). Yet, the minimal differences in sensitivity towards Doxorubicin and the modest differences in sensitivity towards Etoposide are not convincing to declare these new models as “chemoresistant”. The situation is even worse for the previously reported A673 model. This whole set of experiments, while going in the right direction, is an example for an overstatement. The paper would be better if these weak data are omitted from the otherwise strong resources paper and would better fit in a solid follow-up paper with truly convincing chemoresistance models. It is simply not convincing that if even the “better” model can still be killed by Doxorubicin in doses at the nanomolar range, to declare them as “resistance models”. Accordingly, conclusions (e.g. regarding ferroptosis) from these so-called “resistance” models are not valid. The observed differences in ferroptosis could be due to multiple other factors apart of the so-called “resistance”, e.g. by metabolic selection bias during the long-term culture (see Methods). Similarly, the minimal differences in protein abundance of “ferroptosis-related” proteins in “resistant” versus parental cells become only visible if the Y-axis of the shown graph is cropped (data between 2 and 16 AU not shown). The abundance might be statistically significant but the actual difference is minimal and the biological relevance of these differences is unclear. The strong resources aspect of this paper is actually diluted by the attempt to “validate” potential findings from the clinical cohort in preliminary experiments. This Reviewer strongly recommends to take these data out of this paper and to validate them thoroughly in follow-up studies.

We appreciate the reviewer's feedback regarding the chemoresistance models but were disappointed to see the suggestion to remove much of the data. We also disagree about the use of the term chemo-resistant. Resistance is always relevant only to a specific drug concentration, and no cell is expected to be resistant to high drug doses. Therefore, calling cells 'resistant' is a common practice even when it is observed as an increase in the IC50. Nevertheless, following this comment, rather than using the term chemo-resistance, we explained that the cells are adapted after prolonged culture with the chemotherapeutic drugs. We also removed the term 'resistant' in several places in this section. In addition, we moved the A673 results to the supplementary material and only retained RDES data and presented these data as preliminary. Overall, we significantly toned down our claims and stated that future work is needed for robust mechanistic validation. The resources aspect and clinical proteomics remain central to the revised manuscript, as it was before.

Similar to point #1, the role of neutrophils and necrosis is a weakening aspect of this paper. In their rebuttal letter, the authors refer to a previous paper of the same group (Shukrun et al., Cancer Sci., 2024; doi:10.1111/cas.15992) that postulated that neutrophil extracellular traps (NETs) are associated with poor outcome. It should be noted that this previous study was based on a very small cohort (n=46) of EwS patients of which only n=6 died from disease. Some of the main conclusions of this paper are based on graphs such as this presented in Figure 3b and c that show highly overlapping data points across groups and only minimal changes overall.

These figures of the paper of Shukrun et al. (Cancer Sci., 2024) shows that patients with relapse or death of disease have actually more or less the same levels of NETs. The result may be statistically significant (which appears questionable given the massive overlap of data points), but are they biologically and/or clinically relevant? What is the clinical meaning if a patient has e.g. a "NETs release (%) at diagnosis" of 50%? The patient could easily fit in either group (relapse vs no relapse; live vs dead). This Reviewer is not convinced that the abundance of neutrophils and/or NETs is of any clinical meaningful relevance for patient prognosis, risk-stratification, and decision-making. Hence, also the findings of the current study, which are likely based on an extension of the previous cohort, are likely not clinically relevant. Again, the paper would be better off by omitting overstatements on the prognostic/clinical role of these findings and by presenting the current paper as a pure but honest resources article. Given the prior data (Shukrun et al. 2024) and data of the current study, this Reviewer again recommends to either strongly tone down on the relevance of neutrophils and NETs in EwS or even better to take these data out and thoroughly validate them in an independent follow-up study.

Following this reviewer's comment and the editor's feedback, in the revised manuscript, we have moved this data to the Supplementary Information and further toned down our claims, now presenting the association as a preliminary observation, clearly stating that validation in larger cohorts is necessary before any prognostic utility can be claimed.

Reviewer #3:

The authors have adequately addressed my previous comments.

We thank the reviewer for the thoughtful evaluation and are pleased that our revisions have addressed the previous concerns.

Response to Reviewers' Comments

Reviewer #1 (Remarks to the Author)

The authors have addressed my comments in this latest version of the manuscript.

We thank the reviewer for the positive evaluation of our revised manuscript.

Reviewer #2 (Remarks to the Author)

Thank you for the thoughtful revisions. The main text is now appropriately tempered; however, the abstract still reads more assertively than the data support.

In particular, the abstract retains bold claims about chemo - resistance and includes the sentence:

> “We found ferroptosis inhibition as a potential mediator of EWS chemo - resistance and identified novel subclasses of EWS that link the tumor immune landscape with DNA damage repair, ubiquitin - related proteins, and patient prognosis. Multiplexed immunofluorescence imaging supported the association between poor patient prognosis and tumor neutrophils, and the association of macrophages and T - cells with a favorable prognosis.”

As currently written, this phrasing implies causality that I do not see demonstrated in the manuscript. If you wish to keep these points in the abstract, I recommend reframing them as correlations or hypotheses and aligning the strength of language with the evidence shown. For example, consider wording such as:

> “Our analyses suggest that ferroptosis pathways may be associated with chemotherapy response in EWS, and we delineate molecular subclasses that correlate the tumor immune landscape with DNA damage repair, ubiquitin - related proteins, and patient outcome. Multiplexed immunofluorescence indicate possible associations between neutrophils and poorer prognosis, and between macrophages/T cells and more favorable prognosis.”

Relatedly, the title remains unchanged and, in my view, is potentially misleading. The term “immunological regulation of tumor progression” implies a demonstrated causal role; the data presented support association rather than regulation. Please adjust the title to avoid implying

causality. A more accurate and appropriately cautious title might be:

“Proteomic landscape of Ewing sarcoma primary tumors and metastases,”

To summarize: the dataset is sound and valuable, and of clear interest. My recommendation is to harmonize the abstract (and title) with the more measured conclusions in the main text by avoiding causal language where only associations are shown, and by defining terms precisely. This will strengthen the manuscript and ensure readers take away conclusions that are fully supported by the data.

We thank the reviewer for the appreciation of our revision. In accordance with the reviewer's request, we changed the title and abstract of the manuscript.